# PROSAR: PROTOTYPE-GUIDED SEMANTIC AUGMENTATION AND REFINEMENT FOR TIME SERIES CONTRASTIVE LEARNING

## ABSTRACT

Contrastive learning has advanced the representation learning of vision, language, and graphs, yet its success hinges greatly on the data augmentation that helps preserve semantic contents while providing the view diversities. Multivariate time series, however, are noisy, non-stationary in nature, and largely opaque to the human inspection. Therefore, a direct use of the the hand-crafted transforms, such as jitter and scaling, may unfortunately destroy the critical temporal cues or introduce false negatives, weakening the performance of downstream tasks. To address this, we propose ProSAR, a prototype-guided semantic augmentation and refinement framework for time series contrastive learning. Most critically, ProSAR's approach is founded on an information-theoretic principle for co-designing the semantic data augmentations and learnable prototypes, aiming to generate views that maximize the information about an associated semantic prototype while discarding the prototype-irrelevant content. ProSAR then implements this by introducing a novel prototype-conditioned semantic segment extraction mechanism, where the temporal characteristic segments are identified based on their dynamic time warping (DTW) alignment to these learnable time-domain prototypes, ensuring that the generated views can capture high-level semantic events. Building upon these temporal characteristic segments, the targeted augmentations, operating in both the time and frequency domains and informed by the DTW alignments, can thus preserve the temporal dynamics while constructing views that adhere to the information-theoretic objectives. Furthermore, prototypes are dynamically refined in a feedback loop, where the latent representations of these prototypes are refined via clustering under the prototypical contrastive training, and in turn guide evolution of the time-domain prototypes through a decoding consistency mechanism, thus fostering a progressive learning of robust representations. Experiments on diverse time-series benchmarks demonstrate that ProSAR outperforms recent contrastive and self-supervised representation learning methods in the downstream forecasting and classification tasks.

## 1 INTRODUCTION

The proliferation of time series data across diverse domains, from healthcare (Miotto et al., 2016), finance (Heaton et al., 2017) to industrial IoT (Syafrudin et al., 2018) and human activity recognition (Wang et al., 2019), has underscored the imperative need for an analytical tool for the effective representation learning. However, the high costs and numerous efforts associated with manual labeling often render the commonly-used supervised learning impractical, motivating the rapid development of self-supervised learning (SSL) paradigms (Jaiswal et al., 2020; Misra & Maaten, 2020). Among them, contrastive learning (CL) has emerged as a particularly successful approach (Le-Khac et al., 2020; Chen et al., 2020; He et al., 2020), which aims to learn the representations by maximizing the agreement between different views of the same data instance while minimizing the agreement with the views of other data instances, where the views are typically generated through data augmentation (Shorten & Khoshgoftaar, 2019). Alongside some direct efforts to improving augmentation, the field has advanced on parallel fronts—refining the contrastive objective ((Lee et al., 2024)), exploiting multi-frequency structure ((Duan et al., 2024)), and modeling relative similarity (Xu et al., 2025), yet the view-generation module itself remains a fundamental bottleneck.

Despite the success of CL, its efficacy on the time series data is profoundly challenged by the difficulty of designing data augmentations that can preserve some crucial temporal semantics, while ensuring a sufficient view diversity (Wen et al., 2021; Luo et al., 2023). Unlike the images or text, where human intuition can often guide a semantic-preserving transformation, the complex, often non-intuitive, temporal structures in time series make the augmentation design a formidable task (Zheng et al., 2024). Many existing augmentation methods are either based on the hand-picked heuristics, e.g., jittering, scaling, permutation (Zhang et al., 2022; Yue et al., 2022; Eldele et al., 2021), or a direct adaptation from the other modalities, at the risk of incurring distortion or destruction of the pivotal temporal patterns and semantic information (Tian et al., 2025). This cross-modality incompatibility may lead to mismatched patterns and even inadvertent generation of false negative pairs, thus hindering the learning of a robust representation (Meng et al., 2023; Chuang et al., 2020).

To address this, current CL methodologies for time series has strived for the semantic augmentation from an information-theoretic perspective. Theoretical groundwork like the InfoMin principle (Tian et al., 2020) posits that desirable views reduce the mutual information while preserving the task-relevant signals. InfoTS (Luo et al., 2023) extends InfoMin with a criteria to balance between the augmentation fidelity and variety, by further introducing a meta-learner for the adaptive selection. Frameworks such as AutoTCL (Zheng et al., 2024) learn to factorize the instances for augmentation, by separating the informative parts from the task-irrelevant ones. Additionally, FreRA (Tian et al., 2025) learns the frequency-importance, by preserving critical subbands while perturbing non-critical ones to produce the semantics-preserving views. Nonetheless, despite these advances in the semantic-aware augmentation, the preserved content is still learned and inferred indirectly, rather than tied to the explicit and temporally aligned anchors, thus limiting the interpretability and controllability, which can be alleviated by prototype-based anchors. Concurrently, a parallel line of research has utilized prototypes within the contrastive objective to mitigate the false negative problem inherent in instance-wise comparisons (Chuang et al., 2020). Methods such as Prototypical Contrastive Learning (Li et al., 2021a) and MHCCL (Meng et al., 2023) have shown this to be effective for learning a structured latent space. However, they use prototypes typically at the objective level, without leveraging them to guide the upstream data augmentation process.

We are thus motivated to propose in this paper a novel contrastive learning framework for time series, named ProSAR, i.e., prototype-guided semantic augmentation and refinement. ProSAR's core innovation lies in its information-theoretic foundation for the co-design of semantic data augmentations and learnable prototypes. Theoretically, we formalize the semantic data augmentation through lens of the information bottleneck (IB) principle (Tishby et al., 2000), positing that the optimal views for augmentation should maximize the information about their associated semantic prototype, while discarding prototype-irrelevant content from the original instance. Specifically, We further operationalize this theoretical principle by employing learnable time-domain prototypes in conjunction with the dynamic time warping (DTW) technique (Cuturi & Blondel, 2017), to explicitly identify and extract semantically coherent segments from the raw time series. We then apply an augmentation strategy to these segments, informed by their alignment with the time-domain prototypes and by incorporating the time-frequency characteristics, to construct views that adhere to our information-theoretic objectives. We also utilize the latent-space prototypes, refined via clustering of instance representations under the prototypical contrastive training, to guide evolution of the time-domain prototypes through a decoding consistency mechanism, thus creating a positive feedback loop. In contrast to the prior augmentation methods, ProSAR elevates prototypes to serve as explicit anchors for a more interpretable view generation. Furthermore, unlike prototypical CL approaches that primarily utilize prototypes at the objective level, ProSAR establishes a feedback loop where these refined prototypes can actively guide the augmentation policy. Our main contributions can be summarized as follows:

- We propose a novel CL framework for time series, named ProSAR, which leverages the information bottleneck principle to guide a co-design of the data augmentation and prototype learning.

- We design a prototype-guided semantic view generation mechanism that employs learnable time-domain prototypes to explicitly distinguish between task-relevant and irrelevant temporal segments. Guided by our information-theoretic objective, this mechanism then augments these identified segments to selectively preserve semantic content while discarding task-irrelevant variations.

- We further develop a dual-prototype refinement strategy that establishes a positive feedback loop. Under the prototypical contrastive training, the latent-space prototypes learned via clustering are

used to guide the update of the time-domain prototypes through a decoding consistency mechanism, ensuring that the augmentations become progressively aligned with the evolving semantic structure.

- We conduct comprehensive experimental evaluation to demonstrate that ProSAR achieves a superior performance over the state-of-the-art methods in learning discriminative and semantically grounded time series representations.

## 2 RELATED WORK

**Self-Supervised and Contrastive Representation Learning for Time Series.** Early sequence SSL demonstrate that temporal structure can be learned without labels via predictive or contrastive objectives, e.g., CPC (Oord et al., 2018) and TLoss (Franceschi et al., 2019). For time series, methods exploit the temporal neighborhoods (TNC) (Tonekaboni et al., 2021), multi-level context and subseries consistency (TS2Vec) (Yue et al., 2022), disentangled seasonal–trend factors (CoST) (Woo et al., 2022), and Transformer encoders for multivariate sequences (TST) (Zerveas et al., 2021). For vision data, SimCLR (Chen et al., 2020) and MoCo (He et al., 2020) catalyze the modern contrastive learning. Recent time-series advances include SoftCLT, which uses soft assignments at instance and timestamp levels (Lee et al., 2024), and MF-CLR, which imposes cross-frequency consistency with a hierarchical mechanism (Duan et al., 2024). TimesURL integrates the frequency-temporal augmentation, hard negatives, and joint reconstruction (Liu & Chen, 2024). PPT emphasizes that patch order matters for time series, and introduces a patch-order prediction pretext task to learn temporally aware representations (Kim et al., 2025).

**Augmentation and View Generation for Contrastive Learning.** Constructing views that are both diverse and semantics-preserving is central to contrastive learning (Tian et al., 2020). Surveys of time-series augmentation systematize common hand-crafted transforms and analyze their limitations (Wen et al., 2021). TS-TCC adopts heuristic policies, using jitter and scaling as weak transformations and permutation with jitter as strong ones (Eldele et al., 2021). TF-C promotes view quality by aligning time- and frequency-domain representations during pretraining, encouraging consistency across complementary domains (Zhang et al., 2022). From an information-theoretic perspective, InfoMin prescribes reducing mutual information between views while retaining task-relevant content (Tian et al., 2020). Building upon this, InfoTS scores the candidate transforms and adaptively selects them to balance between the fidelity and diversity (Luo et al., 2023). AutoTCL factorizes each instance into informative and task-irrelevant components, and learns parametric augmenters to target these parts (Zheng et al., 2024). A frequency-aware line further learns the band importance: FreRA preserves the critical bands while perturbing non-critical ones to form semantics-preserving views (Tian et al., 2025). Beyond augmentation itself, AutoCL automatically searches for the data augmentations, embedding transformations, contrastive pair construction, and contrastive losses (Jing et al., 2024).

**Prototype- and Cluster-Aware Contrastive Objectives.** Incorporating prototypes or cluster structure into the objective can inject semantic priors beyond instance discrimination. For example, PCL introduces ProtoNCE to pull samples towards the assigned prototypes (Li et al., 2021a). SwAV contrasts cluster assignments via online clustering and assignment consistency (Caron et al., 2020). Contrastive Clustering jointly performs the instance- and cluster-level contrast (Li et al., 2021b), and Graph Contrastive Clustering extends the cluster-aware contrast to graphs (Zhong et al., 2021). For time series, MHCCL uses the hierarchical clustering with downward/upward masking to refine prototypes and perform cluster-wise contrast (Meng et al., 2023). AimTS proposes a two-level prototype-based contrast with series–image cross-modal contrast to better leverage existing augmentations, but does not use the prototypes to drive augmentation itself (Chen et al., 2025). Our approach differs by elevating the learnable prototypes to explicit semantic carriers that guide upstream view generation under the information-theoretic co-design. Departing from the prior augmentation strategies, ProSAR utilizes prototypes as explicit guides for a more interpretable view generation process. Moreover, extending beyond prototypical methods that use prototypes solely at the objective level, ProSAR implements a closed-loop mechanism, where these prototypes are dynamically refined to steer the augmentation policy.

## 3    PROPOSED METHOD

In this section, we introduce ProSAR, a novel self-supervised framework for learning semantically rich representations of time series. ProSAR distinctively integrates learnable prototypes into the information-theoretic design of data augmentations. We first establish the theoretical principles in Section 3.1, demonstrating how prototypes can guide the creation of semantically consistent views and how, in turn, these views are used to refine the prototypes. Subsequently, Section 3.2 details the architectural components and learning objectives that implement this co-design, culminating in a robust and theoretically grounded approach to time series representation learning. For clarity, the key notations employed throughout this paper are summarized in Table 1.

### 3.1    INFORMATION-THEORETIC FOUNDATIONS FOR PROTOTYPE-ENHANCED SEMANTIC AUGMENTATION

We begin by formalizing semantic data augmentation through the lens of information theory. Our objective is to generate the augmented views $\tilde{X}$ of an input time series $X$ that preserve its essential semantic content represented by a latent variable $C$, while discarding irrelevant information. A core challenge in self-supervised learning is that $C$ is unknown. This motivates our introduction of the learnable prototypes $\mathbf{P}$ (specifically, the latent prototypes $\{p_k^z\}$) as tractable proxies for $C$, enabling a co-design of the augmentation strategies and prototype refinement.

#### 3.1.1    INFORMATION BOTTLENECK AND ITS CHALLENGE IN SELF-SUPERVISED LEARNING

We let $X$ denote a random variable representing a raw time series, and $C$ be a latent variable encapsulating its core semantic information. An augmentation $T$ generates an augmented view $\tilde{X} = T(X)$. The design of $T$ is guided by the Information Bottleneck (IB) principle, which employs mutual information (MI) to quantify dependence between variables. The principle formalizes this objective as a trade-off: maximizing the MI with the latent semantics $I(C; \tilde{X})$ to retain essential meaning, while simultaneously minimizing the MI with the input $I(X; \tilde{X})$ to achieve compression:

$$\max_T I(C; \tilde{X}) - \beta I(X; \tilde{X}), \tag{1}$$

where $\beta > 0$ balances semantic fidelity and compression. This aligns with the InfoMin principle for contrastive learning (Tian et al., 2020). In SSL, $C$ is unknown, rendering direct optimization of Eq. (1) infeasible and necessitating a data-driven proxy for $C$.

#### 3.1.2    PROTOTYPES AS SEMANTIC PROXIES AND CONDITIONAL INFORMATION BOTTLENECK

To address the unknown $C$, we introduce $K$ learnable prototypes, $\mathbf{P}$, representing distinct semantic clusters (see Table 1). A prototype assignment variable $P \in \{1, \ldots, K\}$ indicates the semantic cluster most associated with an input time series $X$. In our framework, this assignment is operationalized in the input space: $P$ is determined by identifying the time-domain prototype $p_k^x$ that best aligns with $X$. Thus, $P$ remains a deterministic function of $X$ (i.e., $P = \text{assign}(X, \{p_k^x\})$). Substituting $P$ for $C$ in Eq. (1) yields the *prototype-conditioned IB objective* for augmentation design:

$$\max_T I(P; \tilde{X}) - \beta I(X; \tilde{X}). \tag{2}$$

This links augmentation design to the learned prototypes, setting the stage for their co-design. Compared with PCL (Li et al., 2021a) that uses prototypes at the objective level, ProSAR distinctively leverages these prototypes to actively guide the augmentation generation process itself through an information-theoretic lens.

Table 1: Key Notations for ProSAR, where RV stands for Random Variable.

| Symbol | Description |
|---|---|
| $X, x$ | Time series (RV, instance) |
| $\tilde{X}, \tilde{x}$ | Augmented views of $X, x$ |
| $C$ | Latent semantic variable |
| $\mathbf{P} = \{p_k^x\}$ | Set of $K$ latent prototypes |
| $P$ | Prototype assignment index |
| $f_\theta$ | Encoder network |
| $z$ | Latent representation vector |
| $p_k^x$ | Time-domain prototype |
| $p_k^z$ | Latent prototype |
| $x_S, x_N$ | Semantic, non-semantic parts of $x$ |
| $M_x$ | Binary mask on instance $x$ |
| $I(A; B)$ | Mutual Information |
| $\beta$ | IB objective hyperparameter |
| $\mathcal{L}_{\text{total}}$ | Overall loss function |
| $T$ | Augmentation transformation |
| $D_\psi$ | Decoder network |

To analyze Eq. (2), we first decompose $I(X; \tilde{X})$ by using the fact that $P$ is a function of $X$.

**Proposition 3.1.** *If $P$ is determined by $X$ (i.e., $P = g(X)$ for some function $g$), then $I(X; \tilde{X}) = I(P; \tilde{X}) + I(X; \tilde{X} \mid P)$.*

Substituting this decomposition into Eq. (2), the objective becomes:

$$\max_T I(P; \tilde{X}) - \beta(I(P; \tilde{X}) + I(X; \tilde{X} \mid P)) = \max_T (1 - \beta)I(P; \tilde{X}) - \beta I(X; \tilde{X} \mid P). \quad (3)$$

This transformed objective leads to the following characterization of an optimal augmentation strategy with respect to the current prototype assignments $P$.

**Proposition 3.2** (Prototype-Optimal Augmentation). *For $\beta \in (0, 1)$, an augmentation $T^*$ optimizing Eq. (3) aims to satisfy:*

*(i) $I(X; \tilde{X} \mid P) \to 0$: $\tilde{X}$ retains minimal $X$-information not explained by $P$;*

*(ii) $I(P; \tilde{X}) \to I(P; X)$: $\tilde{X}$ is maximally informative about $P$.*

Please refer to Appendix E for the detailed proof. Proposition 3.2 implements the InfoMin principle (Tian et al., 2020) by using the prototypes. It implies that as prototype assignments $P$ (derived from $f_\theta$ and $\mathbf{P}$) evolve, the optimal augmentation $T^*$ must also co-evolve. Condition (*i*) requires the augmentation to discard information within $X$ that is irrelevant to the prototype assignment $P$, thereby isolating the informative part from the task-irrelevant part. Concurrently, Condition (*ii*) ensures that $\tilde{X}$ remains maximally informative about $P$, preserving the core semantic signal, which is analogous to AutoTCL (Zheng et al., 2024). However, ProSAR explicitly defines this part through its relevance to the learned prototypes $\mathbf{P}$.

### 3.1.3 CO-DESIGN AND ITERATIVE REFINEMENT OF PROTOTYPES AND AUGMENTATIONS

Eq. (3) underscores the critical interdependency for co-design: the optimal augmentation $T$ depends on the quality of the time-domain prototypes $\{p_k^x\}$ that determine the assignment $P$, while these prototypes are themselves refined by learning from the augmented views. This creates a synergistic and iterative process.

**Data Augmentation Guided by Time-Domain Prototypes:** Proposition 3.2 indicates that augmentations should make views highly informative about their assigned prototype $P$, while discarding $P$-irrelevant information from $X$. Since $P$ is determined by alignment with time-domain prototypes $\{p_k^x\}$, this motivates us to transform $X$ based on the segments that are semantically relevant to these time-domain anchors.

**Dual Prototype Refinement via Augmentations:** The refinement of the time-domain prototypes $\{p_k^x\}$ is achieved through a synergistic process involving the encoder $f_\theta$ and a corresponding set of latent prototypes $\{p_k^z\}$. Specifically, the encoder learns from augmented views to produce robust latent representations $(z, \tilde{z})$. These representations are then used to update the latent prototypes $\{p_k^z\}$ to better capture high-level semantic clusters. In turn, these refined latent prototypes guide the evolution of the time-domain prototypes $\{p_k^x\}$, creating an indirect but powerful refinement pathway.

This interplay between the time-domain guidance and latent-space learning fosters a positive feedback loop: *i*) better time-domain prototypes lead to more semantically focused augmentations; *ii*) these augmentations provide clearer signals for learning improved latent representations and, consequently, more accurate latent prototypes; and *iii*) finally, refined latent prototypes enable the generation of higher-quality time-domain prototypes. The term $-\beta I(X; \tilde{X} \mid P)$ in Eq. (3) actively drives this loop by promoting challenging positive views that are dissimilar to $X$ in $P$-irrelevant aspects. Please refer to Appendix E for more details and proofs.

### 3.2 PROSAR: FRAMEWORK IMPLEMENTATION

Building upon the information-theoretic principles of co-design and iterative refinement from Section 3.1, we implement ProSAR in a practical framework. Specifically, Figure 1 provides a visual schematic of the ProSAR architecture, detailing the main components and their interactions, which are further elaborated in the subsequent sections.

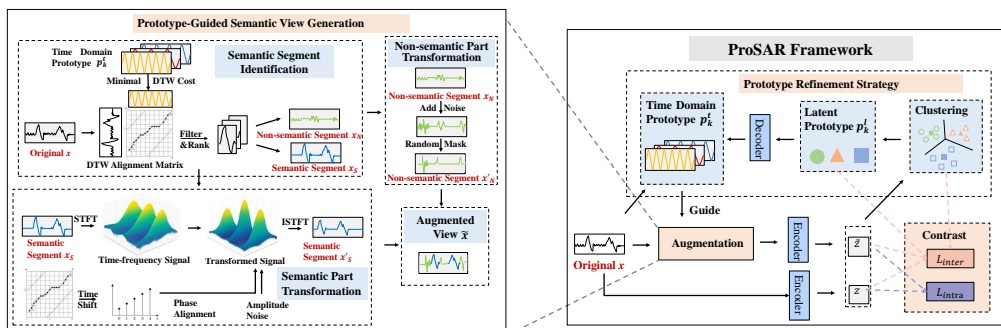

Figure 1: Overview of the ProSAR framework. The left panel details the prototype-guided semantic view generation process. The right panel illustrates the overall training architecture, including augmentation, encoding, contrastive objectives, and the prototype refinement loop.

### 3.2.1 PROTOTYPE-GUIDED SEMANTIC VIEW GENERATION

To implement Proposition 3.2 without access to the true $C$, we employ learnable *time-domain prototype sequences* $\{p_k^x\}$ (Table 1). These act as semantic anchors in the input space, whose learning and connection to latent prototypes $p_k^z$ are detailed in Section 3.2.2. An augmented view $\tilde{x}$ is generated from an input instance $x$, as follows.

(a) **Semantic Segment Identification:** Multiple informative segments, denoted as a set $\{S_{\text{raw},j}\}$, are identified within the input time series $x$. This process leverages the alignment of $x$ with relevant time-domain prototypes $\{p_k^x\}$, computed using Dynamic Time Warping (DTW) (Cuturi & Blondel, 2017). For the input time series $x$, we first identify its best-matching time-domain prototype via DTW. Subsequently, sub-segments of $x$ with a high-quality alignment (i.e., low cumulative DTW path cost) to this specific prototype are considered candidate semantic segments. These candidates are then typically filtered based on their DTW alignment scores to select the most relevant segments. Optionally, to further refine this selection by explicitly seeking portions of $x$ most semantically related to the concepts embodied by the prototypes, a measure of mutual information between a candidate segment and its guiding prototype can be maximized. A binary mask $M_x$ is then defined, where $M_x(t) = 1$ if time step $t$ falls within any of the selected semantic segments $S_{\text{raw},j}$, and $M_x(t) = 0$ otherwise. This mask identifies the semantic part $x_S = x \odot M_x$ and the non-semantic part $x_N = x \odot (1 - M_x)$. This entire step utilizes prototypes to estimate the informative parts of $x$ relevant to its underlying semantic structure as captured by the prototype assignments $P$.

(b) **Transformation:** Once $x_S$ and $x_N$ are defined, the augmented view $\tilde{x}$ is constructed by transforming these parts to satisfy the conditions of Proposition 3.2.

1. **Transform Semantic Part ($x_S$):** The identified semantic part $x_S$ undergoes the following two transformations to produce $x_S'$.
- *DTW-Guided Temporal Alignment in the Frequency Domain:* The DTW alignment path between $x_S$ and the guiding $p_k^x$ reveals local temporal misalignments. This information is used to apply phase compensation in the frequency domain. Identified local time shifts can be compensated by adjusting the phase of the STFT representation of $x_S$. This aims to normalize temporal variations, preserving semantic integrity and helping to ensure $I(P; x_S') \approx I(P; x_S)$.
- *Controlled Noise Injection:* Further, controlled noise, specifically by perturbing frequency domain amplitudes, is applied to the temporally-aligned segment. $x_S'$ is obtained after this step, where the noise helps in reducing superficial information in $x_S$ not essential for identifying $P$, thereby contributing to reducing $I(X; \tilde{X} \mid P)$. Let the result of the two transformations on $x_S$ be $x_S'$.

2. **Perturb Non-Semantic Part ($x_N$):** The non-semantic part $x_N$ is heavily modified to produce $x_N'$. This involves first applying strong noise perturbation to $x_N$, followed by random sub-segment masking within $x_N$. This two-fold process aims to thoroughly corrupt information in $x_N$ that is irrelevant to $P$ for these non-semantic regions.

3. **Construct Augmented View ($\tilde{x}$):** The final augmented view is assembled by combining the transformed semantic part and the perturbed non-semantic part: $\tilde{x} = (x_S' \odot M_x) + (x_N' \odot (1 - M_x))$.

Effectively, $\tilde{x}$ largely preserves the prototype-relevant semantics from $x_S$ while minimizing other information from $x_S$ and heavily disrupting $x_N$.

This adaptive process, which precisely manipulates the prototype-relevant part $x_S$ and the irrelevant part $x_N$ based on guidance from $p_k^{\text{x}}$, is a direct attempt to instantiate the optimal augmentation strategy. The nature and learning of these time-domain prototypes $p_k^{\text{x}}$ and their intrinsic link to latent prototypes $p_k^{\text{z}}$ are crucial for semantic consistency and are further detailed below.

### 3.2.2 Prototype Refinement Strategy

Dynamic prototype updates are crucial for capturing evolving semantic structures. The primary mechanism for refining latent prototypes $p_k^{\text{z}}$ is Latent-Space Clustering, aligning with the iterative improvement principles in Section 3.1.3. We employ the FINCH (Sarfraz et al., 2019) hierarchical clustering algorithm, applied to a combined set of instance representations from original views $\{z_i = f_\theta(x_i)\}$ and their corresponding augmented views $\{\tilde{z}_i = f_\theta(\tilde{x}_i)\}$. Unlike methods such as MHCCL (Meng et al., 2023) that filter false negatives during clustering, ProSAR tackles this challenge at its source. By using prototypes to guide the semantic augmentation process (Section 3.2.1), our framework generates more coherent positive pairs, intrinsically minimizing the risk of false negatives. Following clustering, a set of representative cluster centroids is computed. These centroids then guide the update of the resident latent prototypes $p_k^{\text{z}}$. Specifically, each computed cluster centroid is utilized to update the closest latent prototype $p_k^{\text{z}}$ via an Exponential Moving Average (EMA), which ensures a smooth evolution of these resident prototypes.

The refinement of these latent prototypes $p_k^{\text{z}}$ is coupled with strategies for obtaining and refining the time-domain prototypes $p_k^{\text{x}}$ that directly guide augmentation. Two complementary approaches support this: Input-Space Anchoring and Latent-to-Time-Domain Decoding Consistency. The influence of these can be adaptively managed.

- **Input-Space Anchoring (ISA):** This strategy provides initial, data-driven estimates for time-domain patterns. Raw input segments $S_{\text{raw}}$ are clustered in the time domain. The resulting cluster centroids, denoted as $\{c_k\}$, serve as direct, empirically derived reference points.
- **Latent-to-Time-Domain Decoding Consistency:** This ensures refined latent prototypes $p_k^{\text{z}}$ map to meaningful time-domain patterns. A decoder $D_\psi$ generates time series $\hat{p}_k^{\text{x}} = D_\psi(p_k^{\text{z}})$ from each latent prototype.

The final active time-domain prototypes $p_k^{\text{x}}$ are then updated by fusing information from both sources: a weighted combination of the empirically derived centroids $\{c_k\}$ from ISA and the decoder outputs $\{\hat{p}_k^{\text{x}}\}$. This comprehensive and adaptive strategy aims to refine both latent and time-domain prototypes to robustly capture true semantics, supporting the iterative improvement principles.

### 3.2.3 Learning Objectives

The encoder $f_\theta$ learns representations $z_i = f_\theta(x_i)$ and $\tilde{z}_i = f_\theta(\tilde{x}_i)$. The learning process is driven by a two-component contrastive loss function. The first component is an intra-instance temporal contrastive loss ($L_{\text{intra}}$), inspired by the local contrast mechanism (Tonekaboni et al., 2021), designed to model fine-grained temporal patterns. The second component is an inter-instance semantic contrastive loss ($L_{\text{inter}}$). This loss learns robust semantic relationships by contrasting instances based on their association with learned prototypes, effectively aligning instances to these semantic anchors. The detailed expressions for these losses are provided in Appendix A.

The overall learning objective is a weighted sum of these components:

$$\mathcal{L}_{\text{total}} = \lambda_{\text{intra}} L_{\text{intra}} + \lambda_{\text{inter}} L_{\text{inter}}. \tag{4}$$

## 4 Empirical Evaluation

To thoroughly evaluate the performance of ProSAR, we conduct extensive experiments on both the time series forecasting and classification tasks. Furthermore, detailed ablation studies are performed to analyze the contribution of each component within the ProSAR framework. Our experimental setup, including hyperparameter configurations and full results, is detailed in Appendix B.

Table 2: Univariate time series forecasting results.

| Dataset | ProSAR | | AutoTCL | | FreRA | | PPT | | TimesURL | | InfoTS | | TS2Vec | | TNC | | TS–TCC | | CoST | |
|---|---|---|---|---|---|---|---|---|---|---|---|---|---|---|---|---|---|---|---|---|
| | MSE | MAE | MSE | MAE | MSE | MAE | MSE | MAE | MSE | MAE | MSE | MAE | MSE | MAE | MSE | MAE | MSE | MAE | MSE | MAE |
| ETTh$_1$ | **0.068** | **0.198** | 0.076 | 0.207 | 0.079 | 0.213 | 0.087 | 0.221 | 0.090 | 0.219 | 0.091 | 0.227 | 0.110 | 0.252 | 0.150 | 0.303 | 0.168 | 0.316 | 0.091 | 0.228 |
| ETTh$_2$ | **0.142** | **0.289** | 0.158 | 0.299 | 0.171 | 0.315 | 0.159 | 0.303 | 0.151 | 0.295 | 0.149 | 0.299 | 0.170 | 0.321 | 0.168 | 0.322 | 0.298 | 0.428 | 0.161 | 0.307 |
| ETTm$_1$ | **0.045** | **0.151** | 0.046 | 0.154 | 0.051 | 0.162 | 0.062 | 0.174 | 0.053 | 0.170 | 0.050 | 0.157 | 0.069 | 0.186 | 0.069 | 0.191 | 0.158 | 0.299 | 0.054 | 0.164 |
| Elec | **0.338** | **0.328** | 0.366 | 0.345 | 0.360 | 0.339 | 0.389 | 0.369 | 0.374 | 0.356 | 0.368 | 0.348 | 0.393 | 0.370 | 0.378 | 0.359 | 0.511 | 0.603 | 0.375 | 0.353 |
| WTH | **0.160** | **0.285** | 0.160 | 0.287 | 0.169 | 0.301 | 0.177 | 0.307 | 0.177 | 0.302 | 0.176 | 0.304 | 0.181 | 0.308 | 0.175 | 0.303 | 0.302 | 0.442 | 0.183 | 0.307 |
| Avg. | **0.151** | **0.250** | 0.161 | 0.258 | 0.166 | 0.266 | 0.175 | 0.275 | 0.169 | 0.269 | 0.167 | 0.267 | 0.185 | 0.287 | 0.188 | 0.296 | 0.287 | 0.418 | 0.173 | 0.272 |

Table 3: Multivariate time series forecasting results.

| Dataset | ProSAR | | AutoTCL | | FreRA | | PPT | | TimesURL | | InfoTS | | TS2Vec | | TNC | | TS-TCC | | CoST | |
|---|---|---|---|---|---|---|---|---|---|---|---|---|---|---|---|---|---|---|---|---|
| | MSE | MAE | MSE | MAE | MSE | MAE | MSE | MAE | MSE | MAE | MSE | MAE | MSE | MAE | MSE | MAE | MSE | MAE | MSE | MAE |
| ETTh$_1$ | **0.625** | **0.566** | 0.656 | 0.590 | 0.646 | 0.584 | 0.750 | 0.650 | 0.731 | 0.645 | 0.784 | 1.622 | 0.788 | 0.646 | 0.904 | 0.702 | 0.748 | 0.635 | 0.650 | 0.585 |
| ETTh$_2$ | 1.213 | 0.819 | **1.191** | **0.815** | 1.397 | 0.893 | 1.529 | 0.928 | 1.514 | 0.926 | 1.474 | 0.914 | 1.566 | 0.937 | 1.869 | 1.053 | 2.120 | 1.109 | 1.283 | 0.851 |
| ETTm$_1$ | **0.396** | **0.434** | 0.409 | 0.441 | 0.445 | 0.467 | 0.562 | 0.580 | 0.561 | 0.584 | 0.568 | 0.521 | 0.628 | 0.553 | 0.740 | 0.599 | 0.612 | 0.564 | 0.409 | 0.439 |
| Elec | **0.159** | **0.264** | 0.175 | 0.272 | 0.182 | 0.278 | 0.213 | 0.312 | 0.202 | 0.299 | 0.289 | 0.376 | 0.319 | 0.397 | 0.387 | 0.446 | 0.511 | 0.602 | 0.165 | 0.268 |
| WTH | **0.412** | **0.451** | 0.423 | 0.457 | 0.429 | 0.462 | 0.445 | 0.466 | 0.447 | 0.469 | 0.455 | 0.472 | 0.451 | 0.474 | 0.441 | 0.466 | 0.483 | 0.535 | 0.430 | 0.464 |
| Avg | **0.561** | **0.507** | 0.571 | 0.515 | 0.620 | 0.537 | 0.700 | 0.587 | 0.691 | 0.585 | 0.714 | 0.781 | 0.750 | 0.601 | 0.868 | 0.653 | 0.895 | 0.689 | 0.587 | 0.521 |

## 4.1 TIME SERIES FORECASTING

**Datasets and Baselines.** Our forecasting evaluation leverages five widely adopted benchmark datasets: ETTh1, ETTh2, ETTm1, Electricity, Weather, which have been extensively utilized for benchmarking and publicly available (Zhou et al., 2021). We compare ProSAR against the state-of-the-art self-supervised time-series methods: TNC (Tonekaboni et al., 2021), TS-TCC (Eldele et al., 2021),TS2Vec (Yue et al., 2022), CoST (Woo et al., 2022), InfoTS (Luo et al., 2023), AutoTCL (Zheng et al., 2024),TimesURL (Liu & Chen, 2024), PPT (Kim et al., 2025) and FreRA (Tian et al., 2025).

**Experimental Setup.** The encoder architecture for ProSAR follows CoST (Woo et al., 2022), utilizing a multi-layer dilated CNN backbone, from which we omit the seasonal feature disentangler module to isolate the impact of our proposed augmentation and prototype mechanisms. The forecasting tasks aim to predict $L_y$ future time steps given the preceding $L_x$ observations. Following the evaluation protocol of Yue et al. (2022), representations learned by the pretrained encoder are frozen, and a linear model with L2 regularization is trained to make the predictions. This linear evaluation protocol is applied uniformly across all contrastive learning baselines for fair comparison. Outputs are $L_y$-dimensional for univariate and $L_y \times F$-dimensional for multivariate series . Performance is quantified using standard Mean Squared Error (MSE) and Mean Absolute Error (MAE) metrics over the various prediction lengths $L_y$.

**Quantitative Results.** The average forecasting performance across all the datasets for univariate and multivariate settings is presented in Table 2 and Table 3. ProSAR consistently surpasses AutoTCL, indicating that the prototype-guided semantic augmentation performs better than the other augmentation schemes. In the univariate setting, ProSAR achieves a 6.2% lower MSE and a 3.5% lower MAE, reflecting stronger representations that preserve key dynamics while ensuring view diversity. In the multivariate setting, ProSAR also leads on average with a 1.8% MSE reduction and a 1.6% MAE reduction, winning on 4 of 5 datasets when averaging across prediction lengths per dataset.

## 4.2 TIME SERIES CLASSIFICATION

**Datasets and Baselines.** For the time series classification task, ProSAR is evaluated on the comprehensive UEA multivariate time series archive (Dau et al., 2019). We compare ProSAR against the state-of-the-art self-supervised time-series methods: TNC (Tonekaboni et al., 2021), TS-TCC (Eldele et al., 2021), TS2Vec (Yue et al., 2022), InfoTS (Luo et al., 2023), AutoTCL (Zheng et al., 2024), TimesURL (Liu & Chen, 2024), PPT (Kim et al., 2025) and FreRA (Tian et al., 2025).

**Experimental Setup.** The encoder architecture from TS2Vec (Yue et al., 2022) is adopted for ProSAR in these classification experiments. The self-supervised pretraining strategy follows the forecasting tasks. Evaluation proceeds in a standard supervised manner: a Radial Basis Function

Table 4: Classification result of the UEA dataset.

| Metric | ProSAR | AutoTCL | FreRA | PPT | TimesURL | InfoTS | TS2Vec | TNC | TS–TCC |
|---|---|---|---|---|---|---|---|---|---|
| Avg. ACC | **0.764** | 0.742 | 0.754 | 0.735 | 0.752 | 0.730 | 0.704 | 0.670 | 0.668 |
| Avg. RANK | **1.867** | 3.067 | 2.900 | 3.200 | 2.233 | 3.133 | 4.367 | 5.500 | 5.367 |

Table 5: Ablation studies.

| Metric | ProSAR (Full) | GST | No-$x_S$-T | No-$x_N$-T | StaticProto | RandAug | Jitter | Cutout |
|---|---|---|---|---|---|---|---|---|
| Avg. MSE | **0.151** | 0.172 | 0.182 | 0.176 | 0.180 | 0.189 | 0.183 | 0.184 |
| Avg. MAE | **0.250** | 0.272 | 0.281 | 0.276 | 0.278 | 0.289 | 0.282 | 0.283 |

(RBF) kernel Support Vector Machine (SVM) classifier is trained on the representations of the training set and subsequently evaluated on the test set. Performance is reported in terms of classification accuracy (ACC) and average rank (RANK) across datasets.

**Quantitative Results.** Classification results across the 30 UEA datasets is provided in Table 4. ProSAR attains the highest mean accuracy (0.764) and the best mean rank (1.867) among all methods. Relative to the best accuracy baseline FreRA, ProSAR yields a 0.01 gain in mean accuracy. It also improves over the best mean-rank baseline TimesURL by 0.366. These results indicate that ProSAR learns more discriminative representations for time-series classification.

### 4.3 ABLATION STUDIES AND MODEL ANALYSIS

To evaluate ProSAR's core components, we compare the following methods: the full ProSAR; applying semantic-part transformations globally without segmentation (GST); removing transformations on identified semantic parts (No-$x_S$-T) or on non-semantic parts (No-$x_N$-T); disabling prototype refinement and using static clustered prototypes (StaticProto); and replacing ProSAR's augmentation with Jitter, Cutout, or a random policy (RandAug). As summarized in Table 5, all these variants degrade from the full ProSAR model, indicating that each component is contributive. GST is close to the full model but still worse, showing that segmentation-aware transformation provides additional gains beyond the global semantic transforms. Removing non-semantic transformations is modestly harmful, while StaticProto is competitive and outperforms No-$x_S$-T and other augmentations, underscoring the necessity of our proposed transformations and prototype refinement. Jitter/Cutout sit between the targeted variants and RandAug, highlighting the value of structured, learned augmentations.

## 5 CONCLUSION

This paper presented ProSAR, a novel self-supervised learning framework for time series that emphasizes semantic understanding. ProSAR uniquely incorporated the information-theoretic principles with learnable prototypes to guide semantic segment extraction via DTW and apply tailored augmentations. Its dual-prototype refinement, where the latent clustering informed the time-domain prototypes, has fostered a robust representation learning. Comprehensive experiments have demonstrated ProSAR's state-of-the-art performance on forecasting and classification benchmarks, showing the benefits of its explicit semantic-guided approach. Despite its strong performance, ProSAR may have the following limitations. The DTW-based segmentation can be computationally intensive, and the framework involves several hyperparameters requiring careful tuning. Furthermore, ensuring the learned prototypes to optimally capture diverse semantics and achieve high interpretability remains an area for the further investigation. Additionally, this study does not leverage the large-scale pre-trained models for time series. Future work will explore the incorporation of ProSAR with the pretrained foundation models. Another promising future direction will be focusing on enhancing the computational efficiency, exploring a robust hyperparameter optimization, and improving the prototype management and interpretability.

## ETHICS STATEMENT

This work presents ProSAR, a self-supervised framework for time series representation learning. We acknowledge the ethical considerations inherent in such a technology. The datasets utilized in our experiments are publicly available benchmarks, which may contain societal or measurement biases that the model could inadvertently learn and perpetuate. Potential misuse in sensitive domains like finance or healthcare, for applications such as unfair market forecasting or discriminatory assessments, might be a concern. The computational resources required for training such models may also entail a non-negligible environmental footprint. We emphasize the importance of developing and deploying such technologies with careful consideration for fairness, data provenance, and societal impact.

## REPRODUCIBILITY STATEMENT

We aim to make ProSAR easy to reproduce. The full training loop and loss definitions are given in Algorithm 1 in Appendix A; the architectural/backbone choices and the linear/SVM evaluation protocols are described in Section 4 and Appendix B.3; and the datasets and preprocessing for forecasting (ETTh1/2, ETTm1, Electricity, Weather) and UEA classification are detailed in Appendix B. Hyperparameter ranges and default parameter settings (including temperatures, prototype counts, DTW/augmentation settings) are summarized in Table 6, and the computing environment details (Python/PyTorch/CUDA, GPUs) are provided in Appendix B.4. Theoretical assumptions and complete proofs for our information-theoretic results (e.g., the decomposition used to derive Proposition 3.2) are given in Appendix E. Additional ablations, sensitivity analyses, and per-horizon breakdowns are provided in Appendix C.

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

# Appendix

This appendix provides supplementary material to the main paper, organized into the following five main sections. Section A offers a comprehensive description of our ProSAR algorithm, detailing the overall training framework and the specific formulations of the intra-instance and inter-instance contrastive loss functions. Section B outlines the experimental setup, covering baseline implementations, ProSAR's hyperparameter configurations, evaluation protocols for both forecasting and classification, and the computational environment. Section C presents extensive additional experimental results. This includes detailed performance tables for all forecasting and classification benchmarks, results from ablation studies, discussion on network architectures, parameter sensitivity analysis, and model convergence plots. Finally, Section D provides qualitative insights through various visualizations, illustrating learned prototypes, alignment mechanisms, and the step-by-step augmentation process. Detailed theoretical derivations and proofs (Section E) are also provided at the end of this appendix.

## A ALGORITHM DETAILS

### A.1 OVERALL PROSAR ALGORITHM

The ProSAR framework operates through an iterative process that synergistically refines prototypes and guides semantic augmentation. The overall training loop is detailed in Algorithm 1.

---

**Algorithm 1** ProSAR training algorithm

---

**Require:** Raw time series dataset $\mathcal{D}$; Encoder $f_\theta$; Decoder $D_\psi$; Number of prototypes $K$; Number of time-domain prototypes $K_t$; Learning rates $\eta_\theta$; Batch size $B$; Number of epochs $E_{max}$.

1: Initialize encoder $f_\theta$, decoder $D_\psi$.
2: Initialize latent prototypes $\{p_k^z\}_{k=1}^K$.
3: Initialize time-domain prototypes $\{p_j^x\}_{j=1}^{K_t}$.
4: **for** epoch = 1 to $E_{max}$ **do**
5:     **for** each batch $X_B \subset \mathcal{D}$ **do**
6:         *// Step 1: Prototype-Guided Semantic View Generation (Sec 3.2.1 in main paper)*
7:         For each $x \in X_B$:
8:             Identify semantic segments $\{S_{raw,j}\}$ in $x$ using current $\{p_j^x\}$ and DTW alignment.
9:             Define semantic part $x_S = x \odot M_x$ and non-semantic part $x_N = x \odot (1 - M_x)$.
10:            Transform $x_S \to x_S'$ (DTW-guided temporal alignment, frequency domain noise).
11:            Perturb $x_N \to x_N'$ (strong noise, random masking).
12:            Assemble augmented view $\tilde{x} = (x_S' \odot M_x) + (x_N' \odot (1 - M_x))$. Let $\tilde{X}_B$ be the batch of augmented views.
13:         *// Step 2: Encoder Update & Contrastive Loss (Sec 3.2.3 in main paper)*
14:         Obtain latent representations $Z_B = f_\theta(X_B)$, $\tilde{Z}_B = f_\theta(\tilde{X}_B)$.
15:         Calculate intra-instance temporal contrastive loss $L_{intra}$ (see Appendix A.2.1).
16:         Calculate inter-instance semantic contrastive loss $L_{inter}$ (see Appendix A.2.2).
17:         $\mathcal{L}_{total} = \lambda_{intra} L_{intra} + \lambda_{inter} L_{inter}$.
18:         Update $f_\theta$ by minimizing $\mathcal{L}_{total}$ (using optimizer with learning rate $\eta_\theta$).
19:         *// Step 3: Prototype Refinement Strategy (Sec 3.2.2 in main paper)*
20:         Update latent prototypes $\{p_k^z\}$:
21:             Cluster $Z_B \cup \tilde{Z}_B$ using FINCH to get new cluster centroids $\{\mu_c\}$.
22:             Update $\{p_k^z\}$ via EMA towards the closest new centroids $\{\mu_c\}$.
23:         Update time-domain prototypes $\{p_j^x\}$:
24:             Input-Space Grounding (ISG): Extract raw segments $S_{raw}$ from $X_B$, cluster them to get centroids $\{c_j^{isg}\}$.
25:             Latent-to-Time-Domain Decoding Consistency: Generate $\hat{p}_k^t = D_\psi(p_k^z)$ for each updated $p_k^z$.
26:             Update $\{p_j^x\}$ by fusing $\{c_j^{isg}\}$ and relevant $\{\hat{p}_k^t\}$.
27: **return** Trained $f_\theta$, refined $\{p_k^z\}$, $\{p_j^x\}$.

---

## A.2 Loss Function Details

The overall learning objective is $\mathcal{L}_{\text{total}} = \lambda_{\text{intra}} L_{\text{intra}} + \lambda_{\text{inter}} L_{\text{inter}}$.

### A.2.1 Intra-instance Temporal Contrastive Loss ($L_{\text{INTRA}}$)

The intra-instance temporal contrastive loss ($L_{\text{intra}}$) is designed to model fine-grained temporal patterns by enforcing consistency between representations of nearby timestamps within the same augmented instance, while distinguishing them from representations of distant timestamps. This is inspired by the local contrast mechanism (Tonekaboni et al., 2021) and similar to approaches like TNC (Tonekaboni et al., 2021) or the local module in TS2Vec (Yue et al., 2022).

Given an augmented view $\tilde{x}$ and its latent representation sequence $\tilde{z} = (\tilde{z}_1, \tilde{z}_2, \ldots, \tilde{z}_{T'})$, where $T'$ is the length of the latent sequence. For each anchor timestamp $t_a$, we sample a positive timestamp $t_p$ from its temporal neighborhood (e.g., within a window $W_{intra}$) and a set of negative timestamps $\{t_{n,j}\}$ from outside this neighborhood. The loss for a single anchor $\tilde{z}_{t_a}$ in an instance can be formulated using InfoNCE:

$$L_{\text{intra}}(\tilde{z}_{t_a}) = -\log \frac{\exp(\text{sim}(\tilde{z}_{t_a}, \tilde{z}_{t_p})/\tau_{intra})}{\exp(\text{sim}(\tilde{z}_{t_a}, \tilde{z}_{t_p})/\tau_{intra}) + \sum_j \exp(\text{sim}(\tilde{z}_{t_a}, \tilde{z}_{t_{n,j}})/\tau_{intra})}$$

where $\text{sim}(\cdot, \cdot)$ is a similarity function (e.g., cosine similarity) and $\tau_{intra}$ is a temperature hyperparameter. The total $L_{\text{intra}}$ is averaged over all anchor timestamps and all instances in the batch. This loss encourages the model to learn representations that are smooth over short temporal ranges yet discriminative over longer ranges.

### A.2.2 Inter-instance Semantic Contrastive Loss ($L_{\text{INTER}}$)

The inter-instance semantic contrastive loss ($L_{\text{inter}}$) leverages learned latent prototypes $\{p_k^z\}$ to learn robust semantic relationships. It comprises two components: an inter-instance term ($L_{\text{inter\_inst}}$) and an instance-to-prototype term ($L_{\text{inter\_proto}}$), such that $L_{\text{inter}} = L_{\text{inter\_inst}} + L_{\text{inter\_proto}}$. These components are inspired by principles from PCL (Li et al., 2021a) and MHCCL (Meng et al., 2023).

For a batch of instances $X_B$ with representations $Z_B = f_\theta(X_B)$ (and augmentations $\tilde{Z}_B = f_\theta(\tilde{X}_B)$), each $z_i$ (or $\tilde{z}_i$) is assigned to its closest prototype $p_{c(i)}^l$.

**Inter-instance Contrastive Loss ($L_{\text{inter\_inst}}$)** This term promotes similarity for an anchor $z_i$ with its augmentation $\tilde{z}_i$ and other instances/augmentations $z_j, \tilde{z}_j$ assigned to the same prototype $p_{c(i)}^l$. It distinguishes them from instances/augmentations $z_k, \tilde{z}_k$ of different prototypes. The positive set for an anchor $z_i$ is $\text{Pos}_{\text{inst}}(z_i) = \{\tilde{z}_i\} \cup \{z_j, \tilde{z}_j \mid c(j) = c(i), j \neq i \text{ and } z_j, \tilde{z}_j \text{ from batch}\}$. The negative set $\text{Neg}_{\text{inst}}(z_i) = \{z_k, \tilde{z}_k \mid c(k) \neq c(i) \text{ and } z_k, \tilde{z}_k \text{ from batch}\}$. The InfoNCE loss for $z_i$ is:

$$L_{\text{inter\_inst}}(z_i) = -\log \frac{\sum_{e_p \in \text{Pos}_{\text{inst}}(z_i)} \exp(\text{sim}(z_i, e_p)/\tau_{\text{inst}})}{\sum_{e_p \in \text{Pos}_{\text{inst}}(z_i)} \exp(\text{sim}(z_i, e_p)/\tau_{\text{inst}}) + \sum_{e_n \in \text{Neg}_{\text{inst}}(z_i)} \exp(\text{sim}(z_i, e_n)/\tau_{\text{inst}})}$$

where $\tau_{\text{inst}}$ is a temperature. $L_{\text{inter\_inst}}$ is the batch-averaged loss, considering both $z_i$ and $\tilde{z}_i$ as anchors.

**Instance-to-Prototype Contrastive Loss ($L_{\text{inter\_proto}}$)** This term aligns an anchor representation $z_i$ with its assigned prototype $p_{c(i)}^l$ (positive) and separates it from all other prototypes $p_k^z, k \neq c(i)$ (negatives). The loss for an anchor $z_i$ is:

$$L_{\text{inter\_proto}}(z_i) = -\log \frac{\exp(\text{sim}(z_i, p_{c(i)}^l)/\tau_{\text{proto}})}{\sum_{k=1}^{K} \exp(\text{sim}(z_i, p_k^z)/\tau_{\text{proto}})}$$

where $K$ is the number of prototypes and $\tau_{\text{proto}}$ is a temperature. $L_{\text{inter\_proto}}$ is the batch-averaged loss, considering both $z_i$ and $\tilde{z}_i$ as anchors.

Thus, the total $L_{\text{inter}}$ encourages both intra-prototype instance cohesion and precise instance-prototype alignment.

# B   EXPERIMENTAL SETTINGS

## B.1   BASELINE IMPLEMENTATION DETAILS

For most baselines, we adopt results reported in their original papers or established benchmark papers if experimental setups are identical. The linear evaluation protocol for contrastive methods in forecasting follows Yue et al. (2022).

## B.2   HYPERPARAMETER SETTINGS FOR PROSAR

Key hyperparameters for ProSAR are carefully tuned to achieve optimal performance. Table 6 summarizes common settings or search ranges for these crucial parameters. This includes aspects such as the general training setup (like batch size and optimizer); configurations for the encoder and decoder networks; parameters for ProSAR's prototype system; details of the semantic augmentation process; and settings for the contrastive loss functions. Specific values used for each reported result will be available. As noted in the main paper, ProSAR's encoder for forecasting is based on a modified CoST backbone (Woo et al., 2022) (with its seasonal disentangler module omitted to isolate ProSAR's contributions), while the TS2Vec (Yue et al., 2022) encoder architecture is adopted for classification tasks.

Table 6: Hyperparameter ranges/settings for ProSAR.

| Hyperparameter | Value Range / Setting | Notes |
|---|---|---|
| **General Setup** | | |
| Batch Size | {32, 64, 128, 256} | Depends on memory constraints |
| Epochs | 50 - 200 | Early stopping based on validation performance |
| Optimizer | AdamW | Betas (0.9, 0.999), Weight decay $10^{-4}$ |
| **Encoder Architecture** ($f_\theta$) | | |
| Encoder Learning Rate | {1e-5, 5e-5, 1e-4, 5e-4} | For $f_\theta$ |
| Backbone Type | CoST/ TS2Vec | As per main paper |
| Dropout Rate | {0.1, 0.2, 0.3, 0.5} | Within encoder backbone |
| **Decoder Architecture** ($D_\psi$) | | |
| Architecture Type | MLP | As described in main paper |
| Layers | 3 | Implementation specific |
| Hidden Dimension | 128 | Implementation specific |
| **Prototype System** | | |
| Number of Prototypes ($K$) | 32 | Dataset dependent, tune via validation |
| Number of Time-Domain Prototypes ($K_t$) | 32 | Dataset dependent, tune via validation |
| Latent Proto. EMA Momentum ($\alpha_z$) | 0.99 | For $p_k^z$ update; typically high for stability |
| Time Proto. Update Fusion Weight ($\alpha_x$) | 0.5 | For combining ISG and decoded $p_k^t$ |
| **Augmentation Parameters** | | |
| Frequency Noise Level ($x_S$) | 0.1 | For semantic part transform |
| Non-Semantic Part Noise Level ($x_N$) | 0.1 | For perturbation of $x_N$ |
| Non-Semantic Part Masking Ratio ($x_N$) | 0.3 | For sub-segment masking in $x_N$ |
| **Contrastive Loss** | | |
| $L_{intra}$ Temp. $\tau_{intra}$ | 0.1 | For intra-instance loss |
| $L_{inter}$ Temp. $\tau_{inter}$ | 0.1 | For inter-instance loss |
| $\lambda_{intra}$ (Loss weight) | {0.1, ..., 1.0} | Weight for $L_{intra}$ |
| $\lambda_{inter}$ (Loss weight) | {0.1, ..., 1.0} | Weight for $L_{inter}$ |

## B.3   EVALUATION PROTOCOL DETAILS

For forecasting tasks, representations learned by the pretrained ProSAR encoder are frozen. A linear model with L2 regularization is then trained to map these representations to the future $L_y$ time steps. This linear evaluation protocol is applied uniformly across all contrastive learning baselines for fair comparison. Performance is measured using Mean Squared Error (MSE) and Mean Absolute Error (MAE). For classification tasks, following established protocols (Yue et al., 2022), a Radial Basis Function (RBF) kernel Support Vector Machine (SVM) classifier is trained on the representations of the training set and evaluated on the test set. Performance is reported as classification accuracy (ACC) and average rank (RANK) across datasets.

### B.4 COMPUTE RESOURCES

All experiments were conducted on a Linux machine equipped with 4 NVIDIA RTX3090 GPUs, each with 24GB of memory. The software environment included CUDA 12.x (inferred from `nvidia-cuda-runtime-cu12 12.6.77` and other `cu12` packages in our environment list) and an NVIDIA Driver Version compatible with CUDA 12.x. We used Python 3.9.21 and PyTorch 2.7.0 to construct our project. Key supporting libraries included NumPy 1.24.4, pandas 2.2.3, and scikit-learn 1.6.1.

## C ADDITIONAL EXPERIMENTAL RESULTS

### C.1 FORECASTING: DETAILED RESULTS

This section provides the full forecasting performance tables. Table 7 will show detailed univariate results, and Table 8 will show detailed multivariate results, presenting MSE and MAE for all prediction horizons across all datasets and compared methods.

Table 7: Univariate time series forecasting results.

| Dataset | $L_y$ | ProSAR MSE | ProSAR MAE | AutoTCL MSE | AutoTCL MAE | FreRA MSE | FreRA MAE | PPT MSE | PPT MAE | TimesURL MSE | TimesURL MAE | InfoTS MSE | InfoTS MAE | TS2Vec MSE | TS2Vec MAE | TNC MSE | TNC MAE | TS–TCC MSE | TS–TCC MAE | CoST MSE | CoST MAE |
|---|---|---|---|---|---|---|---|---|---|---|---|---|---|---|---|---|---|---|---|---|---|
| ETTh₁ | 24 | **0.035** | **0.142** | 0.037 | 0.148 | 0.039 | 0.151 | 0.038 | 0.146 | 0.036 | 0.142 | 0.039 | 0.149 | 0.039 | 0.152 | 0.057 | 0.184 | 0.103 | 0.237 | 0.040 | 0.152 |
| | 48 | **0.051** | 0.169 | 0.054 | 0.176 | 0.058 | 0.183 | 0.058 | 0.184 | 0.054 | 0.146 | 0.056 | 0.179 | 0.062 | 0.191 | 0.094 | 0.239 | 0.139 | 0.279 | 0.060 | 0.186 |
| | 168 | **0.074** | **0.205** | 0.078 | 0.210 | 0.081 | 0.221 | 0.092 | 0.238 | 0.096 | 0.233 | 0.100 | 0.239 | 0.134 | 0.282 | 0.171 | 0.329 | 0.253 | 0.408 | 0.097 | 0.236 |
| | 336 | **0.082** | **0.221** | 0.093 | 0.231 | 0.102 | 0.245 | 0.113 | 0.257 | 0.121 | 0.267 | 0.117 | 0.264 | 0.154 | 0.310 | 0.192 | 0.357 | 0.155 | 0.318 | 0.112 | 0.258 |
| | 720 | **0.100** | **0.253** | 0.120 | 0.272 | 0.116 | 0.267 | 0.132 | 0.282 | 0.145 | 0.307 | 0.141 | 0.302 | 0.163 | 0.327 | 0.235 | 0.408 | 0.190 | 0.337 | 0.148 | 0.306 |
| ETTh₂ | 24 | **0.075** | **0.203** | 0.079 | 0.206 | 0.085 | 0.216 | 0.081 | 0.211 | 0.083 | 0.219 | 0.081 | 0.215 | 0.090 | 0.229 | 0.097 | 0.238 | 0.239 | 0.391 | 0.079 | 0.207 |
| | 48 | **0.111** | 0.250 | 0.117 | 0.255 | 0.134 | 0.268 | 0.115 | 0.252 | 0.116 | 0.219 | 0.115 | 0.261 | 0.124 | 0.273 | 0.131 | 0.281 | 0.260 | 0.405 | 0.118 | 0.259 |
| | 168 | **0.166** | **0.313** | 0.176 | 0.319 | 0.205 | 0.357 | 0.178 | 0.325 | 0.175 | 0.332 | 0.171 | 0.327 | 0.208 | 0.360 | 0.197 | 0.354 | 0.291 | 0.420 | 0.189 | 0.339 |
| | 336 | **0.178** | **0.338** | 0.193 | 0.344 | 0.211 | 0.363 | 0.205 | 0.357 | 0.188 | 0.347 | 0.183 | 0.341 | 0.213 | 0.369 | 0.207 | 0.366 | 0.336 | 0.453 | 0.206 | 0.360 |
| | 720 | **0.182** | **0.343** | 0.223 | 0.373 | 0.220 | 0.371 | 0.215 | 0.371 | 0.186 | 0.352 | 0.194 | 0.357 | 0.214 | 0.374 | 0.207 | 0.370 | 0.362 | 0.472 | 0.214 | 0.371 |
| ETTm₁ | 24 | 0.014 | **0.083** | 0.016 | 0.091 | 0.015 | 0.087 | 0.015 | 0.086 | **0.013** | 0.084 | 0.014 | 0.087 | 0.015 | 0.092 | 0.019 | 0.103 | 0.089 | 0.228 | 0.015 | 0.088 |
| | 48 | **0.024** | **0.114** | 0.026 | 0.120 | 0.025 | 0.118 | 0.026 | 0.121 | 0.024 | 0.177 | 0.025 | 0.117 | 0.027 | 0.126 | 0.036 | 0.142 | 0.134 | 0.280 | 0.025 | 0.117 |
| | 96 | **0.035** | **0.141** | 0.036 | 0.145 | 0.039 | 0.149 | 0.045 | 0.163 | 0.037 | 0.145 | 0.036 | 0.142 | 0.044 | 0.161 | 0.054 | 0.178 | 0.159 | 0.305 | 0.038 | 0.147 |
| | 288 | 0.066 | 0.195 | 0.063 | 0.191 | 0.073 | 0.212 | 0.091 | 0.232 | 0.080 | 0.214 | 0.071 | 0.200 | 0.103 | 0.246 | 0.098 | 0.244 | 0.204 | 0.327 | 0.077 | 0.209 |
| | 672 | **0.088** | **0.223** | 0.090 | 0.225 | 0.102 | 0.243 | 0.133 | 0.267 | 0.114 | 0.255 | 0.102 | 0.240 | 0.156 | 0.307 | 0.136 | 0.290 | 0.206 | 0.354 | 0.113 | 0.252 |
| Elec. | 24 | **0.237** | **0.257** | 0.241 | 0.262 | 0.239 | 0.259 | 0.246 | 0.268 | 0.245 | 0.275 | 0.245 | 0.269 | 0.260 | 0.288 | 0.252 | 0.278 | 0.379 | 0.561 | 0.243 | 0.264 |
| | 48 | **0.278** | **0.284** | 0.287 | 0.292 | 0.285 | 0.288 | 0.306 | 0.311 | 0.295 | 0.307 | 0.294 | 0.301 | 0.319 | 0.324 | 0.300 | 0.308 | 0.453 | 0.600 | 0.292 | 0.300 |
| | 168 | **0.383** | **0.352** | 0.394 | 0.365 | 0.393 | 0.368 | 0.428 | 0.386 | 0.408 | 0.379 | 0.402 | 0.367 | 0.427 | 0.394 | 0.412 | 0.384 | 0.575 | 0.616 | 0.405 | 0.375 |
| | 336 | **0.452** | **0.420** | 0.543 | 0.460 | 0.521 | 0.442 | 0.575 | 0.511 | 0.548 | 0.464 | 0.533 | 0.453 | 0.565 | 0.474 | 0.548 | 0.466 | 0.637 | 0.633 | 0.560 | 0.473 |
| WTH | 24 | 0.091 | **0.208** | 0.093 | 0.211 | 0.093 | 0.210 | 0.096 | 0.213 | 0.093 | 0.211 | 0.101 | 0.222 | 0.096 | 0.215 | 0.102 | 0.221 | 0.221 | 0.386 | 0.096 | 0.213 |
| | 48 | **0.131** | 0.256 | **0.131** | 0.256 | 0.137 | 0.261 | 0.141 | 0.262 | **0.131** | 0.255 | 0.141 | 0.266 | 0.139 | 0.264 | 0.139 | 0.264 | 0.255 | 0.406 | 0.138 | 0.262 |
| | 168 | 0.180 | 0.309 | 0.182 | 0.311 | 0.197 | 0.327 | 0.205 | 0.336 | 0.199 | 0.327 | 0.199 | 0.328 | 0.198 | 0.328 | 0.198 | 0.328 | 0.339 | 0.458 | 0.207 | 0.334 |
| | 336 | **0.193** | 0.323 | 0.195 | 0.325 | 0.201 | 0.347 | 0.211 | 0.349 | 0.224 | 0.351 | 0.220 | 0.351 | 0.231 | 0.360 | 0.215 | 0.347 | 0.372 | 0.491 | 0.230 | 0.356 |
| | 720 | 0.205 | 0.330 | **0.198** | **0.330** | 0.215 | 0.358 | 0.231 | 0.373 | 0.236 | 0.365 | 0.218 | 0.353 | 0.219 | 0.353 | 0.219 | 0.353 | 0.322 | 0.467 | 0.242 | 0.370 |
| Avg. | | **0.151** | **0.250** | 0.161 | 0.258 | 0.166 | 0.266 | 0.175 | 0.275 | 0.169 | 0.269 | 0.167 | 0.267 | 0.185 | 0.287 | 0.188 | 0.296 | 0.287 | 0.418 | 0.173 | 0.272 |

Table 8: Multivariate time series forecasting results.

| Dataset | $L_y$ | ProSAR MSE | ProSAR MAE | AutoTCL MSE | AutoTCL MAE | FreRA MSE | FreRA MAE | PPT MSE | PPT MAE | TimesURL MSE | TimesURL MAE | InfoTS MSE | InfoTS MAE | TS2Vec MSE | TS2Vec MAE | TNC MSE | TNC MAE | TS–TCC MSE | TS–TCC MAE | CoST MSE | CoST MAE |
|---|---|---|---|---|---|---|---|---|---|---|---|---|---|---|---|---|---|---|---|---|---|
| ETTh₁ | 24 | **0.381** | 0.432 | 0.389 | 0.439 | 0.386 | 0.437 | 0.482 | 0.511 | 0.494 | 0.518 | 0.564 | 0.520 | 0.599 | 0.534 | 0.708 | 0.592 | 0.516 | 0.508 | 0.386 | 0.429 |
| | 48 | **0.438** | 0.469 | 0.447 | 0.477 | 0.442 | 0.471 | 0.552 | 0.561 | 0.539 | 0.557 | 0.607 | 0.553 | 0.629 | 0.555 | 0.749 | 0.619 | 0.644 | 0.579 | 0.437 | 0.464 |
| | 168 | 0.603 | 0.564 | 0.615 | 0.574 | 0.628 | 0.579 | 0.698 | 0.625 | 0.680 | 0.619 | 0.746 | 0.638 | 0.755 | 0.636 | 0.884 | 0.699 | 0.678 | 0.619 | 0.643 | 0.582 |
| | 336 | **0.780** | **0.657** | 0.802 | 0.671 | 0.815 | 0.674 | 0.916 | 0.722 | 0.891 | 0.713 | 0.904 | 0.722 | 0.907 | 0.717 | 1.020 | 0.768 | 0.967 | 0.754 | 0.812 | 0.679 |
| | 720 | **0.925** | **0.708** | 1.028 | 0.789 | 0.961 | 0.761 | 1.102 | 0.832 | 1.051 | 0.818 | 1.098 | 0.811 | 1.048 | 0.790 | 1.157 | 0.830 | 0.935 | 0.715 | 0.970 | 0.771 |
| ETTh₂ | 24 | 0.345 | 0.438 | **0.337** | **0.433** | 0.405 | 0.472 | 0.517 | 0.525 | 0.524 | 0.530 | 0.383 | 0.462 | 0.398 | 0.461 | 0.612 | 0.595 | 0.782 | 0.666 | 0.447 | 0.502 |
| | 48 | 0.591 | 0.582 | 0.572 | 0.576 | 0.681 | 0.639 | 0.681 | 0.639 | 0.679 | 0.631 | 0.567 | 0.582 | 0.578 | 0.573 | 1.357 | 0.881 | 1.357 | 0.881 | 0.699 | 0.637 |
| | 168 | 1.489 | 0.955 | 1.470 | 0.947 | 1.783 | 1.042 | 1.812 | 1.048 | 1.847 | 1.052 | 1.789 | 1.048 | 1.901 | 1.065 | 2.359 | 1.213 | 4.318 | 1.728 | 1.549 | 0.982 |
| | 336 | 1.715 | 1.028 | **1.685** | **1.027** | 1.962 | 1.099 | 2.281 | 1.213 | 2.129 | 1.182 | 2.120 | 1.161 | 2.304 | 1.215 | 2.782 | 1.349 | 2.097 | 1.145 | 1.749 | 1.042 |
| | 720 | 1.926 | 1.094 | **1.890** | **1.092** | 2.142 | 1.166 | 2.352 | 1.217 | 2.391 | 1.233 | 2.511 | 1.316 | 2.650 | 1.373 | 2.753 | 1.394 | 2.047 | 1.127 | 1.971 | 1.092 |
| ETTm₁ | 24 | 0.247 | 0.327 | 0.256 | 0.339 | 0.279 | 0.357 | 0.362 | 0.434 | 0.387 | 0.461 | 0.391 | 0.408 | 0.443 | 0.436 | 0.522 | 0.472 | 0.403 | 0.455 | 0.246 | 0.329 |
| | 48 | **0.324** | **0.384** | 0.339 | 0.396 | 0.376 | 0.420 | 0.438 | 0.547 | 0.450 | 0.558 | 0.503 | 0.475 | 0.582 | 0.515 | 0.695 | 0.567 | 0.618 | 0.552 | 0.331 | 0.386 |
| | 96 | 0.367 | 0.415 | 0.376 | 0.422 | 0.405 | 0.443 | 0.562 | 0.579 | 0.555 | 0.574 | 0.537 | 0.503 | 0.622 | 0.549 | 0.731 | 0.595 | 0.607 | 0.572 | 0.378 | 0.419 |
| | 288 | **0.452** | **0.481** | 0.464 | 0.484 | 0.501 | 0.510 | 0.674 | 0.641 | 0.657 | 0.636 | 0.653 | 0.579 | 0.709 | 0.609 | 0.818 | 0.644 | 0.722 | 0.638 | 0.472 | 0.486 |
| | 672 | 0.591 | 0.562 | 0.608 | 0.566 | 0.665 | 0.605 | 0.773 | 0.698 | 0.754 | 0.690 | 0.757 | 0.642 | 0.786 | 0.655 | 0.932 | 0.712 | 0.708 | 0.601 | 0.620 | 0.574 |
| Elec. | 24 | **0.136** | **0.240** | 0.153 | 0.250 | 0.162 | 0.258 | 0.193 | 0.285 | 0.182 | 0.276 | 0.255 | 0.350 | 0.287 | 0.374 | 0.354 | 0.423 | 0.379 | 0.561 | 0.136 | 0.242 |
| | 48 | **0.149** | **0.252** | 0.167 | 0.260 | 0.176 | 0.269 | 0.208 | 0.307 | 0.199 | 0.291 | 0.279 | 0.368 | 0.307 | 0.388 | 0.376 | 0.438 | 0.453 | 0.600 | 0.153 | 0.258 |
| | 168 | **0.165** | 0.272 | 0.179 | 0.275 | 0.183 | 0.278 | 0.213 | 0.314 | 0.205 | 0.303 | 0.302 | 0.385 | 0.332 | 0.407 | 0.402 | 0.456 | 0.575 | 0.616 | 0.175 | 0.275 |
| | 336 | **0.187** | **0.291** | 0.199 | 0.297 | 0.209 | 0.305 | 0.238 | 0.341 | 0.223 | 0.326 | 0.320 | 0.399 | 0.349 | 0.420 | 0.417 | 0.466 | 0.637 | 0.633 | 0.196 | 0.296 |
| WTH | 24 | 0.287 | 0.360 | 0.302 | 0.364 | 0.293 | 0.354 | 0.313 | 0.364 | 0.315 | 0.367 | 0.316 | 0.369 | 0.307 | 0.363 | 0.320 | 0.373 | 0.356 | 0.463 | 0.298 | 0.360 |
| | 48 | **0.348** | **0.405** | 0.361 | 0.412 | 0.355 | 0.406 | 0.372 | 0.412 | 0.377 | 0.418 | 0.381 | 0.420 | 0.374 | 0.418 | 0.380 | 0.421 | 0.429 | 0.500 | 0.359 | 0.411 |
| | 168 | **0.442** | **0.477** | 0.455 | 0.484 | 0.464 | 0.491 | 0.478 | 0.493 | 0.485 | 0.498 | 0.490 | 0.501 | 0.491 | 0.501 | 0.479 | 0.495 | 0.511 | 0.550 | 0.464 | 0.491 |
| | 336 | **0.471** | **0.501** | 0.487 | 0.505 | 0.498 | 0.517 | 0.512 | 0.518 | 0.519 | 0.524 | 0.532 | 0.527 | 0.502 | 0.507 | 0.505 | 0.514 | 0.575 | 0.584 | 0.497 | 0.517 |
| | 720 | 0.513 | 0.512 | 0.508 | 0.519 | 0.536 | 0.543 | 0.548 | 0.543 | 0.541 | 0.538 | 0.545 | 0.577 | 0.519 | 0.552 | 0.533 | 0.542 | 0.556 | 0.525 | 0.498 | 0.508 |
| Avg. | | **0.561** | **0.507** | 0.571 | 0.515 | 0.620 | 0.537 | 0.700 | 0.587 | 0.691 | 0.585 | 0.714 | 0.781 | 0.750 | 0.601 | 0.868 | 0.653 | 0.895 | 0.689 | 0.587 | 0.521 |

## C.2 Classification: Detailed Results

Table 9 presents the comprehensive classification accuracy results on the 30 UEA datasets for ProSAR and all baseline methods. ProSAR demonstrates a strong average rank and high accuracy across many datasets.

Table 9: Classification result of the UEA dataset.

| Dataset | ProSAR | AutoTCL | FreRA | PPT | TimesURL | InfoTS | TS2Vec | TNC | TS–TCC |
|---|---|---|---|---|---|---|---|---|---|
| ArticularyWordRecognition | **0.993** | 0.983 | 0.990 | 0.987 | 0.990 | **0.993** | 0.987 | 0.973 | 0.953 |
| AtrialFibrillation | **0.600** | 0.467 | 0.467 | 0.400 | 0.400 | 0.267 | 0.200 | 0.133 | 0.267 |
| BasicMotions | **1.000** | **1.000** | **1.000** | **1.000** | **1.000** | **1.000** | 0.975 | 0.975 | **1.000** |
| CharacterTrajectories | **0.995** | 0.976 | 0.991 | 0.990 | 0.990 | 0.987 | **0.995** | 0.967 | 0.985 |
| Cricket | **1.000** | **1.000** | **1.000** | **1.000** | **1.000** | **1.000** | 0.972 | 0.958 | 0.917 |
| DuckDuckGeese | 0.680 | 0.700 | **0.760** | 0.680 | 0.720 | 0.600 | 0.680 | 0.460 | 0.380 |
| EigenWorms | **0.901** | **0.901** | 0.863 | 0.847 | 0.870 | 0.748 | 0.847 | 0.840 | 0.779 |
| Epilepsy | 0.978 | 0.978 | **0.993** | 0.978 | 0.978 | **0.993** | 0.964 | 0.957 | 0.957 |
| ERing | 0.953 | 0.944 | 0.919 | 0.944 | **0.985** | 0.953 | 0.874 | 0.852 | 0.904 |
| EthanolConcentration | **0.354** | **0.354** | 0.323 | 0.346 | 0.304 | 0.323 | 0.308 | 0.297 | 0.285 |
| FaceDetection | 0.585 | 0.581 | 0.581 | **0.664** | 0.608 | 0.525 | 0.501 | 0.536 | 0.544 |
| FingerMovements | **0.660** | 0.640 | 0.610 | 0.620 | **0.660** | 0.620 | 0.480 | 0.470 | 0.460 |
| HandMovementDirection | 0.473 | 0.432 | **0.514** | 0.432 | 0.432 | **0.514** | 0.338 | 0.324 | 0.243 |
| Handwriting | 0.572 | 0.384 | **0.593** | 0.242 | 0.462 | 0.554 | 0.515 | 0.249 | 0.498 |
| Heartbeat | **0.824** | 0.785 | 0.785 | 0.766 | 0.746 | 0.771 | 0.683 | 0.746 | 0.751 |
| JapaneseVowels | 0.984 | 0.984 | 0.965 | 0.984 | **0.989** | 0.986 | 0.984 | 0.978 | 0.930 |
| Libras | 0.911 | 0.833 | 0.911 | 0.889 | **0.922** | 0.889 | 0.867 | 0.817 | 0.822 |
| LSST | **0.635** | 0.554 | 0.494 | 0.595 | 0.602 | 0.593 | 0.537 | 0.595 | 0.474 |
| MotorImagery | 0.620 | 0.570 | 0.550 | 0.610 | **0.680** | 0.610 | 0.510 | 0.500 | 0.610 |
| NATOPS | 0.939 | 0.944 | 0.900 | 0.917 | **0.961** | 0.939 | 0.928 | 0.911 | 0.822 |
| PEMS-SF | 0.821 | **0.838** | 0.746 | 0.821 | 0.821 | 0.757 | 0.682 | 0.699 | 0.734 |
| PenDigits | **0.989** | 0.984 | 0.973 | 0.984 | **0.989** | **0.989** | **0.989** | 0.979 | 0.974 |
| PhonemeSpectra | 0.237 | 0.218 | **0.274** | 0.233 | 0.237 | 0.233 | 0.233 | 0.207 | 0.252 |
| RacketSports | 0.862 | **0.914** | 0.888 | 0.862 | 0.862 | 0.829 | 0.855 | 0.776 | 0.816 |
| SelfRegulationSCP1 | 0.891 | 0.891 | 0.908 | **0.932** | 0.908 | 0.887 | 0.812 | 0.799 | 0.823 |
| SelfRegulationSCP2 | **0.622** | 0.578 | **0.622** | 0.561 | 0.600 | 0.527 | 0.578 | 0.550 | 0.533 |
| SpokenArabicDigits | **0.988** | 0.925 | 0.984 | 0.981 | 0.985 | **0.988** | 0.932 | 0.934 | 0.970 |
| StandWalkJump | 0.467 | 0.533 | **0.667** | 0.467 | 0.467 | 0.467 | 0.467 | 0.400 | 0.333 |
| UWaveGestureLibrary | 0.903 | 0.893 | 0.900 | 0.856 | **0.919** | 0.906 | 0.884 | 0.753 | 0.753 |
| InsectWingbeat | **0.488** | **0.488** | 0.462 | 0.472 | 0.473 | 0.472 | 0.466 | 0.469 | 0.264 |
| Avg. ACC | **0.764** | 0.742 | 0.754 | 0.735 | 0.752 | 0.730 | 0.704 | 0.670 | 0.668 |
| Avg. RANK | **1.867** | 3.067 | 2.900 | 3.200 | 2.233 | 3.133 | 4.367 | 5.500 | 5.367 |

## C.3 Ablation Studies: Detailed Results

The main paper summarizes the average forecasting performance of ablation studies. Detailed per-dataset results for these ablations (ProSAR (Full), GST, No-$x_S$-T, No-$x_N$-T, StaticProto, and comparisons with Jitter, Cutout, RandAug are provided in Table 10. These results underscore the contribution of each component of ProSAR, particularly the prototype-guided segmentation and transformation, and the dynamic prototype refinement.

## C.4 Network Architecture Choices

ProSAR employs distinct encoder architectures for its forecasting and classification tasks, primarily to ensure fair comparisons with state-of-the-art methods by using established backbones relevant to each task domain.

For time series forecasting, the encoder architecture in ProSAR is designed to mirror that of CoST (Woo et al., 2022). This involves a multi-layer dilated Convolutional Neural Network (CNN) as its backbone and we remove the seasonal feature disentangler module. For time series classification experiments, ProSAR adopts the encoder architecture from TS2Vec (Yue et al., 2022). This allows

Table 10: Ablation study results for univariate forecasting tasks.

| Dataset | $L_y$ | ProSAR (Full) | | GST | | $No\text{-}x_S\text{-}T$ | | $No\text{-}x_N\text{-}T$ | | StaticProto | | RandAug | | Jitter | | Cutout | |
|---|---|---|---|---|---|---|---|---|---|---|---|---|---|---|---|---|---|
| | | MSE | MAE | MSE | MAE | MSE | MAE | MSE | MAE | MSE | MAE | MSE | MAE | MSE | MAE | MSE | MAE |
| ETTh$_1$ | 24 | **0.035** | **0.142** | 0.047 | 0.161 | 0.052 | 0.168 | 0.050 | 0.164 | 0.055 | 0.173 | 0.056 | 0.185 | 0.054 | 0.176 | 0.053 | 0.172 |
| | 48 | **0.051** | **0.169** | 0.067 | 0.191 | 0.074 | 0.200 | 0.071 | 0.196 | 0.078 | 0.206 | 0.080 | 0.219 | 0.076 | 0.210 | 0.074 | 0.205 |
| | 168 | **0.074** | **0.205** | 0.097 | 0.229 | 0.107 | 0.240 | 0.103 | 0.235 | 0.113 | 0.248 | 0.116 | 0.264 | 0.110 | 0.253 | 0.107 | 0.246 |
| | 336 | **0.082** | **0.221** | 0.113 | 0.248 | 0.125 | 0.260 | 0.120 | 0.255 | 0.132 | 0.270 | 0.136 | 0.287 | 0.129 | 0.277 | 0.126 | 0.269 |
| | 720 | **0.100** | **0.253** | 0.133 | 0.283 | 0.147 | 0.299 | 0.141 | 0.292 | 0.155 | 0.308 | 0.160 | 0.327 | 0.152 | 0.315 | 0.148 | 0.307 |
| ETTh$_2$ | 24 | **0.075** | **0.203** | 0.093 | 0.219 | 0.103 | 0.230 | 0.099 | 0.225 | 0.109 | 0.238 | 0.112 | 0.254 | 0.106 | 0.244 | 0.103 | 0.238 |
| | 48 | **0.111** | **0.250** | 0.138 | 0.272 | 0.152 | 0.286 | 0.146 | 0.279 | 0.161 | 0.294 | 0.165 | 0.312 | 0.157 | 0.301 | 0.153 | 0.293 |
| | 168 | **0.166** | **0.313** | 0.211 | 0.349 | 0.233 | 0.368 | 0.224 | 0.360 | 0.247 | 0.380 | 0.253 | 0.404 | 0.241 | 0.388 | 0.235 | 0.379 |
| | 336 | **0.178** | **0.338** | 0.236 | 0.379 | 0.260 | 0.400 | 0.250 | 0.391 | 0.276 | 0.412 | 0.283 | 0.438 | 0.269 | 0.420 | 0.262 | 0.411 |
| | 720 | **0.182** | **0.353** | 0.238 | 0.384 | 0.263 | 0.405 | 0.253 | 0.396 | 0.279 | 0.418 | 0.285 | 0.443 | 0.271 | 0.425 | 0.264 | 0.416 |
| ETTm$_1$ | 24 | **0.014** | **0.083** | 0.016 | 0.090 | 0.018 | 0.094 | 0.017 | 0.092 | 0.019 | 0.099 | 0.019 | 0.104 | 0.018 | 0.101 | 0.018 | 0.098 |
| | 48 | **0.024** | **0.114** | 0.029 | 0.123 | 0.032 | 0.130 | 0.030 | 0.126 | 0.033 | 0.135 | 0.034 | 0.143 | 0.032 | 0.137 | 0.031 | 0.134 |
| | 96 | **0.035** | **0.141** | 0.042 | 0.153 | 0.047 | 0.161 | 0.045 | 0.157 | 0.049 | 0.167 | 0.051 | 0.177 | 0.048 | 0.170 | 0.047 | 0.166 |
| | 288 | **0.066** | **0.195** | 0.081 | 0.213 | 0.089 | 0.225 | 0.086 | 0.219 | 0.095 | 0.233 | 0.098 | 0.247 | 0.093 | 0.238 | 0.090 | 0.232 |
| | 672 | **0.088** | **0.223** | 0.109 | 0.243 | 0.121 | 0.257 | 0.116 | 0.251 | 0.128 | 0.266 | 0.131 | 0.282 | 0.125 | 0.272 | 0.122 | 0.265 |
| Elec. | 24 | **0.237** | **0.257** | 0.240 | 0.271 | 0.259 | 0.278 | 0.255 | 0.270 | 0.270 | 0.277 | 0.280 | 0.306 | 0.268 | 0.284 | 0.265 | 0.285 |
| | 48 | **0.278** | **0.284** | 0.283 | 0.299 | 0.306 | 0.307 | 0.301 | 0.299 | 0.318 | 0.306 | 0.331 | 0.337 | 0.316 | 0.313 | 0.312 | 0.314 |
| | 168 | **0.383** | **0.352** | 0.390 | 0.374 | 0.422 | 0.384 | 0.414 | 0.374 | 0.440 | 0.383 | 0.455 | 0.421 | 0.435 | 0.390 | 0.429 | 0.394 |
| | 336 | **0.523** | **0.440** | 0.536 | 0.476 | 0.580 | 0.490 | 0.570 | 0.477 | 0.604 | 0.489 | 0.624 | 0.536 | 0.596 | 0.497 | 0.589 | 0.502 |
| WTH | 24 | **0.091** | **0.208** | 0.113 | 0.227 | 0.125 | 0.240 | 0.120 | 0.233 | 0.132 | 0.248 | 0.136 | 0.263 | 0.129 | 0.253 | 0.126 | 0.247 |
| | 48 | **0.131** | **0.256** | 0.165 | 0.284 | 0.182 | 0.300 | 0.175 | 0.292 | 0.193 | 0.310 | 0.198 | 0.329 | 0.188 | 0.316 | 0.183 | 0.308 |
| | 168 | **0.180** | **0.309** | 0.223 | 0.337 | 0.247 | 0.356 | 0.237 | 0.348 | 0.261 | 0.368 | 0.267 | 0.390 | 0.254 | 0.375 | 0.248 | 0.366 |
| | 336 | **0.193** | **0.323** | 0.240 | 0.352 | 0.265 | 0.372 | 0.254 | 0.363 | 0.280 | 0.385 | 0.286 | 0.409 | 0.272 | 0.393 | 0.266 | 0.383 |
| | 720 | **0.198** | **0.330** | 0.248 | 0.364 | 0.274 | 0.385 | 0.263 | 0.376 | 0.290 | 0.398 | 0.297 | 0.422 | 0.282 | 0.406 | 0.276 | 0.396 |
| Avg. | | **0.151** | **0.250** | 0.172 | 0.272 | 0.182 | 0.281 | 0.176 | 0.276 | 0.180 | 0.278 | 0.189 | 0.289 | 0.183 | 0.282 | 0.184 | 0.283 |

for a consistent comparison framework with other methods evaluated using this common backbone for classification benchmarks. The main forecasting results presented in Section 4.1 of the paper utilize the ProSAR framework primarily with the CoST-derived backbone. Table 11 serves as an illustrative results for univariate forecasting tasks, comparing the performance of ProSAR's core methodology when integrated with the CoST backbone versus the TS2Vec backbone. As indicated by the metrics, a slight decrease in performance is observed when the TS2Vec backbone is used for these forecasting tasks instead of the CoST backbone, highlighting the effective synergy of the selected CoST-based architecture for ProSAR's forecasting setup. This also demonstrates the adaptability of the ProSAR framework's core components, which are potentially compatible with other encoders, with such integrations being subjects for future exploration.

Table 11: Univariate forecasting results with different backbones.

| Method | ETTh$_1$ | ETTh$_2$ | ETTm$_1$ | Elec. | WTH |
|---|---|---|---|---|---|
| ProSAR (CoST backbone) | 0.068 / 0.198 | 0.142 / 0.289 | 0.045 / 0.151 | 0.338 / 0.328 | 0.160 / 0.285 |
| ProSAR (TS2Vec backbone) | 0.072 / 0.208 | 0.150 / 0.305 | 0.048 / 0.160 | 0.370 / 0.350 | 0.167 / 0.301 |

## C.5 Parameter Sensitivity Analysis

This section explores ProSAR's sensitivity to key hyperparameters. We vary parameters of loss component weights ($\lambda_{\text{intra}}$, $\lambda_{\text{inter}}$) over reasonable ranges and observe their impact on performance on the ETTh1 dataset. Figure 2 presents this analysis, showing Mean Squared Error (MSE) and Mean Absolute Error (MAE) versus hyperparameter values. This analysis helps understand the robustness of ProSAR and provides guidance for hyperparameter selection.

## C.6 Convergence Analysis

To demonstrate the training stability of ProSAR, we plot the training loss curves for five representative datasets. Figure 3 shows the evolution of the total loss $\mathcal{L}_{\text{total}}$, as well as its main components $L_{\text{intra}}$ and $L_{\text{inter}}$ ($L_{\text{inter\_inst}}$ and $L_{\text{inter\_proto}}$), over training epochs. The curves across these diverse datasets indicate that ProSAR generally converges smoothly.

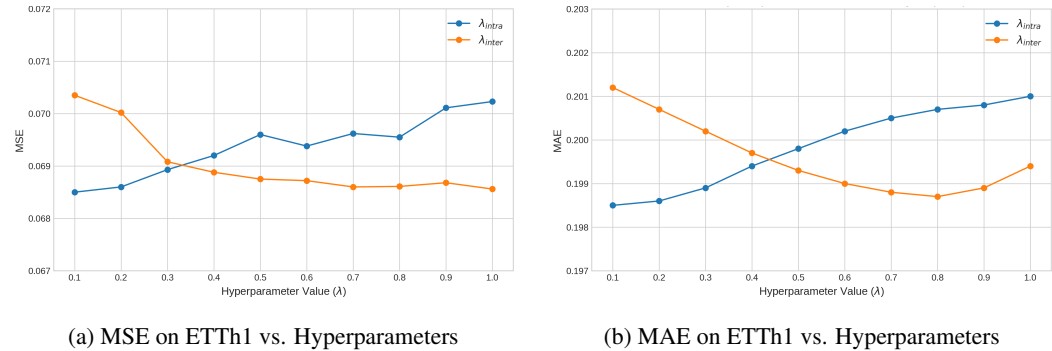

(a) MSE on ETTh1 vs. Hyperparameters      (b) MAE on ETTh1 vs. Hyperparameters

Figure 2: Sensitivity analysis of ProSAR on the ETTh1 dataset, illustrating the impact of key hyperparameter variations on (a) Mean Squared Error (MSE) and (b) Mean Absolute Error (MAE).

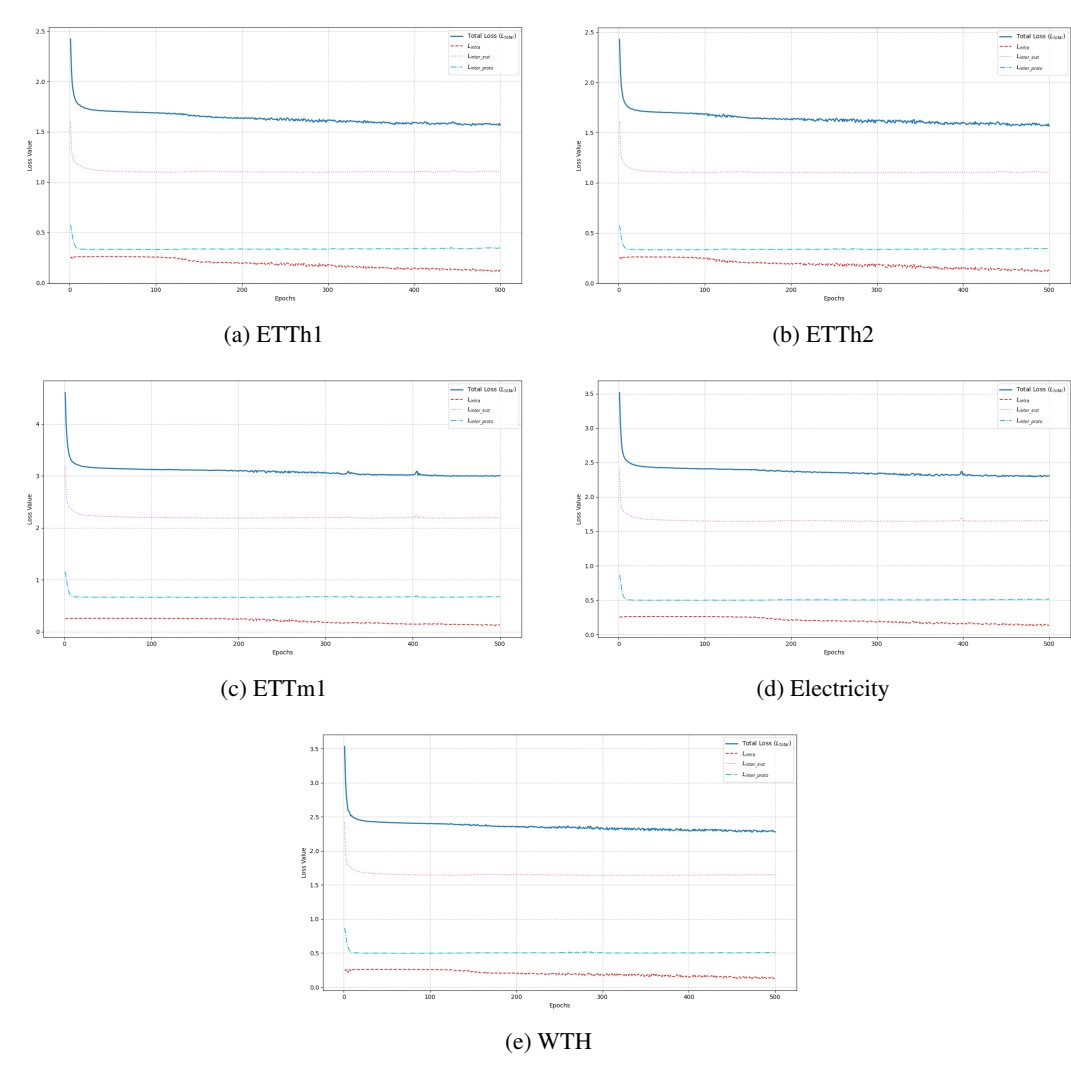

Figure 3: Convergence of ProSAR training losses on five representative datasets: ETTh1, ETTh2, ETTm1, Electricity, and WTH. Each subfigure shows the total loss ($\mathcal{L}_{\text{total}}$) and its primary components over epochs.

# D  VISUALIZATIONS AND QUALITATIVE ANALYSIS

This section provides visual examples and qualitative analysis of ProSAR's key mechanisms, using the Cricket dataset as a representative example for illustration. These visualizations aim to offer a clearer understanding of how ProSAR identifies semantic content and generates diverse, yet semantically consistent, augmented views.

## D.1  VISUALIZATION OF LEARNED TIME-DOMAIN PROTOTYPES

To offer insight into the patterns ProSAR learns to recognize, Figure 4 illustrates the evolution of time-domain prototypes $\{p_k^t\}$. The visualizations compare example prototypes from an early stage of training (left column), where they might appear more noisy or less defined, with their counterparts from a later, more converged stage of training (right column) on the Cricket dataset. This comparison aims to show how these prototypes gradually learn to capture more distinct and semantically meaningful temporal structures as training progresses, refining their ability to guide the augmentation process effectively.

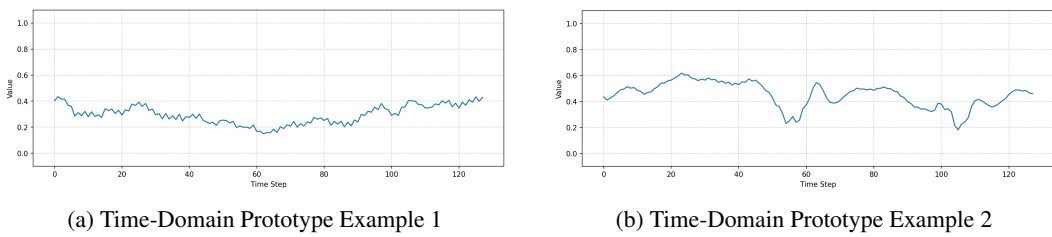

(a) Time-Domain Prototype Example 1      (b) Time-Domain Prototype Example 2

Figure 4: Visualizations of learned time-domain prototypes.

## D.2  VISUALIZATION OF LATENT SPACE PROTOTYPES

To understand the structure of the learned semantic anchors in the latent space, Figure 5 presents two t-SNE visualizations of the latent prototypes $\{p_k^z\}$ derived from the Cricket dataset after training ProSAR. An effective set of latent prototypes should ideally be well-distributed in the latent space, indicating that they capture diverse semantic clusters. These visualizations (from different training stages) help assess the separation and grouping of these latent concepts, which underpins the model's ability to differentiate between semantically distinct time series patterns.

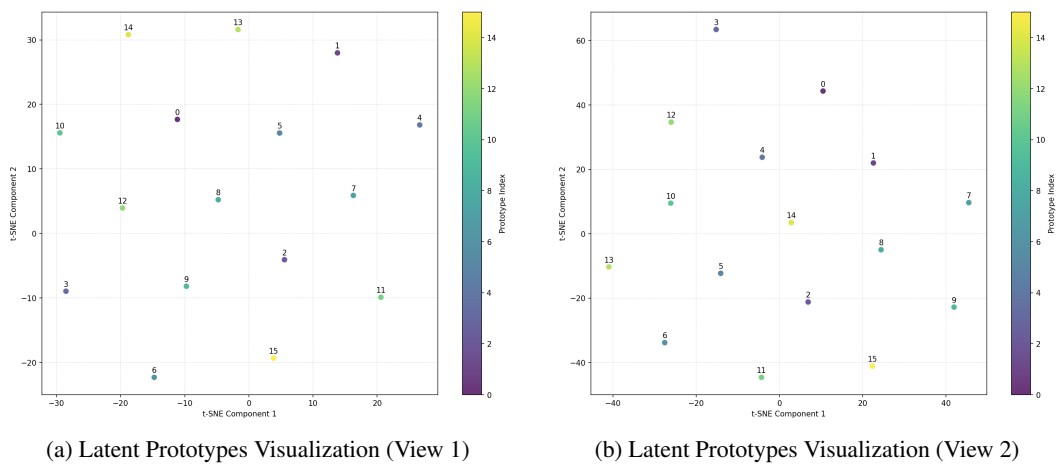

(a) Latent Prototypes Visualization (View 1)      (b) Latent Prototypes Visualization (View 2)

Figure 5: Visualizations of learned latent space prototypes.

## D.3 Visualization of DTW Alignment Matrix

Dynamic Time Warping (DTW) alignment is crucial for ProSAR's semantic segment identification, enabling the model to find correspondences between an input series and a time-domain prototype despite temporal variations. Figure 6 shows examples of DTW alignment matrices. In these matrices, paths that run horizontally or vertically for extended periods indicate significant time insertions or deletions, often corresponding to non-semantic or misaligned segments that might be "broken" or excluded from the core semantic segment. Conversely, paths that approximate a diagonal line, even with local deviations, signify strong temporal alignment and are considered part of the aligned semantic structure.

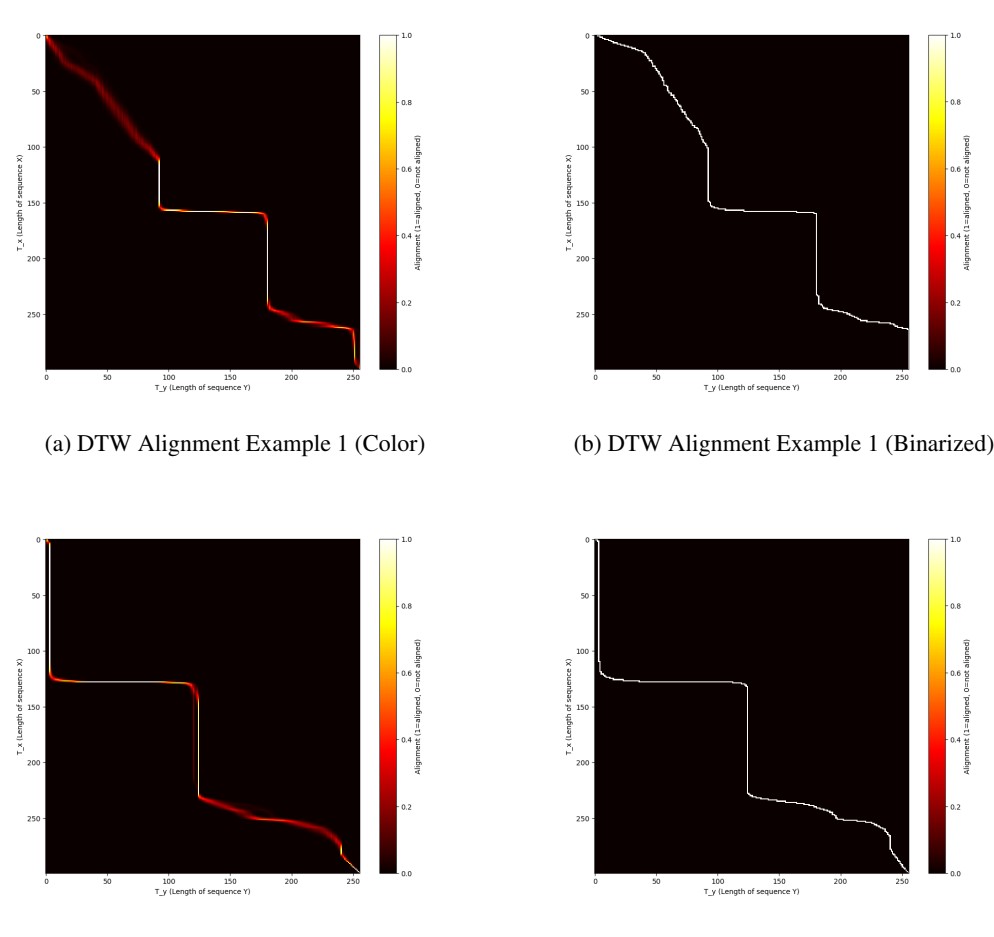

(a) DTW Alignment Example 1 (Color)  (b) DTW Alignment Example 1 (Binarized)

(c) DTW Alignment Example 2 (Color)  (d) DTW Alignment Example 2 (Binarized)

Figure 6: Visualizations of DTW alignment matrices (color indicating path costs) and their binarized counterparts used for segmentation.

## D.4 Visualizations of Semantic Segment Identification and Augmentation

Figure 7 provides a detailed, step-by-step illustration of ProSAR's prototype-guided augmentation mechanism applied to a sample time series from the Cricket dataset. This process aims to preserve core semantic content while introducing diversity, guided by the information-theoretic principles outlined in the main paper. The stages are: (a) The original input series $x$. (b) The specific time-domain prototype $p_k^t$ selected by DTW as most similar to $x$. (c) The identified semantic segment $x_S = x \odot M_x$, extracted from $x$ based on high-quality alignment with $p_k^t$. $M_x$ is the binary mask derived from DTW. (d) The original non-semantic part $x_N = x \odot (1 - M_x)$, representing regions of $x$ with poor alignment to $p_k^t$. (e) The transformed semantic segment $x_S'$. This involves DTW-guided

temporal alignment normalization and controlled noise injection in the frequency domain, designed to preserve essential semantics while discarding P-irrelevant nuisance variability. (f) The perturbed non-semantic part $x'_N$. This part is heavily modified through noise and random sub-segment masking to corrupt P-irrelevant information. (g) The final augmented view $\tilde{x}$, constructed by combining the transformed semantic part and the perturbed non-semantic part: $\tilde{x} = (x'_S \odot M_x) + (x'_N \odot (1 - M_x))$. This visual walkthrough clarifies how learnable prototypes and DTW-guided segmentation enable ProSAR to generate semantically coherent and diverse augmentations.

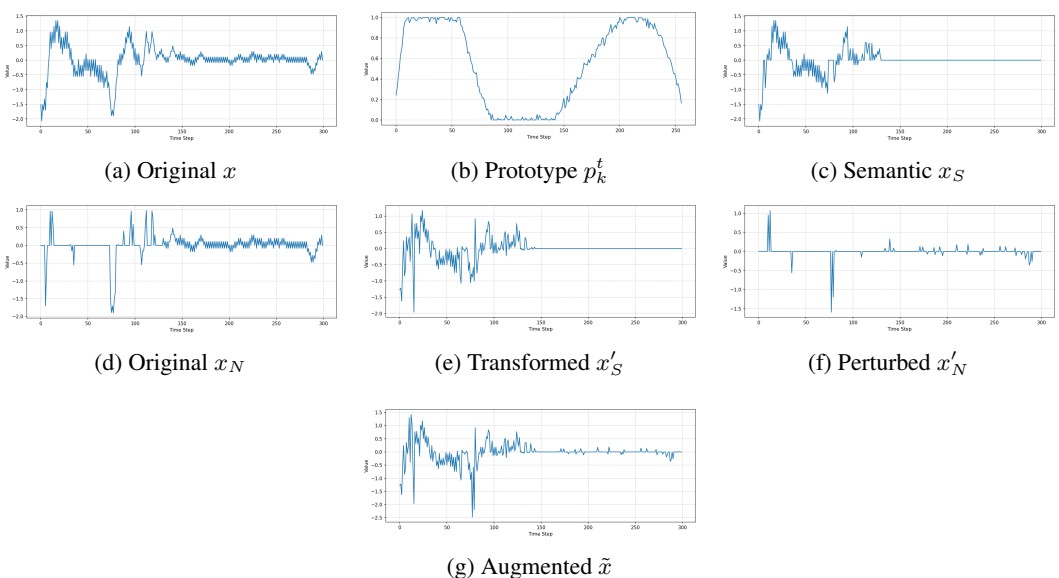

(a) Original $x$      (b) Prototype $p_k^t$      (c) Semantic $x_S$

(d) Original $x_N$      (e) Transformed $x'_S$      (f) Perturbed $x'_N$

(g) Augmented $\tilde{x}$

Figure 7: Step-by-step visualization of ProSAR's prototype-guided semantic augmentation process (a-g).

## E    THEORETICAL DERIVATIONS AND PROOFS

This appendix provides complete and self–contained proofs for the theoretical claims in the main paper. For ease of cross–referencing we follow the numbering of the propositions and theorems used in the main text while introducing additional appendix-only labels where necessary.

**Notation recap.** $X$: raw time series; $\tilde{X} = T(X)$: augmented view; $Z = f_\theta(X)$: latent representation; $P = g(X)$: hard assignment to prototypes; $\beta \in (0, 1)$: IB trade-off parameter.

### E.1    PROOF OF PROPOSITION 3.1 (SEMANTIC INFORMATION DECOMPOSITION)

We must show $I(X; \tilde{X}) = I(P; \tilde{X}) + I(X; \tilde{X} \mid P)$.

**Step 1 (chain rule).** For any triplet $(V_1, V_2, V_3)$ the chain rule for mutual information states $I(V_1, V_2; V_3) = I(V_1; V_3) + I(V_2; V_3 \mid V_1)$. Setting $(V_1, V_2, V_3) = (P, X, \tilde{X})$ yields

$$I(P, X; \tilde{X}) = I(P; \tilde{X}) + I(X; \tilde{X} \mid P). \tag{5}$$

**Step 2 (deterministic prototype map).** Because $P = g(X)$ is deterministic, $H(P, X) = H(X) + H(P \mid X) = H(X)$, and similarly, $H(P, X \mid \tilde{X}) = H(X \mid \tilde{X}) + H(P \mid X, \tilde{X}) = H(X \mid \tilde{X})$. Hence $I(P, X; \tilde{X}) = H(X) - H(X \mid \tilde{X}) = I(X; \tilde{X})$.

**Step 3 (substitution).** Substituting the above equality into equation 5 proves the desired identity. ∎

## E.2 PROOF OF PROPOSITION 3.2 (PROTOTYPE-OPTIMAL AUGMENTATION)

Recall the optimization objective

$$\mathcal{L}(T) = (1 - \beta)\, I(P; \tilde{X}) \, - \, \beta\, I(X; \tilde{X} \mid P), \quad 0 < \beta < 1. \tag{6}$$

**Term 1.** Because of the data–processing inequality (DPI) for the Markov chain $P \leftarrow X \rightarrow \tilde{X}$, $I(P; \tilde{X}) \leq I(P; X)$. The upper bound is attained iff $\tilde{X}$ retains every bit of information that $X$ contains about $P$, i.e. $I(P; \tilde{X}) = I(P; X)$.

**Term 2.** The conditional mutual information $I(X; \tilde{X} \mid P) \geq 0$ with equality *iff* $X$ and $\tilde{X}$ are conditionally independent given $P$ $(X \leftrightarrow P \leftrightarrow \tilde{X})$.

**Optimality conditions.** Maximizing equation 6 therefore requires simultaneously

(i) $I(X; \tilde{X} \mid P) \rightarrow 0$,

(ii) $I(P; \tilde{X}) \rightarrow I(P; X)$.

These two conditions are exactly those stated in the main text. ∎

## E.3 INFORMATION-THEORETIC INTERPRETATION OF PROTOTYPE UPDATES

Prototype refinement can be viewed through the *Information Bottleneck* (IB) lens. Assume the downstream target is an (unknown) random variable $Y_{\text{task}}$ that depends on the representation $Z = f_\theta(X)$. A *good* assignment variable $P$ should

(a) retain as much information as possible about $Y_{\text{task}}$, i.e. maximise $I(P; Y_{\text{task}})$;

(b) yet be a *compressed* summary of $Z$, controlled by $I(P; Z)$.

**IB formulation.** This trade-off is formalized by

$$\max_{q(P|Z)} \; I(P; Y_{\text{task}}) \, - \, \gamma\, I(P; Z), \qquad \gamma > 0. \tag{7}$$

**Self-supervised surrogate.** Because $Y_{\text{task}}$ is unavailable in self-supervised pre-training, we adopt a common surrogate: *maximise $I(P; Z)$ while penalising the complexity of $P$*. Two equivalent forms appear in the literature:

(i) a *rate–distortion* variant $\max_q \; I(P; Z) - \lambda H(P)$ with fixed $\lambda$;

(ii) a *capacity-constrained* variant $\max_q \; I(P; Z)$ s. t. $H(P) \leq C$.

*Remark.* In PROSAR we simply fix the prototype count to $K$; this enforces a constant upper bound $H(P) \leq \log K$, so the capacity-constrained form is satisfied implicitly and we do not write $H(P)$ explicitly in later proofs.

*Remark.* In the absence of labels, keeping the original $-\gamma I(P; Z)$ term would drive $P$ to discard all information. Hence we replace the supervised fidelity term $I(P; Y_{\text{task}})$ by $I(P; Z)$ *and* moves the compression pressure to a separate regulariser such as $H(P)$ or a fixed cluster count $K$.

**On the use of prototypes as semantic proxies.** In Eq. (3) the unknown semantic variable $C$ is replaced by the prototype index $P = g(X)$. This mirrors the replacement of $Y_{\text{task}}$ by $Z$ in the information–bottleneck objective: both substitutions turn unobservable quantities into deterministic proxies that are refined during training. Because $P$ is a deterministic function of $X$, substituting $C \rightarrow P$ does not affect the mutual–information identities, while the iterative refinement loop guarantees that the quality of this proxy can only improve. Extensive experiments confirm that the learned prototypes indeed capture task–relevant semantics.

**Hard clustering as a variational IB solver.** Let the encoder outputs be i.i.d. samples from an isotropic Gaussian mixture, $p(z \mid P = k) = \mathcal{N}(\mu_k, \sigma^2 I)$ with uniform priors. The log-likelihood of data is $\log p(Z \mid \{\mu_k\}) = -\frac{1}{2\sigma^2} \sum_k \sum_{i \in C_k} \|z_i - \mu_k\|_2^2 + \text{const.}$ Maximising this likelihood under *hard assignments* $q(P = k \mid z) = \mathbf{1}[k = \arg\min_j \|z - \mu_j\|]$ is precisely Lloyd's distortion minimization equation 9. Moreover,

$$I(P; Z) = H(Z) - H(Z \mid P) \qquad\qquad \text{(definition)}$$

$$\geq H(Z) - \frac{d}{2N} \sum_{k=1}^{K} \sum_{i \in C_k} \log\!\Big(\tfrac{2\pi e}{d} \|z_i - \mu_k\|_2^2\Big) \quad \text{(Gaussian max–entropy bound)} \tag{8}$$

so *decreasing distortion increases a lower bound on* $I(P; Z)$. Hence any algorithm that *monotonically decreases* equation 9 performs a *greedy ascent* on the IB surrogate equation 7. This directly connects to Theorem E.1 proved next.

Having established the IB perspective, we now formalise the guarantee for *arbitrary* distortion–decreasing hard-clustering (Theorem E.1), and then instantiate it for FINCH and input-space $k$-means.

### E.4 Prototype–IB Optimality for Distortion–Decreasing Hard Clustering

Below we prove a generic result (Theorem E.1) showing that *any* hard-clustering algorithm that monotonically decreases the within–cluster $\ell_2$ distortion also monotonically increases a lower bound on $I(P; Z)$. Afterwards we instantiate the theorem for FINCH and for vanilla $k$-means applied in the input space.

**Theorem E.1** (Prototype-IB Optimality). *Let* $Z = \{z_i\}_{i=1}^{N} \subset \mathbb{R}^d$. *A hard-clustering iteration produces assignments* $P^{(t)} \in [K]^N$ *and centroids* $\mu_k^{(t)}$. *Define the within–cluster $\ell_2$ distortion*

$$D_t = \sum_{k=1}^{K} \sum_{i \in C_k^{(t)}} \big\| z_i - \mu_k^{(t)} \big\|_2^2, \tag{9}$$

*where* $C_k^{(t)} = \{\, i : P_i^{(t)} = k \,\}$ *and* $\mu_k^{(t)} = \dfrac{1}{|C_k^{(t)}|} \displaystyle\sum_{i \in C_k^{(t)}} z_i$. *If the algorithm guarantees* $D_{t+1} \leq D_t$

*for every* $t$, *then*

$$H\big(Z \,|\, P^{(t+1)}\big) \ \leq \ H\big(Z \,|\, P^{(t)}\big), \qquad I\big(P^{(t+1)}; Z\big) \ \geq \ I\big(P^{(t)}; Z\big).$$

*Consequently the sequence* $\{I(P^{(t)}; Z)\}_{t \geq 0}$ *is monotone non-decreasing and converges at a (possibly local) minimum of* $D_t$.

*Proof.* **Upper bound on** $H(Z \mid P)$. For each cluster $k$ the empirical conditional distribution $p_k$ has covariance matrix $\Sigma_k = \frac{1}{|C_k|} \sum_{i \in C_k} (z_i - \mu_k)(z_i - \mu_k)^\top$. Maximum-entropy principle gives $h(p_k) \leq \frac{1}{2} \log\!\big[(2\pi e)^d \det \Sigma_k\big]$. Using A.M.–G.M. inequality $\det \Sigma_k \leq (\frac{1}{d} \text{tr}\, \Sigma_k)^d$ and $\text{tr}\, \Sigma_k = \frac{1}{|C_k|} \sum_{i \in C_k} \|z_i - \mu_k\|_2^2$, we obtain the bound

$$H(Z \mid P) \ \leq \ \frac{d}{2N} \sum_k \sum_{i \in C_k} \log\!\Big(\tfrac{2\pi e}{d} \|z_i - \mu_k\|_2^2\Big). \tag{10}$$

The right-hand side is a strictly increasing function of every squared distance and hence of the aggregate distortion $D_t$.

**Monotone decrease of** $H(Z \mid P)$. Because $D_{t+1} \leq D_t$ by assumption, equation 10 implies $H(Z \mid P^{(t+1)}) \leq H(Z \mid P^{(t)})$.

**Monotone ascent of** $I(P; Z)$. $H(Z)$ is dataset-constant, therefore $I(P^{(t+1)}; Z) - I(P^{(t)}; Z) = H(Z \mid P^{(t)}) - H(Z \mid P^{(t+1)}) \geq 0$. Since distortion is bounded below, $\{D_t\}$ and hence $\{I(P^{(t)}; Z)\}$ converge. $\qquad\square$

E.5 CONSEQUENCES FOR FINCH AND INPUT-SPACE $k$-MEANS

**Corollary E.2** (FINCH updates). *Each merge operation in FINCH (Sarfraz et al., 2019) strictly decreases the distortion equation 9, hence by Theorem E.1 the sequence $I(P_{\mathrm{FINCH}}^{(t)}; Z)$ is monotone non-decreasing.*

*Proof.* Let clusters $A, B$ be merged because $A$'s centroid is $B$'s nearest neighbour and *vice-versa*. Writing $n_A, n_B$ for the sizes and $\mu_A, \mu_B$ for the centroids, standard variance decomposition gives

$$\sum_{z \in A \cup B} \|z - \mu_{A \cup B}\|_2^2 = \sum_{z \in A} \|z - \mu_A\|_2^2 + \sum_{z \in B} \|z - \mu_B\|_2^2 + \frac{n_A n_B}{n_A + n_B} \|\mu_A - \mu_B\|_2^2.$$

Because $A, B$ are mutual nearest neighbours, the cross term is *smaller* than the distortion that would be incurred by keeping them separate and measuring each point in $A$ relative to $\mu_B$ (or vice-versa). Hence the global distortion decreases. □

**Corollary E.3** (Input-space $k$-means grounding). *Lloyd iterations of (mini-batch) $k$-means on DTW-aligned raw segments strictly decrease their within-cluster distortion; therefore Theorem E.1 applies and $I(P_{\mathrm{time}}^{(t)}; S_{\mathrm{raw}})$ monotonically increases.*

*Proof.* Classic proof of Lloyd's algorithm shows that the assignment step followed by centroid re-estimation never increases distortion; equality holds only at a local optimum. Substituting $Z \leftarrow S_{\mathrm{raw}}$ gives the present statement. □

E.6 MONOTONIC IMPROVEMENT OF THE JOINT OBJECTIVE

The last two subsections have established that (a) every *prototype-update* step based on a distortion–decreasing hard-clustering algorithm strictly lowers the conditional entropy $H(Z \mid P)$ and therefore *raises* a provable lower bound on the semantic capacity $I(P; Z)$; and (b) for a *fixed* set of prototype assignments $P$, Proposition 3.2 tells us how to choose an augmentation mapping $T$ that *maximises* the prototype-conditioned IB functional $\mathcal{F}(P, T)$. We now glue these two facts together and show that *alternating* them in a loop produces a training trajectory whose objective value can never decrease.

Recall the prototype-conditioned objective

$$\mathcal{F}(P, T) = (1 - \beta) I(P; \tilde{X}) - \beta I(X; \tilde{X} \mid P), \quad \tilde{X} = T(X), \ 0 < \beta < 1.$$

The first term rewards semantic fidelity of the view $\tilde{X}$, whereas the second term penalises prototype-irrelevant overlap with the original instance $X$.

**Theorem E.4** (Monotone Improvement Loop). *Let the following two-step iteration be applied for $t = 0, 1, 2, \ldots$:*

*(a) **Prototype update.** Given the current encoder $f_{\theta^{(t)}}$ and augmentation $T^{(t)}$, recompute latent representations $Z^{(t)} = f_{\theta^{(t)}}(X)$ and obtain new assignments $P^{(t+1)}$ by a hard-clustering step that satisfies $D_{t+1} \leq D_t$ (e.g. FINCH or Lloyd $k$-means).*

*(b) **Augmentation update.** With the assignments $P^{(t+1)}$ fixed, choose an augmentation $T^{(t+1)} \in \arg\max_T \mathcal{F}(P^{(t+1)}, T)$ .*

*Define the* expected joint objective $\mathcal{J}^{(t)} = \mathbb{E}_X \left[ \mathcal{F}(P^{(t)}, T^{(t)}) \right]$. *Then*

$$\mathcal{J}^{(t+1)} \geq \mathcal{J}^{(t)} \quad \text{for all } t.$$

*Proof.* **Step (a).** $T^{(t)}$ and $f_{\theta^{(t)}}$ are frozen, hence $\tilde{X}$ is unchanged while we *only* replace $P^{(t)}$ by $P^{(t+1)}$. By Corollary E.2 / E.3, the prototype update lowers $D_t$ and therefore raises $I(P; Z)$. Because $\tilde{X}$ is a deterministic function of $X$, the conditional term $I(X; \tilde{X} \mid P)$ is unaffected by merely relabeling the cluster index. Hence

$$\mathcal{F}(P^{(t+1)}, T^{(t)}) \geq \mathcal{F}(P^{(t)}, T^{(t)}). \tag{11}$$

**Step (b).** With $P^{(t+1)}$ fixed, the augmentation step chooses $T^{(t+1)}$ as a global maximiser of $\mathcal{F}(P^{(t+1)}, \cdot)$; therefore

$$\mathcal{F}\big(P^{(t+1)}, T^{(t+1)}\big) \;\geq\; \mathcal{F}\big(P^{(t+1)}, T^{(t)}\big). \tag{12}$$

**Combine.** We chain inequalities 11 and equation 12, then take expectation over the minibatch distribution of $X$, to obtain the desired monotone sequence $\mathcal{J}^{(t)}$. $\qquad\square$

**Interpretation and practice.** Theorem E.4 provides a formal basis for the intuitive *positive-feedback loop* inherent in PROSAR's design: an *improved* set of prototype assignments $P$ (capturing semantics in $Z$ more effectively) enables the augmentation module to generate views $\tilde{X}$ that are *more tightly aligned with these underlying semantics* (as per Proposition 3.2). These higher-quality, semantically focused positive views can sharpen the contrastive loss signals used to train the encoder $f_\theta$. Consequently, this leads to a *better* encoder $f_\theta$, which in turn produces latent representations $Z$ with potentially reduced noise and clearer semantic structure. This clearer structure allows the subsequent clustering step to obtain *even better* prototypes. The process then repeats, ideally under progressively improving conditions.

Each full loop iteration is guaranteed, under the assumptions of the theorem, *never to decrease* the joint objective $\mathcal{J}^{(t)}$. This characteristic can contribute to stable convergence behavior in practice and may reduce the need for delicate scheduling heuristics when alternating between prototype refinement and augmentation module fine-tuning. The inequality in $\mathcal{J}^{(t+1)} \geq \mathcal{J}^{(t)}$ would become an equality if (a) the clustering step reaches a fixed point in terms of distortion (and thus the $I(P; Z)$ lower bound), *and* (b) the current augmentation mapping $T^{(t)}$ already satisfies the optimality conditions of Proposition 3.2 with respect to $P^{(t+1)}$.

# F  DECLARATION ON THE USE OF LARGE LANGUAGE MODELS

In the preparation of this manuscript, a large language model (LLM) was employed as a support tool for a specific, non-core task. Its application was restricted to:

- Language Refinement: Enhancing the clarity, fluency, and grammatical precision of the text.
- Code Structuring: Assisting in the refactoring and commenting of the project's source code for improved readability. project.

We wish to clarify that all foundational scientific contributions of this paper—encompassing the central concept and architectural design of ProSAR, the novel training methodology, and the subsequent analysis of results—originated solely from the human authors. The role of the LLM was strictly confined to the execution-level task of language refinement and did not influence the intellectual conception or strategic direction of this research. The authors have critically reviewed and edited all model-assisted text and take full and final responsibility for the entirety of this work.

