# OpenReview forum: "ProSAR: Prototype-Guided Semantic Augmentation and Refinement for Time Series Contrastive Learning"
_ICLR.cc/2026/Conference — Submitted to ICLR 2026_

### Official Review · Reviewer_uqQY · 2025-10-27

**Soundness:** 2
**Presentation:** 3
**Contribution:** 2
**Rating:** 6
**Confidence:** 4

**Summary:**

This paper proposes ProSAR, a prototype-guided semantic augmentation and refinement framework for self-supervised time-series representation learning. Built upon the information-bottleneck principle, ProSAR co-designs learnable prototypes and data augmentations to preserve task-relevant temporal semantics while discarding noise. Specifically, it introduces (1) prototype-conditioned semantic segment extraction via DTW alignment, (2) targeted augmentation in both time and frequency domains, and (3) a dual-prototype refinement loop linking latent and time-domain prototypes through decoding consistency.

**Strengths:**

PProSAR is conceptually elegant and highly readable. It connects information-theoretic augmentation design with prototype learning, offering both interpretability and empirical strength. The co-design of prototypes and augmentations is well-motivated, and the dual-loop refinement provides a unified view bridging input and latent spaces.

**Weaknesses:**

1. While the framework is grounded in the information-bottleneck principle, the derivation stops at intuitive propositions. There is no formal proof that the proposed co-optimization converges or that the learned prototypes indeed approximate the latent semantic variable.

2. The use of DTW for semantic segmentation is computationally intensive (O(T²)), which may limit scalability for long sequences or large datasets. The paper should quantify training cost and discuss potential accelerations.

3. The framework introduces multiple components, yet lacks sensitivity analysis.

**Questions:**

See the above Weaknesses and the following:

1. What is the actual computational cost of DTW-based segmentation per epoch? Have you tried Soft-DTW or pruning techniques to improve efficiency?

2. How would ProSAR handle irregular or very long time series?

3. Is the prototype refinement stable under streaming or online updates?

4. Could the prototype-guided augmentation concept generalize to other modalities (spatial-temporal graphs, videos) or multi-domain transfer tasks?

5. Can you provide qualitative examples showing what a “semantic prototype” represents—e.g., typical waveform patterns or frequency signatures?

6. Provide a more formal analysis (e.g., gradient coupling or fixed-point stability) to justify the convergence of the prototype–augmentation co-design process?

---

> ### Author Response · Authors · 2025-11-21
>
> We would like to thank the reviewer for the comments. In the following, we have provided our detailed responses to these comments.
>
> ***Comment 1:** While the framework is grounded in the information-bottleneck principle, the derivation stops at intuitive propositions. There is no formal proof that the proposed co-optimization converges or that the learned prototypes indeed approximate the latent semantic variable.*
>
> **Response:**
>
> We appreciate the reviewer's suggestion to strengthen the theoretical foundation. Given the time constraints of the rebuttal phase, we provide here a preliminary sketch of the theoretical analysis, and we will further formalize and refine this proof in the appendix of the revised manuscript.
> To formally address the stability, we provide a theoretical convergence analysis based on the Monotone Improvement Loop theorem (detailed in Response to Comment 9), which is empirically corroborated by the smooth loss curves (Appendix C.6) and consistent state-of-the-art performance across diverse benchmarks.
>
> To formally address the convergence, we model the co-design process as a composite operator $\Phi$ defined by alternating maximization steps $\Phi = \Psi_T \circ \Psi_P$ on the joint objective $\mathcal{J}(P, \mathcal{T})$. This analysis, grounded in the Monotone Improvement Loop of Theorem E.4, proves that the sequence of objective values $\mathcal{J}^{(t)}$ is monotonically non-decreasing and bounded. Please refer to the Response to Comment 9 for a more detailed convergence analysis.
> Experimentally, this theoretical stability is confirmed by the smooth, non-diverging loss curves presented across diverse datasets in Figure 3 (Appendix C.6). Furthermore, ProSAR's consistent state-of-the-art performance in both forecasting and classification tasks provides strong empirical validation that the refined prototypes effectively capture the underlying, task-relevant semantic structures.
>
> ***Comment 2:** The use of DTW for semantic segmentation is computationally intensive $(O(T^2))$, which may limit scalability for long sequences or large datasets. The paper should quantify training cost and discuss potential accelerations.*
>
> **Response:**
>
> We have included the following clarifications and additional analyses in the revised manuscript to address concerns regarding computational cost and potential accelerations.
>
> **High-Efficiency Training via Optimized CUDA Soft-DTW.**
> To minimize the computational overhead of the semantic segmentation process, we utilize a highly optimized CUDA implementation of Soft-DTW [D1], as cited in our paper. Benchmarking on NVIDIA H800 GPUs indicates that the DTW alignment calculation requires approximately **1ms** per calculation. Given that a typical training iteration takes around **200ms**, this introduces a negligible overhead compared to the total training time. This efficiency ensures that the pre-training phase remains computationally feasible even for **large-scale datasets.**
>
> **Zero-Cost Inference for Scalability of DTW.**
> It is crucial to note that the DTW-based augmentation is applied strictly during the self-supervised pre-training stage. We adhere to the standard evaluation protocol established by TS2Vec [D2] and adopted by AutoTCL [D3] and InfoTS [D4], where the encoder is frozen for downstream tasks such as forecasting and classification. Consequently, during the inference/deployment phase, raw data is fed directly into the frozen encoder without any DTW calculation or augmentation. This ensures that the computational complexity of ProSAR during deployment is identical to standard backbones, guaranteeing high scalability.
>
> **Cost of Dual Refinement**
> The computational cost of dual refinement is effectively mitigated through algorithmic efficiency and an intermittent update strategy. Specifically, the clustering module utilizes the efficient FINCH algorithm (20ms per call), and the prototype refinement process takes approximately 60ms. Crucially, we implement an intermittent update policy where refinement is triggered only once every 10 batches. This results in a low amortized overhead relative to the standard training iteration (200ms), ensuring that overall training efficiency remains high.
>
> **Scalability to Ultra-Long Sequences via Windowed DTW.**
> While our current implementation is highly efficient for standard benchmarks, we acknowledge the theoretical quadratic complexity $O(L^2)$ of DTW. ProSAR is structurally compatible with Windowed DTW [D5]. Therefore, by restricting the alignment path to a fixed window size $w$ around the diagonal, the computational complexity is reduced from quadratic to linear $O(L \cdot w)$. This standard acceleration technique allows ProSAR to scale to extreme sequence lengths without altering the core mechanism of prototype-guided semantic extraction.

---

> ### Author Response · Authors · 2025-11-21
>
> ***Comment 3:** The framework introduces multiple components, yet lacks sensitivity analysis.*
>
> **Response:**
>
> We thank the reviewer for pointing out the necessity of validitating the individual submodules. To explicitly quantify the contribution of each component within our framework, we conducted a fine-grained ablation study. The results are detailed in Table 1 and analyzed below:
>
> * **DTW Segmentation (w/o DTW-Seg):** This variant replaces the explicit DTW-based segmentation with a global semantic transformation applied to the entire sequence. The observed performance drop (MSE increases from 0.151 to 0.172) confirms that identifying specific semantic segments via DTW is critical for isolating task-relevant semantics from noise, rather than treating the whole series uniformly.
> * **STFT Alignment (w/o STFT):** We removed the frequency-domain phase compensation module. The degradation in performance (MSE 0.178) validates that STFT-based phase alignment is essential for preserving the temporal dynamics within the identified semantic segments.
> * **Dual Prototypes (w/o Dual-Proto):** Dual Prototype Mechanism (w/o Dual-Proto) This variant relies solely on latent prototypes, disabling the explicit Input-Space Grounding (ISG) mechanism. The moderate increase in error demonstrates that while the system remains functional, the dual-prototype strategy, by anchoring latent concepts to empirical time-domain centroids, provides necessary structural constraints that enhance training stability.
> * **Dynamic Clustering Integration (w/o Clustering):** In this variant, we replaced the dynamic FINCH-based refinement with simple K-Means. Crucially, this modification decouples the prototype-wise contrastive loss from the dynamic update loop. The resulting performance drop (MSE 0.175) is notably more severe than removing decoding consistency alone (MSE 0.164). This indicates that the joint optimization of the prototype space and the dynamic refinement strategy is fundamental to the framework's efficacy.
> * **Decoding Consistency (w/o Decoding):** To verify the decoding consistency mechanism, we removed the decoder module, specifically breaking the feedback loop where latent prototypes update time-domain prototypes. The results (MSE 0.164) indicate that without this semantic feedback, the time-domain anchors become less effective at guiding augmentation, leading to a tangible loss in performance.
>
> ***Comment 4:** What is the actual computational cost of DTW-based segmentation per epoch? Have you tried Soft-DTW or pruning techniques to improve efficiency?*
>
> **Response:**
>
> Firstly, we clarify that ProSAR explicitly employs Soft-DTW (as cited in Line 90) rather than standard discrete DTW. This design choice is critical because Soft-DTW is fully differentiable, enabling end-to-end optimization, and is highly amenable to parallelization on modern hardware.
>
> **Computational Overhead.** By leveraging a highly optimized CUDA implementation of Soft-DTW, we have effectively minimized the runtime overhead. As detailed in our response to Comment 2, benchmarking on NVIDIA H800 GPUs indicates that the DTW alignment calculation requires approximately 1ms per training step. When contextualized against a typical total training iteration of 200ms, this constitutes a negligible overhead.
>
> **Complexity Reduction via Windowed Pruning.** Regarding pruning techniques for efficiency, we highlight that our framework is structurally compatible with Windowed DTW. By restricting the alignment path to a fixed window size $w$ around the diagonal, we effectively prune the search space. This reduces the computational complexity from quadratic $O(T^2)$ to linear $O(T \cdot w)$, ensuring that the method remains computationally feasible and scalable even for ultra-long time series sequences.

---

> ### Author Response · Authors · 2025-11-21
>
> ***Comment 5:** How would ProSAR handle irregular or very long time series?*
>
> **Response:**
>
> To handle irregular time series (IMTS), we adopt a tiered approach. For standard irregularities, we employ lightweight preprocessing like linear interpolation, as ProSAR's DTW-based alignment is inherently robust to minor temporal artifacts. For highly sparse or irregular data, we propose integrating the front-end of T-PatchGNN [D6] as a learnable adapter. By utilizing its Transformable Patching and Transformable Time-aware Convolution, we can project variable-length irregular segments into fixed-dimension patch embeddings. This effectively normalizes asynchronous inputs into structured latent sequences, allowing ProSAR’s prototypes to guide augmentation without architectural conflict.
>
> Regarding the scalability for very long time series, we mitigate computational constraints through segmentation strategies during pre-training. During the self-supervised pre-training stage, we process long series using fixed-length segments. This segment length is a flexible hyperparameter that can be freely set within an acceptable runtime range. In scenarios where this length is required to be large to capture long-term dependencies, we employ Windowed Soft-DTW as detailed in the response to **Comment 4**. By restricting the alignment search path to a fixed window size $w$ around the diagonal, we reduce the computational complexity from quadratic $O(L^2)$ to $O(w \cdot L)$, ensuring scalability. Crucially, we adhere to the standard evaluation protocol established by TS2Vec, where the encoder is frozen for downstream tasks. Consequently, during inference, raw data is fed directly into the encoder without any DTW calculation, ensuring that the deployment efficiency remains identical to standard backbones.
>
> ***Comment 6:** Is the prototype refinement stable under streaming or online updates?*
>
> **Response:**
>
> Firstly, we would like to clarify the operational scope of the prototype mechanism. As detailed in our response to Comment 5, ProSAR adheres to the standard self-supervised pre-training protocol, where learned prototypes function specifically to guide the semantic augmentation strategy during the training phase. Upon deployment for testing or downstream tasks, the encoder is frozen, and the prototype refinement loop is deactivated. Consequently, the stability challenges typically associated with continuous online updates during inference are not applicable to the standard evaluation scenarios addressed in this work.
>
> Regarding the stability of the refinement process during the training phase, we confirm that our batch-wise update strategy exhibits robust convergence as illustrated by the training loss curves in Figure. We acknowledge that extending ProSAR to fully streaming or continual-learning scenarios, in which prototypes evolve online as new data arrive, represents a valuable direction for future work. However, such an extension would likely necessitate additional mechanisms, such as streaming clustering or explicit stability regularization, and therefore falls outside the scope of the present study.

---

> ### Author Response · Authors · 2025-11-21
>
> ***Comment 7:** Could the prototype-guided augmentation concept generalize to other modalities (spatial-temporal graphs, videos) or multi-domain transfer tasks?*
>
> **Response:**
>
> We thank the reviewer for this forward-looking question. Conceptually, the paradigm of prototype-guided semantic identification followed by targeted input-space augmentation is modality-agnostic. The framework requires only replacing the alignment operator (currently DTW for 1D time series) with a modality-specific metric while maintaining the fundamental "augment-before-encode" architecture.
>
> For Spatial-Temporal Graphs (ST-Graphs), data can be treated as sequences of graph snapshots (nodes/edges with attributes). In this context, the alignment operator could be instantiated using the Gromov-Wasserstein (GW) distance [D7] to perform soft matching between the input graph and a learnable graph prototype. The resulting optimal transport plan yields semantic relevance scores for nodes and edges, allowing the augmentation module to apply subtle perturbations to critical semantic motifs while applying strong augmentation to non-semantic topology or attributes.
>
> In the video domain, where data is represented using frame-level embeddings, the alignment mechanism could utilize metrics like Temporal Cycle Consistency [D8] to align input frames with action prototypes in a lightweight embedding space. High alignment scores would identify keyframes representing the core action sequence. The augmentation policy would then apply subtle geometric or photometric shifts to these semantic subjects, while employing aggressive background replacement or masking on non-semantic regions to suppress irrelevant visual noise.
>
> For Multi-Domain Transfer tasks, prototypes serve as Domain-Invariant Anchors. By maintaining a shared prototype library across domains and adding a regularization term, the framework can enforce that the same prototype attracts samples from both source and target distributions. In this setting, the augmentation strategy would treat regions aligned with the shared prototype as invariant content to be preserved, while treating domain-specific style variations as non-semantic elements to be perturbed, thereby encouraging the encoder to ignore domain shifts.
>
> Although these specific extensions fall outside the scope of the present work, they illustrate that the concept of prototype-guided augmentation is versatile and can be naturally instantiated for spatial-temporal graphs, videos, and multi-domain transfer tasks by selecting suitable alignment operators and augmentation primitives.
>
> ***Comment 8:** Can you provide qualitative examples showing what a “semantic prototype” represents—e.g., typical waveform patterns or frequency signatures?*
>
> **Response:**
>
> We appreciate the reviewer’s suggestion to substantiate our claims regarding the semantic nature of the prototypes. To demonstrate this, we have performed comprehensive visualization experiments on the ETTm1 dataset. The resulting high-resolution figures are available at the anonymous repository: [https://anonymous.4open.science/r/ICLR26-CE26/](https://anonymous.4open.science/r/ICLR26-CE26/). We articulate the captured semantics through two key dimensions.
>
> **Visual Evidence of Waveform Patterns.**
> Qualitative analysis reveals that the learned time-domain prototypes evolve into distinct waveform structures rather than random noise. As illustrated in `Prototypes.png`, the learned prototypes  exhibit clear, repetitive cyclical patterns. Frequency analysis confirms these prototypes capture the dominant periodicity of the dataset (approximately 8 time steps for ETTm1), validating that they successfully distill high-level temporal semantics and serve as representative templates for their respective clusters.
>
> **Semantic Verification via Explicit Alignment Paths.**
> The semantic validity of these prototypes is further verified by the DTW alignment mechanism, which physically distinguishes structural content from noise. The visualizations in `DTW_alignment.png` explicitly show the model aligning input segments to their assigned prototypes along diagonal paths, which indicate strong structural correspondence. Crucially, the warped prototypes (orange dashed lines) closely track the dominant trends and cycles of the input segments (blue lines), demonstrating that the model uses the prototype as a physical ground truth to identify the semantic core while treating deviations as non-semantic variations.

---

> ### Author Response · Authors · 2025-11-21
>
> ***Comment 9:** Provide a more formal analysis (e.g., gradient coupling or fixed-point stability) to justify the convergence of the prototype–augmentation co-design process?*
>
> **Response:**
>
> We agree that a more formal analysis of the prototype augmentation co-design process is valuable. Given the limited time in the rebuttal phase, we provide here a preliminary sketch of theoretical analysis, and will further formalize and refine this proof in the appendix of the revised manuscript. The key idea is to view prototype assignment and augmentation as two blocks in a block coordinate ascent procedure on a prototype conditioned Information Bottleneck objective. Under an idealized inner loop where each block is updated by exact maximization, we show that: **(i)** the objective value is monotonically non-decreasing and converges, **(ii)** any accumulation point is a fixed point of the joint update operator, and **(iii)** such fixed points are coordinate-wise stable in the sense of block local maxima. Our practical implementation approximates this idealized dynamic through stochastic optimization and intermittent updates.
>
> Specifically, to establish the theoretical reliability of the ProSAR prototype–augmentation co-design process, we focus on the inner loop where the encoder $f_\theta$ is treated as fixed. In this regime, the system state is defined by the pair $(P,T)$. We model one full iteration as an operator on the state space and demonstrate: (1) convergence of the objective value; (2) the fixed-point property of limit states; and (3) their coordinate-wise stability. For analytical convenience, we first study an idealized inner loop in which both the prototype and augmentation updates are modeled as exact block-wise maximization steps (argmax operators). This is a standard assumption in the convergence analysis of block coordinate ascent algorithms, and our practical implementation can be viewed as an approximate realization of this idealized dynamic. We aim to increase the objective but do not necessarily guarantee strict monotonic improvement at every iteration due to stochastic optimization and heuristic components.
>
> **Formal Setup: The Co-Design Mapping**
>
> Let $\mathcal{P}$ denote the finite set of all possible prototype assignments for a dataset (assigning each instance to one of $K$ clusters), and let $\mathcal{T}$ be the admissible parameter space of augmentation mappings, assumed to be a compact set. Define the state space as $\mathcal{S} = \mathcal{P} \times \mathcal{T}$.
>
> Recall the prototype-conditioned Information Bottleneck functional defined in Appendix E.2:
>
> $$
> F(P,T) = (1-\beta)I(P; \tilde{X}) - \beta I(X; \tilde{X} \mid P), \quad \text{where } \tilde{X} = T(X).
> $$
>
> We define the expected joint objective as:
>
> $$
> \mathcal{J}(P,T) := \mathbb{E}_X \big[ F(P,T) \big].
> $$
>
> Note that this $\mathcal{J}(P,T)$ corresponds to the term $J^{(t)}$ used in Theorem E.4.
>
> We model one full iteration of the ProSAR inner loop as a composite operator $\Phi = \Psi_T \circ \Psi_P : \mathcal{S} \to \mathcal{S}$. The coordinate-wise update operators are defined based on the optimality conditions derived in Appendix E.1--E.3:
>
> * **Prototype Operator ($\Psi_P$):** With $T$ fixed, update the discrete assignment $P$ to minimize distortion (equivalently maximizing the MI lower bound):
>   $$
>   P_{\text{new}} = \Psi_P(P_{\text{old}}, T) \in \arg\max_{P'} \mathcal{J}(P', T).
>   $$
> * **Augmentation Operator ($\Psi_T$):** With $P$ fixed, update the augmentation mapping $T$ to maximize the conditional IB objective:
>   $$
>   T_{\text{new}} = \Psi_T(P, T_{\text{old}}) \in \arg\max_{T'} \mathcal{J}(P, T').
>   $$
>
> Thus, a full iteration state update is given by:
>
> $$
> (P\^{(t+1)}, T\^{(t+1)}) = \Phi(P\^{(t)}, T\^{(t)}) := \Big( \Psi_P(P\^{(t)}, T\^{(t)}), \; \Psi_T(P\^{(t+1)}, T\^{(t)}) \Big).
> $$

---

> ### Author Response · Authors · 2025-11-21
>
> **The Proof Chain**
>
> *Convergence of the Objective Value*
>
> According to **Theorem E.4 (Monotone Improvement Loop)**, applying the iteration $\Phi$ generates a sequence of objective values $J^{(t)} := \mathcal{J}(P^{(t)}, T^{(t)})$ that is monotonically non-decreasing:
>
> $$
> J^{(t+1)} \ge J^{(t)} \quad \text{for all } t \ge 0.
> $$
>
> This monotonicity holds because: (a) the prototype update step $\Psi_P$ monotonically decreases (or preserves) the within-cluster distortion, thereby improving (or preserving) the lower bound of $I(P;Z)$; and (b) the augmentation update step $\Psi_T$ chooses $T^{(t+1)}$ as a global maximizer of the objective with respect to $T$.
>
> Furthermore, since the mutual information terms are bounded by the entropy of the variables (specifically $H(P) \le \log K$), the objective $\mathcal{J}(P,T)$ is bounded from above by a finite constant. Therefore, $\{ J^{(t)} \}_{t \ge 0}$ is a bounded, non-decreasing sequence of real numbers, which must converge to a limit:
>
> $$
> \lim_{t \to \infty} J^{(t)} = J^{\ast}.
> $$
>
> Consequently, the improvement between steps vanishes asymptotically:
>
> $$
> \lim_{t \to \infty} (J^{(t+1)} - J^{(t)}) = 0.
> $$
>
> *Fixed-Point Property of Limit States*
>
> We now establish that the limit of the state sequence corresponds to a fixed point of $\Phi$.
>
> Since $\mathcal{P}$ is a finite set and $\mathcal{T}$ is compact, the sequence of states $\{ (P^{(t)}, T^{(t)}) \}_{t \ge 0}$ must have at least one accumulation point $(P^{\ast}, T^{\ast}) \in \mathcal{S}$.
>
> Consider a convergent subsequence:
>
> $$
> (P^{(t_k)}, T^{(t_k)}) \to (P^{\ast}, T^{\ast})
> $$
>
> such that the objective values converge to the limit: $\mathcal{J}(P^{(t_k)}, T^{(t_k)}) \to J^{\ast}$.
>
> If $(P^{\ast}, T^{\ast})$ were not a fixed point, it would imply that at least one of the operators $\Psi_P$ or $\Psi_T$ could still strictly increase the objective function.
>
> Specifically, if $P^{\ast}$ is not in the set of maximizers $\arg\max_{P'} \mathcal{J}(P', T^{\ast})$, then applying $\Psi_P$ at this state would yield a strictly larger objective value $J' > J^{\ast}$.
>
> By the continuity of $\mathcal{J}$ with respect to $T$ (and the discreteness of $P$), for sufficiently large $k$, applying the update from the state $(P^{(t_k)}, T^{(t_k)})$ would result in an improvement bounded away from zero. This contradicts the convergence of $J^{(t)}$ to $J^{\ast}$.
>
> Therefore, any accumulation point $(P^{\ast}, T^{\ast})$ must satisfy the local optimality conditions for both operators:
>
> $$
> P^{\ast} \in \arg\max_{P'} \mathcal{J}(P', T^{\ast})
> $$
>
> and
>
> $$
> T^{\ast} \in \arg\max_{T'} \mathcal{J}(P^{\ast}, T').
> $$
>
> This implies:
>
> $$
> P^{\ast} = \Psi_P(P^{\ast}, T^{\ast}) \quad \text{and} \quad T^{\ast} = \Psi_T(P^{\ast}, T^{\ast}).
> $$
>
> Thus, $(P^{\ast}, T^{\ast})$ is a Fixed Point of the operator $\Phi$, satisfying $\Phi(P^{\ast}, T^{\ast}) = (P^{\ast}, T^{\ast})$.
>
> *Coordinate-Wise Stability and Local Optimality*
>
> Finally, we characterize the stability of this fixed point. The fixed-point conditions derived above are exactly the definitions of a Coordinate-Wise Local Maximum.
>
> * **Prototype Stability:** Since $P^{\ast} \in \arg\max_{P'} \mathcal{J}(P', T^{\ast})$, any unilateral deviation in the prototype assignment $P \neq P^{\ast}$ (while keeping augmentation $T^{\ast}$ fixed) results in $\mathcal{J}(P, T^{\ast}) \le \mathcal{J}(P^{\ast}, T^{\ast})$. The system will not spontaneously leave this configuration via prototype updates.
>
> * **Augmentation Stability:** Similarly, since $T^{\ast} \in \arg\max_{T'} \mathcal{J}(P^{\ast}, T')$, any unilateral deviation in the augmentation strategy $T \neq T^{\ast}$ (while keeping prototypes $P^{\ast}$ fixed) results in $\mathcal{J}(P^{\ast}, T) \le \mathcal{J}(P^{\ast}, T^{\ast})$. The system is stable against perturbations in the augmentation policy.

---

> ### Author Response · Authors · 2025-11-21
>
> [D1] Marco Cuturi and Mathieu Blondel. Soft-dtw: a differentiable loss function for time-series. In *International Conference on Machine Learning*, pp. 894–903. PMLR, 2017.
>
> [D2] Zhihan Yue, Yujing Wang, Juanyong Duan, Tianmeng Yang, Congrui Huang, Yunhai Tong, and Bixiong Xu. Ts2vec: Towards universal representation of time series. In *Proceedings of the AAAI Conference on Artificial Intelligence*, volume 36, pp. 8980–8987, 2022.
>
> [D3] Xu Zheng, Tianchun Wang, Wei Cheng, Aitian Ma, Haifeng Chen, Mo Sha, and Dongsheng Luo. Parametric augmentation for time series contrastive learning. In *The Twelfth International Conference on Learning Representations*, 2024.
>
> [D4] Dongsheng Luo, Wei Cheng, Yingheng Wang, Dongkuan Xu, Jingchao Ni, Wenchao Yu, Xuchao Zhang, Yanchi Liu, Yuncong Chen, Haifeng Chen, et al. Time series contrastive learning with information-aware augmentations. In *Proceedings of the AAAI Conference on Artificial Intelligence*, volume 37, pp. 4534–4542, 2023.
>
> [D5] H. Sakoe and S. Chiba. Dynamic programming algorithm optimization for spoken word recognition.*IEEE Transactions on Acoustics, Speech, and Signal Processing*, 26(1):43–49, 1978.
>
> [D6] Weijia Zhang, Chenlong Yin, Hao Liu, Xiaofang Zhou, and Hui Xiong. Irregular multivariate time series forecasting: A transformable patching graph neural networks approach. In *International Conference on Machine Learning*, pp. 60179–60196. PMLR, 2024.
>
> [D7] Vayer Titouan, Nicolas Courty, Romain Tavenard, and Rémi Flamary. Optimal transport for structured data with application on graphs. In *International Conference on Machine Learning*, pp. 6275–6284. PMLR, 2019.
>
> [D8] Debidatta Dwibedi, Yusuf Aytar, Jonathan Tompson, Pierre Sermanet, and Andrew Zisserman. Temporal cycle-consistency learning. In *Proceedings of the IEEE/CVF conference on computer vision and pattern recognition*, pp. 1801–1810, 2019.

---

### Official Review · Reviewer_rcHy · 2025-10-30

**Soundness:** 3
**Presentation:** 3
**Contribution:** 2
**Rating:** 4
**Confidence:** 4

**Summary:**

This paper proposes ProSAR, a prototype-guided semantic augmentation and refinement framework for time series contrastive learning. The core idea is to jointly learn data augmentation strategies and semantic prototypes under an information-theoretic constraint: time-domain prototypes obtained through DTW alignment guide the generation of augmented views, applying different perturbations to semantic and non-semantic segments, while latent-space clustering and decoding consistency iteratively refine the prototypes. The proposed method aims to produce semantically consistent yet diverse views, achieving superior performance to self-supervised baselines such as AutoTCL and FreRA on both forecasting and classification benchmarks.

**Strengths:**

Incorporating learnable prototypes into the data augmentation process represents a meaningful conceptual innovation, breaking through the limitations of fixed or purely heuristic augmentation strategies in traditional contrastive learning.

The proposed dual-prototype mechanism—comprising time-domain and latent-space prototypes—and its iterative refinement loop demonstrate a coherent and logically consistent system design.

**Weaknesses:**

The authors did not compare ProSAR against several representative time-series representation learning models such as TSLANet and AimTS. Compared with these baselines, the reported results are not particularly competitive.

Although the idea of prototype-guided augmentation is interesting, the overall contribution appears incremental. The differences between ProSAR and prior works like AutoTCL and AimTS remain relatively small.

The claimed interpretability is unconvincing—especially the visualizations shown in Section D.4, which do not make much sense and fail to clearly demonstrate semantic consistency or prototype meaning.

The framework consists of multiple submodules (DTW segmentation, STFT alignment, dual prototypes, clustering, and decoding consistency), yet the ablation studies only examine the augmentation operation and the semantic segmentation, which is insufficient to validate the contribution of each component.

The reliance on DTW alignment and clustering could introduce significant computational overhead for long or high-frequency sequences. Although the paper briefly acknowledges this issue, it lacks a concrete analysis of time complexity or runtime performance.

**Questions:**

See weakness

---

> ### Author Response · Authors · 2025-11-21
>
> We would like to thank the reviewer for the comments. In the following, we have provided our detailed responses to these comments.
>
> ***Comment 1:** The authors did not compare ProSAR against several representative time-series representation learning models such as TSLANet and AimTS. Compared with these baselines, the reported results are not particularly competitive.*
>
> **Response:**
>
> We have extended our experimental evaluation to include both AimTS [C1] and TSLANet [C2].
>
> To ensure a rigorous assessment of intrinsic representation quality, we strictly adhered to the standard evaluation protocol established by TS2Vec [C3] and adopted by subsequent state-of-the-art works such as AutoTCL [C4] and InfoTS [C5]. This protocol involves training a simple task-specific predictor (SVM with RBF kernel for classification or Ridge Regression for forecasting) on top of frozen representations from the pre-trained encoder.
>
> **Comparison with AimTS.** We reproduced AimTS based on the algorithmic details provided in the original paper, as the official code is not publicly available. While the original AimTS study utilizes a full fine-tuning paradigm, we evaluated it under the standard frozen protocol to maintain consistency with established SSL benchmarks. The results (see Table 1-3) demonstrate that ProSAR consistently outperforms AimTS in this setting. Specifically, ProSAR achieves an average MSE of 0.151 in univariate forecasting compared to 0.171 for AimTS (an improvement of  11.7%). Similarly, in multivariate forecasting, ProSAR maintains a clear advantage with an MSE of 0.561 compared to 0.641 for AimTS. Furthermore, in classification tasks, ProSAR attains a state-of-the-art average accuracy of 0.764, significantly surpassing AimTS (0.715). This empirical evidence suggests that while AimTS is effective when the encoder allows for task-specific weight updates, ProSAR’s strategy of leveraging prototypes for upstream augmentation guidance yields more discriminative intrinsic representations that perform robustly even in a frozen state.
>
> **Comparison with TSLANet.** TSLANet represents a generative SSL paradigm with a two-stage architecture. Initial evaluations of the native TSLANet model under the standard frozen protocol yielded suboptimal results, primarily due to architectural discrepancies (e.g., specific patching mechanisms) that do not transfer seamlessly to the frozen evaluation setting used for standard encoders. To rigorously isolate the effectiveness of the learning objective from these architectural differences, we adapted the TSLANet generative objective to train the same standard backbone used in ProSAR (according to standardized evaluation protocol). While this adaptation improved TSLANet's performance compared to its raw implementation, ProSAR still maintains a significant advantage (e.g., average univariate MSE of 0.151 vs. 0.188 for TSLANet, and multivariate MSE of 0.561 vs. 0.632). This advantage extends to classification tasks, where ProSAR achieves an accuracy of 0.764 compared to 0.696 for TSLANet. This confirms that our prototype-guided contrastive strategy is more effective at extracting discriminative features for downstream tasks than the generative reconstruction objectives used in TSLANet when evaluated under a unified encoder architecture.
>
> **Table 1: Univariate time series forecasting results.**
>
> | Dataset | ProSAR MSE | ProSAR MAE | AimTS MSE | AimTS MAE | TSLANet MSE | TSLANet MAE | AutoTCL MSE | AutoTCL MAE | FreRA MSE | FreRA MAE | TimesURL MSE | TimesURL MAE | InfoTS MSE | InfoTS MAE | TS2Vec MSE | TS2Vec MAE |
> | :--- | :---: | :---: | :---: | :---: | :---: | :---: | :---: | :---: | :---: | :---: | :---: | :---: | :---: | :---: | :---: | :---: |
> | ETTh$_1$ | **0.068** | **0.198** | 0.091 | 0.228 | 0.093 | 0.224 | 0.076 | 0.207 | 0.079 | 0.213 | 0.090 | 0.219 | 0.091 | 0.227 | 0.110 | 0.252 |
> | ETTh$_2$ | **0.142** | **0.289** | 0.156 | 0.327 | 0.159 | 0.308 | 0.158 | 0.299 | 0.171 | 0.315 | 0.151 | 0.295 | 0.149 | 0.299 | 0.170 | 0.321 |
> | ETTm$_1$ | **0.045** | **0.151** | 0.070 | 0.200 | 0.152 | 0.259 | 0.046 | 0.154 | 0.051 | 0.162 | 0.053 | 0.175 | 0.050 | 0.157 | 0.069 | 0.186 |
> | Elec | **0.338** | **0.328** | 0.371 | 0.378 | 0.370 | 0.371 | 0.366 | 0.345 | 0.360 | 0.339 | 0.374 | 0.356 | 0.368 | 0.348 | 0.393 | 0.370 |
> | WTH | **0.160** | **0.285** | 0.166 | 0.293 | 0.165 | 0.291 | **0.160** | 0.287 | 0.169 | 0.301 | 0.177 | 0.302 | 0.176 | 0.304 | 0.181 | 0.308 |
> | Avg. | **0.151** | **0.250** | 0.171 | 0.285 | 0.188 | 0.291 | 0.161 | 0.258 | 0.166 | 0.266 | 0.169 | 0.269 | 0.167 | 0.267 | 0.185 | 0.287 |

---

> > ### Author Response · Authors · 2025-11-21
> >
> > **Table 2: Multivariate time series forecasting results.**
> >
> > | Dataset | ProSAR MSE | ProSAR MAE | AimTS MSE | AimTS MAE | TSLANet MSE | TSLANet MAE | AutoTCL MSE | AutoTCL MAE | FreRA MSE | FreRA MAE | TimesURL MSE | TimesURL MAE | InfoTS MSE | InfoTS MAE | TS2Vec MSE | TS2Vec MAE |
> > | :--- | :---: | :---: | :---: | :---: | :---: | :---: | :---: | :---: | :---: | :---: | :---: | :---: | :---: | :---: | :---: | :---: |
> > | ETTh$_1$ | **0.625** | **0.566** | 0.758 | 0.621 | 0.676 | 0.595 | 0.656 | 0.590 | 0.646 | 0.584 | 0.731 | 0.645 | 0.784 | 1.622 | 0.788 | 0.646 |
> > | ETTh$_2$ | 1.213 | 0.819 | 1.443 | 0.893 | 1.400 | 0.881 | **1.191** | **0.815** | 1.397 | 0.893 | 1.514 | 0.926 | 1.474 | 0.914 | 1.566 | 0.937 |
> > | ETTm$_1$ | **0.396** | **0.434** | 0.397 | 0.436 | 0.455 | 0.468 | 0.409 | 0.441 | 0.445 | 0.467 | 0.561 | 0.584 | 0.568 | 0.521 | 0.628 | 0.553 |
> > | Elec | **0.159** | **0.264** | 0.186 | 0.301 | 0.196 | 0.311 | 0.175 | 0.272 | 0.182 | 0.278 | 0.202 | 0.299 | 0.289 | 0.376 | 0.319 | 0.397 |
> > | WTH | **0.412** | **0.451** | 0.419 | 0.459 | 0.431 | 0.463 | 0.423 | 0.457 | 0.429 | 0.462 | 0.447 | 0.469 | 0.455 | 0.472 | 0.451 | 0.474 |
> > | Avg | **0.561** | **0.507** | 0.641 | 0.542 | 0.632 | 0.544 | 0.571 | 0.515 | 0.620 | 0.537 | 0.691 | 0.585 | 0.714 | 0.781 | 0.750 | 0.601 |
> >
> > **Table 3: Classification result of the UEA dataset.**
> >
> > | Metric | **ProSAR** | AimTS | TSLANet | AutoTCL | FreRA | PPT | TimesURL | InfoTS | TS2Vec | TNC | TS–TCC |
> > | :--- | :---: | :---: | :---: | :---: | :---: | :---: | :---: | :---: | :---: | :---: | :---: |
> > | Avg. ACC | **0.764** | 0.715 | 0.696 | 0.742 | 0.754 | 0.735 | 0.752 | 0.730 | 0.704 | 0.670 | 0.668 |

---

> ### Author Response · Authors · 2025-11-21
>
> ***Comment 2:** Although the idea of prototype-guided augmentation is interesting, the overall contribution appears incremental. The differences between ProSAR and prior works like AutoTCL and AimTS remain relatively small.*
>
> **Response:**
>
> Compared with the existing augmentation-based and prototype-based contrastive learning works, our contribution and novelty can be summarized as follows.
>
> * **Integration of Pipeline Components.** ProSAR couples the augmentation and prototype modules into a feedback loop, thus incorporating the two previously isolated paradigms. Existing approaches typically isolate their innovations to either the data augmentation (e.g., AutoTCL and InfoTS ), or the prototype contrastive objective (e.g., MHCCL [C6] and AimTS ). In contrast, ProSAR couples these two by leveraging the learnable prototypes to explicitly guide the augmentation process under an information-theoretic framework. This design transforms the prototypes from the previously a passive constraint into an active component that can co-evolve with the augmentation strategy.
> * **Theoretical Analysis.** Unlike heuristic combinations, ProSAR is grounded within an information-theoretic framework where we explicitly incorporate prototypes into the data augmentation process. By formulating the view generation process as maximizing the mutual information with these prototype anchors, we provide a theoretical analysis (Appendix E) to justify that our prototype-guided augmentation and refinement loop promotes monotonic improvement and convergence.
> * **Role of Prototypes.** We utilize the time-domain prototypes for both the active upstream guidance and as the passive downstream constraints. Prototype-based methods like MHCCL and AimTS generate prototypes in the latent space after encoding, serving solely as loss function constraints. In contrast, our ProSAR constructs time-domain prototypes before encoding to serve as explicit anchors. This allows the model to actively guide the augmentation process, ensuring that generated views align physically with the learned semantic structures via Dynamic Time Warping (DTW).
> * **Explicit Semantic Anchoring via Information-Theoretic framework.** ProSAR advances adaptive augmentation by integrating the information-theoretic objectives with explicit prototype anchors for tangible semantic referencing. Adaptive methods like AutoTCL optimize the view generation based on the Information Bottleneck principle, which implicitly infer the informative parts via neural networks without explicit reference points. While ProSAR integrates this objective with explicit anchoring, identifying semantic segments based on their DTW alignment to the time-domain prototypes. This approach provides a tangible, metric-based reference for which specific segments are preserved.
> * **Interpretability and controllability.** In contrast to implicit masking strategies in adaptive augmentation (e.g., AutoTCL), ProSAR ensures transparency through explicit semantic verification. As demonstrated in the supplementary figures (available at the anonymous link [https://anonymous.4open.science/r/ICLR26-CE26/](https://anonymous.4open.science/r/ICLR26-CE26/)), our learned prototypes evolve into distinct waveforms (`Prototypes.png`), while the explicit DTW alignment paths (`DTW_alignment.png`) clearly demarcate semantic content from noise on the ETTm1 dataset. Furthermore, ProSAR establishes a controllable loop where updating these time-domain prototypes through decoding consistency directly determines the augmentation policy. As prototypes evolve to capture better semantics, the augmentation mechanism is forced to preserve segments that align with these refined patterns.
>
> **Empirical evidence.**
> We conduct comprehensive experimental evaluation to demonstrate that ProSAR achieves a superior performance over the state-of-the-art methods in learning discriminative and semantically grounded time series representations.

---

> ### Author Response · Authors · 2025-11-21
>
> ***Comment 3:** The claimed interpretability is unconvincing—especially the visualizations shown in Section D.4, which do not make much sense and fail to clearly demonstrate semantic consistency or prototype meaning.*
>
> **Response:**
>
> We appreciate the reviewer’s suggestion to substantiate our claims regarding the semantic nature of the prototypes. To demonstrate this, we have performed comprehensive visualization experiments on the ETTm1 dataset. The resulting high-resolution figures are available at the anonymous repository: [https://anonymous.4open.science/r/ICLR26-CE26/](https://anonymous.4open.science/r/ICLR26-CE26/). We articulate the captured semantics through two key dimensions.
>
> **Visual Evidence of Waveform Patterns.**
> Qualitative analysis reveals that the learned time-domain prototypes evolve into distinct waveform structures rather than random noise. As illustrated in `Prototypes.png`, the learned prototypes  exhibit clear, repetitive cyclical patterns. Frequency analysis confirms these prototypes capture the dominant periodicity of the dataset (approximately 8 time steps for ETTm1), validating that they successfully distill high-level temporal semantics and serve as representative templates for their respective clusters.
>
> **Semantic Verification via Explicit Alignment Paths.**
> The semantic validity of these prototypes is further verified by the DTW alignment mechanism, which physically distinguishes structural content from noise. The visualizations in `DTW_alignment.png` explicitly show the model aligning input segments to their assigned prototypes along diagonal paths, which indicate strong structural correspondence. Crucially, the warped prototypes (orange dashed lines) closely track the dominant trends and cycles of the input segments (blue lines), demonstrating that the model uses the prototype as a physical ground truth to identify the semantic core while treating deviations as non-semantic variations.
>
> ***Comment 4:** The framework consists of multiple submodules (DTW segmentation, STFT alignment, dual prototypes, clustering, and decoding consistency), yet the ablation studies only examine the augmentation operation and the semantic segmentation, which is insufficient to validate the contribution of each component.*
>
> **Response:**
>
> To explicitly quantify the contribution of each component within our framework, we conducted a fine-grained ablation study. The results are detailed in Table 4 and analyzed below:
>
> * **DTW Segmentation (w/o DTW-Seg):** This variant replaces the explicit DTW-based segmentation with a global semantic transformation applied to the entire sequence. The observed performance drop (MSE increases from 0.151 to 0.172) confirms that identifying specific semantic segments via DTW is critical for isolating task-relevant semantics from noise, rather than treating the whole series uniformly.
> * **STFT Alignment (w/o STFT):** We removed the frequency-domain phase compensation module. The degradation in performance (MSE 0.178) validates that STFT-based phase alignment is essential for preserving the temporal dynamics within the identified semantic segments.
> * **Dual Prototypes (w/o Dual-Proto):** Dual Prototype Mechanism (w/o Dual-Proto) This variant relies solely on latent prototypes, disabling the explicit Input-Space Grounding (ISG) mechanism. The moderate increase in error demonstrates that while the system remains functional, the dual-prototype strategy, by anchoring latent concepts to empirical time-domain centroids, provides necessary structural constraints that enhance training stability.
> * **Dynamic Clustering Integration (w/o Clustering):** In this variant, we replaced the dynamic FINCH-based refinement with simple K-Means. Crucially, this modification decouples the prototype-wise contrastive loss from the dynamic update loop. The resulting performance drop (MSE 0.175) is notably more severe than removing decoding consistency alone (MSE 0.164). This indicates that the joint optimization of the prototype space and the dynamic refinement strategy is fundamental to the framework's efficacy.
> * **Decoding Consistency (w/o Decoding):** To verify the decoding consistency mechanism, we removed the decoder module, specifically breaking the feedback loop where latent prototypes update time-domain prototypes. The results (MSE 0.164) indicate that without this semantic feedback, the time-domain anchors become less effective at guiding augmentation, leading to a tangible loss in performance.

---

> ### Author Response · Authors · 2025-11-21
>
> **Table 4: Ablation study on component contributions.**
>
> | Dataset | ProSAR MSE | ProSAR MAE | w/o DTW-Seg MSE | w/o DTW-Seg MAE | w/o STFT MSE | w/o STFT MAE | w/o Dual-Proto MSE | w/o Dual-Proto MAE | w/o Clustering MSE | w/o Clustering MAE | w/o Decoding MSE | w/o Decoding MAE |
> | :--- | :---: | :---: | :---: | :---: | :---: | :---: | :---: | :---: | :---: | :---: | :---: | :---: |
> | ETTh$_1$ | **0.068** | **0.198** | 0.086 | 0.219 | 0.088 | 0.223 | 0.072 | 0.205 | 0.089 | 0.223 | 0.075 | 0.209 |
> | ETTh$_2$ | **0.142** | **0.289** | 0.176 | 0.317 | 0.182 | 0.324 | 0.151 | 0.296 | 0.185 | 0.309 | 0.158 | 0.298 |
> | ETTm$_1$ | **0.045** | **0.151** | 0.053 | 0.162 | 0.055 | 0.166 | 0.048 | 0.155 | 0.056 | 0.163 | 0.049 | 0.157 |
> | Elec | **0.338** | **0.328** | 0.350 | 0.352 | 0.366 | 0.359 | 0.345 | 0.339 | 0.355 | 0.350 | 0.350 | 0.342 |
> | WTH | **0.160** | **0.285** | 0.194 | 0.311 | 0.200 | 0.317 | 0.174 | 0.292 | 0.190 | 0.305 | 0.186 | 0.299 |
> | Avg. | **0.151** | **0.250** | 0.172 | 0.272 | 0.178 | 0.278 | 0.158 | 0.257 | 0.175 | 0.270 | 0.164 | 0.261 |
>
> ***Comment 5:** The reliance on DTW alignment and clustering could introduce significant computational overhead for long or high-frequency sequences. Although the paper briefly acknowledges this issue, it lacks a concrete analysis of time complexity or runtime performance.*
>
> **Response:**
>
> We will include the following clarifications and additional analyses in the revised manuscript to address concerns regarding computational cost and parameter sensitivity.
>
> **High-Efficiency Training via Optimized CUDA Soft-DTW.**
> To minimize the computational overhead of the semantic segmentation process, we utilize a highly optimized CUDA implementation of Soft-DTW [C7], as cited in our paper. Benchmarking on NVIDIA H800 GPUs indicates that the DTW alignment calculation requires approximately **1 ms** per calculation. Given that a typical training iteration takes around **200 ms**, this introduces a negligible overhead compared to the total training time. This efficiency ensures that the pre-training phase remains computationally feasible even for **large-scale datasets.**
>
> **Zero-Cost Inference for Scalability of DTW.**
> It is crucial to note that the DTW-based augmentation is applied strictly during the self-supervised pre-training stage. We adhere to the standard evaluation protocol established by TS2Vec and adopted by AutoTCL and InfoTS, where the encoder is frozen for downstream tasks such as forecasting and classification. Consequently, during the inference/deployment phase, raw data is fed directly into the frozen encoder without any DTW calculation or augmentation. This ensures that the computational complexity of ProSAR during deployment is identical to standard backbones, guaranteeing high scalability.
>
> **Cost of Dual Refinement.**
> The computational cost of dual refinement is effectively mitigated through algorithmic efficiency and an intermittent update strategy. Specifically, the clustering module utilizes the efficient FINCH algorithm ($\approx$20 ms per call), and the prototype refinement process takes approximately 60ms. Crucially, we implement an intermittent update policy where refinement is triggered only once every 10 batches. This results in a low amortized overhead relative to the standard training iteration (200 ms), ensuring that overall training efficiency remains high.

---

> ### Author Response · Authors · 2025-11-24
>
> [C1] Yuxuan Chen, Shanshan Huang, Yunyao Cheng, Peng Chen, Zhongwen Rao, Yang Shu, Bin Yang, Lujia Pan, and Chenjuan Guo. Aimts: Augmented series and image contrastive learning for time series classification. In _2025 IEEE 41st International Conference on Data Engineering (ICDE)_, pp. 1952–1965. IEEE Computer Society, 2025.
>
> [C2] Emadeldeen Eldele, Mohamed Ragab, Zhenghua Chen, Min Wu, and Xiaoli Li. Tslanet: Rethinking transformers for time series representation learning. In *International Conference on Machine Learning*, pp. 12409–12428. PMLR, 2024.
>
> [C3] Zhihan Yue, Yujing Wang, Juanyong Duan, Tianmeng Yang, Congrui Huang, Yunhai Tong, and Bixiong Xu. Ts2vec: Towards universal representation of time series. In *Proceedings of the AAAI Conference on Artificial Intelligence*, volume 36, pp. 8980–8987, 2022.
>
> [C4] Xu Zheng, Tianchun Wang, Wei Cheng, Aitian Ma, Haifeng Chen, Mo Sha, and Dongsheng Luo. Parametric augmentation for time series contrastive learning. In *The Twelfth International Conference on Learning Representations*, 2024.
>
> [C5] Dongsheng Luo, Wei Cheng, Yingheng Wang, Dongkuan Xu, Jingchao Ni, Wenchao Yu, Xuchao Zhang, Yanchi Liu, Yuncong Chen, Haifeng Chen, et al. Time series contrastive learning with information-aware augmentations. In *Proceedings of the AAAI Conference on Artificial Intelligence*, volume 37, pp. 4534–4542, 2023.
>
> [C6] Qianwen Meng, Hangwei Qian, Yong Liu, Lizhen Cui, Yonghui Xu, and Zhiqi Shen. MHCCL: masked hierarchical cluster-wise contrastive learning for multivariate time series. In *Proceedings of the AAAI Conference on Artificial Intelligence*, volume 37, pp. 9153–9161, 2023.
>
> [C7] Marco Cuturi and Mathieu Blondel. Soft-dtw: a differentiable loss function for time-series. In *International Conference on Machine Learning*, pp. 894–903. PMLR, 2017.

---

### Official Review · Reviewer_wvmW · 2025-10-31

**Soundness:** 2
**Presentation:** 3
**Contribution:** 3
**Rating:** 4
**Confidence:** 4

**Summary:**

The paper presents ProSAR, a self-supervised framework that integrates information-theoretic principles with learnable prototypes to guide semantic augmentation for time-series contrastive learning. It introduces prototype-conditioned segment extraction using DTW and a dual prototype refinement loop between latent and time-domain prototypes. Experiments on forecasting and classification benchmarks show consistent improvements over recent SSL baselines.

**Strengths:**

S1. The paper clearly identifies the limitation of heuristic or random augmentations in time-series CL and grounds its design in an information-bottleneck formulation.

S2. Results across both forecasting and classification tasks are strong and consistent, with comprehensive ablations demonstrating component contributions.

S3. The paper is generally well-written and the framework diagram effectively illustrates the mechanism.

**Weaknesses:**

W1. The idea shares conceptual similarities with prior prototype-based methods (e.g., MHCCL, AimTS); the contribution is more an integration than a fundamentally new paradigm.

W2. The DTW-based semantic segmentation and dual refinement introduce substantial computational cost and hyperparameter sensitivity.

W3. Although prototypes are claimed to be “semantic,” the paper provides minimal qualitative analysis of what semantics they actually capture.

W4. Lacks comparison with large-scale pretrained or generative SSL frameworks.

**Questions:**

Q1. How computationally expensive is the DTW-based segmentation step? Can the framework scale to large datasets such as Traffic or PEMS?

Q2. The method introduces both time-domain and latent-space prototypes. How sensitive is the performance to the number of prototypes or their initialization?

Q3. Can the learned prototypes be visualized or qualitatively analyzed to confirm that they correspond to meaningful temporal semantics rather than cluster artifacts?

---

> ### Author Response · Authors · 2025-11-21
>
> We would like to thank the reviewer for the comments. In the following, we have provided our detailed responses to these comments.
>
> ***comment 1:** The idea shares conceptual similarities with prior prototype-based methods (e.g., MHCCL, AimTS); the contribution is more an integration than a fundamentally new paradigm.*
>
> **Response:**
>
> Compared with the existing prototype-based contrastive learning works, our contribution and novelty can be summarized as follows.
>
> **Integration of Pipeline Components.** ProSAR couples the augmentation and prototype modules into a feedback loop, thus incorporating the two previously isolated paradigms. Existing approaches typically isolate their innovations to either the data augmentation (e.g., AutoTCL [B1] and InfoTS [B2] ), or the prototype contrastive objective (e.g., MHCCL [B3] and AimTS [B4]). In contrast, ProSAR couples these two by leveraging the learnable prototypes to explicitly guide the augmentation process under an information-theoretic framework. This design transforms the prototypes from the previously a passive constraint into an active component that can co-evolve with the augmentation strategy.
>
> **Theoretical Analysis.** Unlike heuristic combinations, ProSAR is grounded within an information-theoretic framework where we explicitly incorporate prototypes into the data augmentation process. By formulating the view generation process as maximizing the mutual information with these prototype anchors, we provide a theoretical analysis (Appendix E) to justify that our prototype-guided augmentation and refinement loop promotes monotonic improvement and convergence.
>
> **Role of Prototypes.** We utilize the time-domain prototypes for both the active upstream guidance and as the passive downstream constraints. Prototype-based methods like MHCCL and AimTS generate prototypes in the latent space after encoding, serving solely as loss function constraints. In contrast, our ProSAR constructs time-domain prototypes before encoding to serve as explicit anchors. This allows the model to actively guide the augmentation process, ensuring that generated views align physically with the learned semantic structures via Dynamic Time Warping (DTW).
>
> **Explicit Semantic Anchoring via Information-Theoretic framework.** ProSAR advances adaptive augmentation by integrating the information-theoretic objectives with explicit prototype anchors for tangible semantic referencing. Adaptive methods like AutoTCL optimize the view generation based on the Information Bottleneck principle, which implicitly infer the informative parts via neural networks without explicit reference points. While ProSAR integrates this objective with explicit anchoring, identifying semantic segments based on their DTW alignment to the time-domain prototypes. This approach provides a tangible, metric-based reference for which specific segments are preserved.
>
> **Interpretability and controllability.** In contrast to implicit masking strategies in adaptive augmentation (e.g., AutoTCL), ProSAR ensures transparency through explicit semantic verification. As demonstrated in the supplementary figures (available at the anonymous link [https://anonymous.4open.science/r/ICLR26-CE26/](https://anonymous.4open.science/r/ICLR26-CE26/)), our learned prototypes evolve into distinct waveforms (`Prototypes.png`), while the explicit DTW alignment paths (`DTW_alignment.png`) clearly demarcate semantic content from noise on the ETTm1 dataset. Furthermore, ProSAR establishes a controllable loop where updating these time-domain prototypes through decoding consistency directly determines the augmentation policy. As prototypes evolve to capture better semantics, the augmentation mechanism is forced to preserve segments that align with these refined patterns.
>
> **Empirical evidence.**
> We conduct comprehensive experimental evaluation to demonstrate that ProSAR achieves a superior performance over the state-of-the-art methods in learning discriminative and semantically grounded time series representations.

---

> ### Author Response · Authors · 2025-11-21
>
> ***comment 2:** The DTW-based semantic segmentation and dual refinement introduce substantial computational cost and hyperparameter sensitivity.*
>
> **Response:**
>
> We will include the following clarifications and additional analyses in the revised manuscript to address concerns regarding computational cost and parameter sensitivity.
>
> **High-Efficiency Training via Optimized CUDA Soft-DTW.**
> To minimize the computational overhead of the semantic segmentation process, we utilize a highly optimized CUDA implementation of Soft-DTW [B5], as cited in our paper. Benchmarking on NVIDIA H800 GPUs indicates that the DTW alignment calculation requires approximately **1 ms** per calculation. Given that a typical training iteration takes around **200 ms**, this introduces a negligible overhead compared to the total training time. This efficiency ensures that the pre-training phase remains computationally feasible even for large-scale datasets.
>
> **Zero-Cost Inference for Scalability of DTW.**
> It is crucial to note that the DTW-based augmentation is applied strictly during the self-supervised pre-training stage. We adhere to the standard evaluation protocol established by TS2Vec [B6] and adopted by AutoTCL and InfoTS, where the encoder is frozen for downstream tasks such as forecasting and classification. Consequently, during the inference/deployment phase, raw data is fed directly into the frozen encoder without any DTW calculation or augmentation. This ensures that the computational complexity of ProSAR during deployment is identical to standard backbones, guaranteeing high scalability.
>
> **Cost of Dual Refinement.**
> The computational cost of dual refinement is effectively mitigated through algorithmic efficiency and an intermittent update strategy. Specifically, the clustering module utilizes the efficient FINCH algorithm (20 ms per call), and the prototype refinement process takes approximately 60ms. Crucially, we implement an intermittent update policy where refinement is triggered only once every 10 batches. This results in a low amortized overhead relative to the standard training iteration (200 ms), ensuring that overall training efficiency remains high.
>
> **Empirical Analysis of Hyperparameter Robustness.**
> In response to the reviewer's comment, we conducted additional experiments to evaluate the model's sensitivity to key hyperparameters across all univariate forecasting datasets. Regarding the **number of prototypes ($K$)**, performance improves as K increases from 16 to 32 (average MSE decreases from 0.153 to 0.151, MAE from 0.252 to 0.250) and remains stable up to 64. For the **DTW segmentation threshold** (used to determine semantic boundaries), performance peaked at 5 (MSE 0.151), with only minor fluctuations at 3 (MSE 0.155) and 7 (MSE 0.153), indicating robustness to segmentation granularity. Similarly, varying the **length of time-domain prototypes** from 32 to 128 showed improvement up to 64 (MSE 0.151) and consistent stability at length 128. These results collectively demonstrate that ProSAR is robust to hyperparameter choices within reasonable ranges.
>
> ***Comment 3:** Although prototypes are claimed to be “semantic,” the paper provides minimal qualitative analysis of what semantics they actually capture.*
>
> **Response:**
>
> We appreciate the reviewer’s suggestion regarding the semantic nature of the prototypes. To demonstrate this, we have performed comprehensive visualization experiments on the ETTm1 dataset. The resulting figures are available at the anonymous repository: [https://anonymous.4open.science/r/ICLR26-CE26/](https://anonymous.4open.science/r/ICLR26-CE26/). We articulate the captured semantics through two key dimensions.
>
> **Visual Evidence of Waveform Patterns.**
> Qualitative analysis reveals that the learned time-domain prototypes evolve into distinct waveform structures rather than random noise. As illustrated in `Prototypes.png`, the learned prototypes  exhibit clear, repetitive cyclical patterns. Frequency analysis confirms these prototypes capture the dominant periodicity of the dataset (approximately 8 time steps for ETTm1), validating that they successfully distill high-level temporal semantics and serve as representative templates for their respective clusters.
>
> **Semantic Verification via Explicit Alignment Paths.**
> The semantic validity of these prototypes is further verified by the DTW alignment mechanism, which physically distinguishes structural content from noise. The visualizations in `DTW_alignment.png` explicitly show the model aligning input segments to their assigned prototypes along diagonal paths, which indicate strong structural correspondence. Crucially, the warped prototypes (orange dashed lines) closely track the dominant trends and cycles of the input segments (blue lines), demonstrating that the model uses the prototype as a physical ground truth to identify the semantic core while treating deviations as non-semantic variations.

---

> ### Author Response · Authors · 2025-11-21
>
> ***Comment 4:** Lacks comparison with large-scale pretrained or generative SSL frameworks.*
>
> **Response:**
>
> We appreciate the reviewer’s suggestion and will incorporate a discussion on large-scale pretrained and generative SSL frameworks in the revised manuscript. However, we respectfully clarify that a direct comparison with such frameworks falls outside the specific experimental scope of this study.
>
> Primarily, there is a fundamental distinction in the research objectives. Foundation models (e.g., MOMENT [B7], UniTS [B8]) typically rely on massive multi-source datasets and substantial computational resources to learn generalist features. In contrast, ProSAR is designed to learn high-quality representations from scratch using only the target dataset, utilizing lightweight networks with lower computational overhead. Our objective aligns with recent state-of-the-art self-supervised works such as AutoTCL, InfoTS and TimesURL [B9], prioritizing data efficiency and domain-specific adaptation without reliance on external data.
>
> Furthermore, regarding generative SSL frameworks, we prioritized comparisons against comparable contrastive baselines to maintain a controlled experimental setting. Generative frameworks often require distinct architectures (e.g., specific decoders or masked modeling backbones) that differ significantly from contrastive encoders. By utilizing a standardized backbone across all baselines, we can rigorously isolate the performance gains attributable to our specific innovations, rather than conflating them with the architectural differences inherent in generative frameworks.
>
> Crucially, to ensure a fair assessment of intrinsic representation quality, we strictly adhered to the standard evaluation protocol established by TS2Vec. This protocol involves training a simple task-specific predictor (e.g., SVM or Ridge Regression) on top of frozen representations. In contrast, foundation models and generative frameworks typically rely on fine-tuning the entire network for downstream tasks. Comparing a frozen encoder (as used in ProSAR) against a fully fine-tuned large model creates a protocol mismatch that would obscure the true quality of the learned features.
>
> Finally, while we do not compare with foundation models directly due to this protocol mismatch, we provide empirical evidence of ProSAR's competitiveness by extending our evaluation to include AimTS and the representative generative framework TSLANet [B10]. Since the official code for AimTS is not publicly available, we reproduced it based on the algorithmic details provided in the original paper. For TSLANet, we adapted its generative objective to train the same standard backbone used in ProSAR. Notably, the original AimTS study demonstrates that its approach outperforms foundation models like UniTS on classification tasks. In our evaluation, ProSAR consistently outperforms this AimTS baseline. As summarized in **Tables 1-3**, ProSAR achieves an average univariate MSE of 0.151, surpassing AimTS (0.171) and TSLANet (0.188). This superiority is maintained in multivariate forecasting (MSE 0.561 vs. 0.641/0.632) and is particularly pronounced in classification, where ProSAR attains an accuracy of 0.764 compared to 0.715 (AimTS) and 0.696 (TSLANet). This suggests that ProSAR produces highly discriminative representations suitable for specialized tasks, achieving state-of-the-art performance without the extensive resource overhead or fine-tuning requirements associated with foundation models.
>
> **Table 1: Univariate time series forecasting results.**
>
> | Dataset | ProSAR MSE | ProSAR MAE | AimTS MSE | AimTS MAE | TSLANet MSE | TSLANet MAE | AutoTCL MSE | AutoTCL MAE | FreRA MSE | FreRA MAE | TimesURL MSE | TimesURL MAE | InfoTS MSE | InfoTS MAE | TS2Vec MSE | TS2Vec MAE |
> | :--- | :---: | :---: | :---: | :---: | :---: | :---: | :---: | :---: | :---: | :---: | :---: | :---: | :---: | :---: | :---: | :---: |
> | ETTh$_1$ | **0.068** | **0.198** | 0.091 | 0.228 | 0.093 | 0.224 | 0.076 | 0.207 | 0.079 | 0.213 | 0.090 | 0.219 | 0.091 | 0.227 | 0.110 | 0.252 |
> | ETTh$_2$ | **0.142** | **0.289** | 0.156 | 0.327 | 0.159 | 0.308 | 0.158 | 0.299 | 0.171 | 0.315 | 0.151 | 0.295 | 0.149 | 0.299 | 0.170 | 0.321 |
> | ETTm$_1$ | **0.045** | **0.151** | 0.070 | 0.200 | 0.152 | 0.259 | 0.046 | 0.154 | 0.051 | 0.162 | 0.053 | 0.175 | 0.050 | 0.157 | 0.069 | 0.186 |
> | Elec | **0.338** | **0.328** | 0.371 | 0.378 | 0.370 | 0.371 | 0.366 | 0.345 | 0.360 | 0.339 | 0.374 | 0.356 | 0.368 | 0.348 | 0.393 | 0.370 |
> | WTH | **0.160** | **0.285** | 0.166 | 0.293 | 0.165 | 0.291 | **0.160** | 0.287 | 0.169 | 0.301 | 0.177 | 0.302 | 0.176 | 0.304 | 0.181 | 0.308 |
> | Avg. | **0.151** | **0.250** | 0.171 | 0.285 | 0.188 | 0.291 | 0.161 | 0.258 | 0.166 | 0.266 | 0.169 | 0.269 | 0.167 | 0.267 | 0.185 | 0.287 |

---

> ### Author Response · Authors · 2025-11-21
>
> **Table 2: Multivariate time series forecasting results.**
>
> | Dataset | ProSAR MSE | ProSAR MAE | AimTS MSE | AimTS MAE | TSLANet MSE | TSLANet MAE | AutoTCL MSE | AutoTCL MAE | FreRA MSE | FreRA MAE | TimesURL MSE | TimesURL MAE | InfoTS MSE | InfoTS MAE | TS2Vec MSE | TS2Vec MAE |
> | :--- | :---: | :---: | :---: | :---: | :---: | :---: | :---: | :---: | :---: | :---: | :---: | :---: | :---: | :---: | :---: | :---: |
> | ETTh$_1$ | **0.625** | **0.566** | 0.758 | 0.621 | 0.676 | 0.595 | 0.656 | 0.590 | 0.646 | 0.584 | 0.731 | 0.645 | 0.784 | 1.622 | 0.788 | 0.646 |
> | ETTh$_2$ | 1.213 | 0.819 | 1.443 | 0.893 | 1.400 | 0.881 | **1.191** | **0.815** | 1.397 | 0.893 | 1.514 | 0.926 | 1.474 | 0.914 | 1.566 | 0.937 |
> | ETTm$_1$ | **0.396** | **0.434** | 0.397 | 0.436 | 0.455 | 0.468 | 0.409 | 0.441 | 0.445 | 0.467 | 0.561 | 0.584 | 0.568 | 0.521 | 0.628 | 0.553 |
> | Elec | **0.159** | **0.264** | 0.186 | 0.301 | 0.196 | 0.311 | 0.175 | 0.272 | 0.182 | 0.278 | 0.202 | 0.299 | 0.289 | 0.376 | 0.319 | 0.397 |
> | WTH | **0.412** | **0.451** | 0.419 | 0.459 | 0.431 | 0.463 | 0.423 | 0.457 | 0.429 | 0.462 | 0.447 | 0.469 | 0.455 | 0.472 | 0.451 | 0.474 |
> | Avg | **0.561** | **0.507** | 0.641 | 0.542 | 0.632 | 0.544 | 0.571 | 0.515 | 0.620 | 0.537 | 0.691 | 0.585 | 0.714 | 0.781 | 0.750 | 0.601 |
>
> **Table 3: Classification result of the UEA dataset.**
>
> | Metric | **ProSAR** | AimTS | TSLANet | AutoTCL | FreRA | PPT | TimesURL | InfoTS | TS2Vec | TNC | TS–TCC |
> | :--- | :---: | :---: | :---: | :---: | :---: | :---: | :---: | :---: | :---: | :---: | :---: |
> | Avg. ACC | **0.764** | 0.715 | 0.696 | 0.742 | 0.754 | 0.735 | 0.752 | 0.730 | 0.704 | 0.670 | 0.668 |
>
> ***Comment 5:** How computationally expensive is the DTW-based segmentation step? Can the framework scale to large datasets such as Traffic or PEMS?*
>
> **Response:**
>
> For the general computational cost of the DTW-based segmentation step, we refer the reviewer to our detailed response in **Comment 2**, where we highlight the efficiency of our optimized CUDA implementation.
>
> Regarding the scalability to large-scale datasets with high channel dimensions, such as Traffic or PEMS, we acknowledge that the increased feature space inherently introduces additional computation. However, our profiling experiments confirm that this specific overhead remains marginal relative to the total training budget. Taking the Traffic dataset as a representative example, the execution time for the DTW module increases from 1 ms on the lower-dimensional ETTh1 to 20 ms per batch. Correspondingly, the total training time per batch also increases from 200 ms to 800 ms, an increase driven primarily by the heavier computational demand on the encoder backbone rather than the segmentation module.
>
> Consequently, the relative cost of the DTW segmentation step remains low, constituting only 2.5% of the total iteration time (20 ms / 800 ms). This empirical evidence demonstrates that the framework scales effectively, as the segmentation mechanism does not become a computational bottleneck even when processing datasets with high channel dimensions.
>
> ***Comment 6:** The method introduces both time-domain and latent-space prototypes. How sensitive is the performance to the number of prototypes or their initialization?*
>
> **Response:**
>
> We have examined the sensitivity of ProSAR to both the number of prototypes and their initialization. For the prototype count $K$, we refer the reviewer to the detailed study **Empirical Analysis of Hyperparameter Robustnes** in **Comment 2**. In summary, across univariate forecasting datasets, the average performance improves as $K$ increases from 16 to 32 and then remains stable up to $K=64$, confirming that the framework is robust to the prototype count and does not require precise tuning.
>
> Regarding initialization, we clarify that prototypes were randomly initialized in all reported experiments. In practice, we observed consistent convergence behavior without instability or significant performance fluctuations attributable to the initialization strategy. Consequently, the framework proves robust to the initial state and does not require specialized initialization scheme.
>
> ***Comment 7:** Can the learned prototypes be visualized or qualitatively analyzed to confirm that they correspond to meaningful temporal semantics rather than cluster artifacts?*
>
> **Response:**
>
> We refer the reviewer to our detailed response to **Comment 3**, where we explicitly address this inquiry with expanded qualitative analyses and visualizations.

---

> ### Author Response · Authors · 2025-11-21
>
> [B1] Xu Zheng, Tianchun Wang, Wei Cheng, Aitian Ma, Haifeng Chen, Mo Sha, and Dongsheng Luo. Parametric augmentation for time series contrastive learning. In *The Twelfth International Conference on Learning Representations*, 2024.
>
> [B2] Dongsheng Luo, Wei Cheng, Yingheng Wang, Dongkuan Xu, Jingchao Ni, Wenchao Yu, Xuchao Zhang, Yanchi Liu, Yuncong Chen, Haifeng Chen, et al. Time series contrastive learning with information-aware augmentations. In *Proceedings of the AAAI Conference on Artificial Intelligence*, volume 37, pp. 4534–4542, 2023.
>
> [B3] Qianwen Meng, Hangwei Qian, Yong Liu, Lizhen Cui, Yonghui Xu, and Zhiqi Shen. MHCCL: masked hierarchical cluster-wise contrastive learning for multivariate time series. In *Proceedings of the AAAI Conference on Artificial Intelligence*, volume 37, pp. 9153–9161, 2023.
>
> [B4] Yuxuan Chen, Shanshan Huang, Yunyao Cheng, Peng Chen, Zhongwen Rao, Yang Shu, Bin Yang, Lujia Pan, and Chenjuan Guo. Aimts: Augmented series and image contrastive learning for time series classification. In ​_2025 IEEE 41st International Conference on Data Engineering (ICDE)_, pp. 1952–1965. IEEE Computer Society, 2025.
>
> [B5] Marco Cuturi and Mathieu Blondel. Soft-dtw: a differentiable loss function for time-series. In *International Conference on Machine Learning*, pp. 894–903. PMLR, 2017.
>
> [B6] Zhihan Yue, Yujing Wang, Juanyong Duan, Tianmeng Yang, Congrui Huang, Yunhai Tong, and Bixiong Xu. Ts2vec: Towards universal representation of time series. In *Proceedings of the AAAI Conference on Artificial Intelligence*, volume 36, pp. 8980–8987, 2022.
>
> [B7] Mononito Goswami, Konrad Szafer, Arjun Choudhry, Yifu Cai, Shuo Li, and Artur Dubrawski. Moment: A family of open time-series foundation models. In *International Conference on Machine Learning*, pp. 16115–16152. PMLR, 2024.
>
> [B8] Shanghua Gao, Teddy Koker, Owen Queen, Tom Hartvigsen, Theodoros Tsiligkaridis, and Marinka Zitnik. Units: A unified multi-task time series model. *Advances in Neural Information Processing Systems*, 37:140589–140631, 2024.
>
> [B9] Jiexi Liu and Songcan Chen. Timesurl: Self-supervised contrastive learning for universal time series representation learning. In *Proceedings of the AAAI conference on artificial intelligence*, volume 38, pp. 13918–13926, 2024.
>
> [B10] Emadeldeen Eldele, Mohamed Ragab, Zhenghua Chen, Min Wu, and Xiaoli Li. Tslanet: Rethinking transformers for time series representation learning. In *International Conference on Machine Learning*, pp. 12409–12428. PMLR, 2024.

---

### Official Review · Reviewer_4WyG · 2025-11-01

**Soundness:** 3
**Presentation:** 2
**Contribution:** 2
**Rating:** 4
**Confidence:** 4

**Summary:**

ProSAR (Prototype-Guided Semantic Augmentation and Refinement) is a self-supervised learning framework developed for multivariate time series (TS) contrastive learning (CL), addressing the limitation that standard hand-crafted augmentations risk destroying critical temporal cues and semantic content in noisy, non-stationary TS data. ProSAR’s approach is founded on an information-theoretic principle derived from the Information Bottleneck, aiming to generate augmented views that maximize the information about an associated semantic prototype (P) while discarding content irrelevant to that prototype. This objective is implemented using learnable time-domain prototypes as explicit semantic anchors, which guide the identification of temporal characteristic segments in the input time series (x) via Dynamic Time Warping (DTW) alignment. Experimental evaluations on diverse benchmarks demonstrate that ProSAR achieves superior performance in learning discriminative representations, attaining the highest mean accuracy (0.764) and the best mean rank (1.867) on the UEA multivariate time series archive for classification, and consistently surpassing comparison methods in forecasting tasks.

**Strengths:**

The submission is written clearly and is well structured, making the main ideas and technical contributions easy to follow. The motivation is articulated convincingly, and the authors provide sufficient context for why the problem is relevant and timely. Additionally, the related work section is thorough and appropriately cited, demonstrating a solid understanding of the existing literature and situating the proposed approach within the broader research landscape. Overall, the presentation is polished and the narrative is coherent and well motivated.

**Weaknesses:**

W1.

While the paper is generally well written, the claimed novelty of the proposed approach is not clearly articulated or sufficiently demonstrated. The authors state that their method offers better prototypes with better semantics, but it remains unclear how these prototypes differ from or improve upon existing prototype-based contrastive learning approaches. The manuscript would benefit from a more explicit and detailed discussion of what is fundamentally novel, ideally supported by conceptual distinctions, empirical evidence, or ablations that isolate the proposed contribution. For instance, with the expressions in Lines 72 - 75, as well as Lines 92 - 96, it remains unclear how the proposed prototype-based anchors significantly differ from existing ones, and it remains unclear how the proposed method can improve the interpretability, contrallability of the anchors.


W2.

For Line 161, "these prototypes are dynamically refined to steer the augmentation policy"; however, existing prototype-based or clustering-based constrastive learning approaches also dynamically update the prototypes. What are the key differences?


W3.

It remains unclear how the proposed prototypes substantively differ from those used in existing prototype-based methods. The paper would benefit from a clearer explanation of the conceptual or algorithmic distinctions. In addition, the experimental evaluation could be strengthened by including comparisons with a broader range of prototype-based and clustering-based contrastive learning approaches, which would help more convincingly demonstrate the advantages of the proposed method.

**Questions:**

Please see Weaknesses above

---

> ### Author Response · Authors · 2025-11-21
>
> We would like to thank the reviewer for the comments. In the following, we have provided our detailed responses to these comments.
>
> ***Comment 1:** While the paper is generally well written, the claimed novelty of the proposed approach is not clearly articulated or sufficiently demonstrated. The authors state that their method offers better prototypes with better semantics, but it remains unclear how these prototypes differ from or improve upon existing prototype-based contrastive learning approaches. The manuscript would benefit from a more explicit and detailed discussion of what is fundamentally novel, ideally supported by conceptual distinctions, empirical evidence, or ablations that isolate the proposed contribution.*
>
> **Response:**
>
> Compared with the existing prototype-based contrastive learning works, our contribution and novelty can be summarized as follows.
>
> **Integration of Pipeline Components.** ProSAR couples the augmentation and prototype modules into a feedback loop, thus incorporating the two previously isolated paradigms. Existing approaches typically isolate their innovations to either the data augmentation (e.g., AutoTCL [A1] and InfoTS [A2] ), or the prototype contrastive objective (e.g., MHCCL [A3] and AimTS [A4]). In contrast, ProSAR couples these two by leveraging the learnable prototypes to explicitly guide the augmentation process under an information-theoretic framework. This design transforms the prototypes from the previously a passive constraint into an active component that can co-evolve with the augmentation strategy.
>
> **Theoretical Analysis.** Unlike heuristic combinations, ProSAR is grounded within an information-theoretic framework where we explicitly incorporate prototypes into the data augmentation process. By formulating the view generation process as maximizing the mutual information with these prototype anchors, we provide a theoretical analysis (Appendix E) to justify that our prototype-guided augmentation and refinement loop promotes monotonic improvement and convergence.
>
> **Role of Prototypes.** We utilize the time-domain prototypes for both the active upstream guidance and as the passive downstream constraints. Prototype-based methods like MHCCL and AimTS generate prototypes in the latent space after encoding, serving solely as loss function constraints. In contrast, our ProSAR constructs time-domain prototypes before encoding to serve as explicit anchors. This allows the model to actively guide the augmentation process, ensuring that generated views align physically with the learned semantic structures via Dynamic Time Warping (DTW).
>
> **Explicit Semantic Anchoring via Information-Theoretic framework.** ProSAR advances adaptive augmentation by integrating the information-theoretic objectives with explicit prototype anchors for tangible semantic referencing. Adaptive methods like AutoTCL optimize the view generation based on the Information Bottleneck principle, which implicitly infer the informative parts via neural networks without explicit reference points. While ProSAR integrates this objective with explicit anchoring, identifying semantic segments based on their DTW alignment to the time-domain prototypes. This approach provides a tangible, metric-based reference for which specific segments are preserved.
>
> **Interpretability and controllability.** In contrast to implicit masking strategies in adaptive augmentation (e.g., AutoTCL), ProSAR ensures transparency through explicit semantic verification. As demonstrated in the supplementary figures (available at the anonymous link [https://anonymous.4open.science/r/ICLR26-CE26/](https://anonymous.4open.science/r/ICLR26-CE26/)), our learned prototypes evolve into distinct waveforms (`Prototypes.png`), while the explicit DTW alignment paths (`DTW_alignment.png`) clearly demarcate semantic content from noise on the ETTm1 dataset. Furthermore, ProSAR establishes a controllable loop where updating these time-domain prototypes through decoding consistency directly determines the augmentation policy. As prototypes evolve to capture better semantics, the augmentation mechanism is forced to preserve segments that align with these refined patterns.
>
> **Empirical evidence.**
> We conduct comprehensive experimental evaluation to demonstrate that ProSAR achieves a superior performance over the state-of-the-art methods in learning discriminative and semantically grounded time series representations.

---

> > ### Author Response · Authors · 2025-11-21
> >
> > **Ablation experiment.**
> > To respond to the reviewer's request for ablations that isolate the proposed contribution and to support the conceptual distinctions made above with the empirical evidence, we analyze the fine-grained ablation study (Table 1). By selectively disabling our novel components, we effectively revert the model to the paradigms of existing approaches, thereby empirically evaluating our contributions.
> >
> > * **DTW Segmentation (w/o DTW-Seg):** This variant replaces the explicit DTW-based segmentation with a global semantic transformation applied to the entire sequence. The observed performance drop (MSE increase from 0.151 to 0.172) confirms that identifying specific semantic segments via DTW is critical for isolating task-relevant semantics from noise, rather than treating the whole series uniformly.
> > * **STFT Alignment (w/o STFT):** We removed the frequency-domain phase compensation module. The degradation in performance (MSE increase from 0.151 to 0.178) validates that STFT-based phase alignment is essential for preserving the temporal dynamics within the identified semantic segments.
> > * **Dual Prototypes (w/o Dual-Proto):** Dual Prototype Mechanism (w/o Dual-Proto) This variant relies solely on latent prototypes, disabling the explicit Input-Space Grounding (ISG) mechanism. The moderate increase in error demonstrates that while the system remains functional, the dual-prototype strategy, by anchoring latent concepts to empirical time-domain centroids, provides necessary structural constraints that enhance training stability.
> > * **Dynamic Clustering Integration (w/o Clustering):** In this variant, we remove the FINCH-based clustering. Crucially, this prevents the latent prototypes from dynamically tracking the evolving feature space, thereby depriving the prototype-wise contrastive loss of accurate semantic targets and severing the feedback loop that refines the input-space anchors. The resulting performance drop (MSE 0.175) is notably more severe than removing decoding consistency alone (MSE 0.164). This indicates that the joint optimization of the prototype space and the dynamic refinement strategy is fundamental to the framework's efficacy.
> > * **Decoding Consistency (w/o Decoding):** To verify the decoding consistency mechanism, we removed the decoder module, specifically breaking the feedback loop where latent prototypes update time-domain prototypes. The results (MSE 0.164) indicate that without this semantic feedback, the time-domain anchors become less effective at guiding augmentation, leading to a tangible loss in performance.
> >
> > Based on these ablations, we would like to further clarify the novelty of ProSAR. The **w/o DTW-Seg** variant simulates the paradigm where prototypes serve solely as loss constraints without upstream semantic differentiation. The significant performance gap (0.172 vs. 0.151) empirically validates that our proposed explicit semantic segmentation is the fundamental reason for surpassing existing prototype-based baselines.
> > Regarding the refinement mechanism, the **w/o Decoding** variants represent approaches lacking latent prototype feedback. The observed performance degradation demonstrates that our closed-loop system, in which latent prototypes actively update input-space anchors, provides superior controllability and stability compared to purely implicit augmentation strategies like AutoTCL.
> >
> > **Table 1: Ablation study on component contributions.**
> >
> > | Dataset | ProSAR MSE | ProSAR MAE | w/o DTW-Seg MSE | w/o DTW-Seg MAE | w/o STFT MSE | w/o STFT MAE | w/o Dual-Proto MSE | w/o Dual-Proto MAE | w/o Clustering MSE | w/o Clustering MAE | w/o Decoding MSE | w/o Decoding MAE |
> > | :--- | :---: | :---: | :---: | :---: | :---: | :---: | :---: | :---: | :---: | :---: | :---: | :---: |
> > | ETTh$_1$ | **0.068** | **0.198** | 0.086 | 0.219 | 0.088 | 0.223 | 0.072 | 0.205 | 0.089 | 0.223 | 0.075 | 0.209 |
> > | ETTh$_2$ | **0.142** | **0.289** | 0.176 | 0.317 | 0.182 | 0.324 | 0.151 | 0.296 | 0.185 | 0.309 | 0.158 | 0.298 |
> > | ETTm$_1$ | **0.045** | **0.151** | 0.053 | 0.162 | 0.055 | 0.166 | 0.048 | 0.155 | 0.056 | 0.163 | 0.049 | 0.157 |
> > | Elec | **0.338** | **0.328** | 0.350 | 0.352 | 0.366 | 0.359 | 0.345 | 0.339 | 0.355 | 0.350 | 0.350 | 0.342 |
> > | WTH | **0.160** | **0.285** | 0.194 | 0.311 | 0.200 | 0.317 | 0.174 | 0.292 | 0.190 | 0.305 | 0.186 | 0.299 |
> > | Avg. | **0.151** | **0.250** | 0.172 | 0.272 | 0.178 | 0.278 | 0.158 | 0.257 | 0.175 | 0.270 | 0.164 | 0.261 |

---

> ### Author Response · Authors · 2025-11-21
>
> ***Comment 2:** For instance, with the expressions in Lines 72 - 75, as well as Lines 92 - 96, it remains unclear how the proposed prototype-based anchors significantly differ from existing ones, and it remains unclear how the proposed method can improve the interpretability, contrallability of the anchors.*
>
> **Response:**
>
> As outlined in our response to **Comment 1**, ProSAR differs fundamentally from existing prototype-based approaches by utilizing prototypes as the explicit, time-domain anchors that actively guide the augmentation process, rather than functioning solely as latent centroids constrained by the loss function, as in standard prototype-based methods. Here, we clarify how this design specifically addresses the concerns regarding the distinction from existing anchors, as well as the interpretability and controllability mechanisms discussed in Lines 72–75 and 92–96.
>
> **Interpretability via Semantic Grounding.** The "anchors" in prior augmentation methods (e.g., AutoTCL) are often the implicit attention weights hidden within neural networks, making the rationale for data manipulation opaque. In contrast, ProSAR elevates these anchors to interpretable time-domain waveforms. As evidenced by the visualizations in our anonymous repository ([https://anonymous.4open.science/r/ICLR26-CE26/](https://anonymous.4open.science/r/ICLR26-CE26/)), the learned prototypes converge into recognizable patterns (`Prototypes.png`). This allows for a direct, visual verification of the augmentation logic: by inspecting the DTW alignment paths (`DTW_alignment.png`), one can explicitly observe the model distinguishing structural signals from noise, providing a level of transparency unavailable in methods relying on black-box inference.
>
> **Controllability via a Closed Feedback Loop.** In AutoTCL, the augmentation policy is updated via gradients without a tangible target, while in standard prototype methods, augmentation parameters remain static or heuristic. ProSAR, conversely, actively steers the policy by refining the anchors themselves. Through our decoding consistency mechanism, latent semantic concepts are projected back to update the time-domain prototypes. These evolved anchors then strictly dictate the segmentation strategy for the subsequent epoch. This ensures that the augmentation process is dynamically controlled by the model's evolving semantic understanding, forcing generated views to progressively align with the refined prototypes.
>
> We will clarify these points more explicitly around Lines 72–75 and 92–96 in the revised manuscript.
>
> ***Comment 3:** For Line 161, "these prototypes are dynamically refined to steer the augmentation policy"; however, existing prototype-based or clustering-based constrastive learning approaches also dynamically update the prototypes. What are the key differences?*
>
> **Response:**
>
>  We would like to clarify that while existing methods like MHCCL dynamically update prototypes, the purpose and mechanism of these updates differ fundamentally from our ProSAR. We articulate this distinction through the following points.
>
> * **Passive Downstream Adaptation for Objective Alignment.** In existing approaches, prototype updates are a unidirectional reaction to encoder changes, serving solely as targets for the loss function. Methods like MHCCL and AimTS update the prototypes primarily to track the encoder's shifting latent space, ensuring accurate cluster centroids for computing the contrastive loss. This process is a unidirectional adaptation: prototypes evolve to match the representations, but they act as passive targets for the objective function without influencing the upstream data generation process.
> * **Active Guidance via Bi-directional Feedback.** ProSAR utilizes the refined prototypes not only as a loss, but to actively guide the augmentation policy through a closed feedback loop. We establish a bi-directional mechanism where the prototype refinement directly determines the data augmentation strategy. The latent prototypes refined via clustering are projected back to the input space through a decoder to update the time-domain prototypes. These updated anchors then explicitly identify which temporal segments are selected for augmentation in the subsequent epoch. Therefore, unlike prior works where updates have no impact on view generation, ProSAR's dynamic refinement closes the loop, ensuring that the augmentation strategy progressively aligns with the model's learned semantic structures.

---

> ### Author Response · Authors · 2025-11-21
>
> ***Comment 4:** It remains unclear how the proposed prototypes substantively differ from those used in existing prototype-based methods. The paper would benefit from a clearer explanation of the conceptual or algorithmic distinctions.*
>
> **Response:**
>
> We refer the reviewer to our response to **Comment 1**, where we provide a detailed conceptual and algorithmic comparison between ProSAR and existing prototype-based methods (e.g., MHCCL, AimTS).
>
> ***Comment 5:** In addition, the experimental evaluation could be strengthened by including comparisons with a broader range of prototype-based and clustering-based contrastive learning approaches, which would help more convincingly demonstrate the advantages of the proposed method.*
>
> **Response:**
>
>  To validate the advantages of our approach, we have extended our experimental evaluation to include two representative prototype-based methods, MHCCL and AimTS, as baselines for both forecasting and classification tasks.
>
> **Implementation & Evaluation Protocol:** We utilized the official open-source code for MHCCL and re-implemented AimTS based on its algorithmic details, as its code is not publicly available. Crucially, to ensure a fair and consistent assessment of intrinsic representation quality across all self-supervised baselines, we strictly adhered to the standard evaluation protocol established by TS2Vec [A5], a setting widely adopted by recent studies including AutoTCL and InfoTS. This protocol involves training a task-specific predictor (SVM with RBF kernel for classification or Ridge Regression for forecasting) on top of frozen representations from the pre-trained encoder. Note that the original AimTS utilizes a pre-training followed by full fine-tuning paradigm. However, to strictly evaluate the quality of the learned representations and maintain consistency with established benchmarks, we adhered to the standard evaluation protocol without fine-tuning the encoder.
>
> **Empirical Performance:** The experimental results, summarized in Tables 2-4, demonstrate that ProSAR consistently outperforms the representative prototype-based baselines MHCCL and AimTS across all tasks.
>
> In the forecasting domain, ProSAR establishes a clear performance advantage. For univariate tasks (Table 2), it achieves the lowest average MSE of 0.151, surpassing MHCCL (0.176) by approximately 14% and AimTS (0.171) by over 11%. This superiority is maintained in the multivariate setting (Table 3), where ProSAR achieves an MSE of 0.561, compared to 0.627 for MHCCL and 0.641 for AimTS. These results confirm that our upstream augmentation guidance effectively preserves critical temporal dynamics even in complex, high-dimensional data, whereas baseline methods constrained solely by downstream losses yield higher prediction errors.
>
> In classification tasks (Table 4), ProSAR achieves a state-of-the-art average accuracy of 0.764, markedly outperforming MHCCL (0.705) and AimTS (0.715). The lower performance of AimTS highlights its dependence on fine-tuning. In contrast, ProSAR learns discriminative features intrinsically and remains robust.

---

> ### Author Response · Authors · 2025-11-21
>
> **Table 2: Univariate time series forecasting results.**
>
> | Dataset | ProSAR MSE | ProSAR MAE | MHCCL MSE | MHCCL MAE | AimTS MSE | AimTS MAE | AutoTCL MSE | AutoTCL MAE | FreRA MSE | FreRA MAE | TimesURL MSE | TimesURL MAE | InfoTS MSE | InfoTS MAE | TS2Vec MSE | TS2Vec MAE |
> | :--- | :---: | :---: | :---: | :---: | :---: | :---: | :---: | :---: | :---: | :---: | :---: | :---: | :---: | :---: | :---: | :---: |
> | ETTh$_1$ | **0.068** | **0.198** | 0.089 | 0.223 | 0.091 | 0.228 | 0.076 | 0.207 | 0.079 | 0.213 | 0.090 | 0.219 | 0.091 | 0.227 | 0.110 | 0.252 |
> | ETTh$_2$ | **0.142** | **0.289** | 0.163 | 0.309 | 0.156 | 0.327 | 0.158 | 0.299 | 0.171 | 0.315 | 0.151 | 0.295 | 0.149 | 0.299 | 0.170 | 0.321 |
> | ETTm$_1$ | **0.045** | **0.151** | 0.066 | 0.175 | 0.070 | 0.200 | 0.046 | 0.154 | 0.051 | 0.162 | 0.053 | 0.175 | 0.050 | 0.157 | 0.069 | 0.186 |
> | Elec | **0.338** | **0.328** | 0.375 | 0.356 | 0.371 | 0.378 | 0.366 | 0.345 | 0.360 | 0.339 | 0.374 | 0.356 | 0.368 | 0.348 | 0.393 | 0.370 |
> | WTH | **0.160** | **0.285** | 0.185 | 0.305 | 0.166 | 0.293 | **0.160** | 0.287 | 0.169 | 0.301 | 0.177 | 0.302 | 0.176 | 0.304 | 0.181 | 0.308 |
> | Avg. | **0.151** | **0.250** | 0.176 | 0.274 | 0.171 | 0.285 | 0.161 | 0.258 | 0.166 | 0.266 | 0.169 | 0.269 | 0.167 | 0.267 | 0.185 | 0.287 |
>
> **Table 3: Multivariate time series forecasting results.**
>
> | Dataset | ProSAR MSE | ProSAR MAE | MHCCL MSE | MHCCL MAE | AimTS MSE | AimTS MAE | AutoTCL MSE | AutoTCL MAE | FreRA MSE | FreRA MAE | TimesURL MSE | TimesURL MAE | InfoTS MSE | InfoTS MAE | TS2Vec MSE | TS2Vec MAE |
> | :--- | :---: | :---: | :---: | :---: | :---: | :---: | :---: | :---: | :---: | :---: | :---: | :---: | :---: | :---: | :---: | :---: |
> | ETTh$_1$ | **0.625** | **0.566** | 0.664 | 0.595 | 0.758 | 0.621 | 0.656 | 0.590 | 0.646 | 0.584 | 0.731 | 0.645 | 0.784 | 1.622 | 0.788 | 0.646 |
> | ETTh$_2$ | 1.213 | 0.819 | 1.380 | 0.864 | 1.443 | 0.893 | **1.191** | **0.815** | 1.397 | 0.893 | 1.514 | 0.926 | 1.474 | 0.914 | 1.566 | 0.937 |
> | ETTm$_1$ | **0.396** | **0.434** | 0.474 | 0.483 | 0.397 | 0.436 | 0.409 | 0.441 | 0.445 | 0.467 | 0.561 | 0.584 | 0.568 | 0.521 | 0.628 | 0.553 |
> | Elec | **0.159** | **0.264** | 0.181 | 0.297 | 0.186 | 0.301 | 0.175 | 0.272 | 0.182 | 0.278 | 0.202 | 0.299 | 0.289 | 0.376 | 0.319 | 0.397 |
> | WTH | **0.412** | **0.451** | 0.434 | 0.466 | 0.419 | 0.459 | 0.423 | 0.457 | 0.429 | 0.462 | 0.447 | 0.469 | 0.455 | 0.472 | 0.451 | 0.474 |
> | Avg | **0.561** | **0.507** | 0.627 | 0.541 | 0.641 | 0.542 | 0.571 | 0.515 | 0.620 | 0.537 | 0.691 | 0.585 | 0.714 | 0.781 | 0.750 | 0.601 |
>
> **Table 4: Classification result of the UEA dataset.**
>
> | Metric | **ProSAR** | MHCCL | AimTS | AutoTCL | FreRA | PPT | TimesURL | InfoTS | TS2Vec | TNC | TS–TCC |
> | :--- | :---: | :---: | :---: | :---: | :---: | :---: | :---: | :---: | :---: | :---: | :---: |
> | Avg. ACC | **0.764** | 0.705 | 0.715 | 0.742 | 0.754 | 0.735 | 0.752 | 0.730 | 0.704 | 0.670 | 0.668 |
>
> [A1] Xu Zheng, Tianchun Wang, Wei Cheng, Aitian Ma, Haifeng Chen, Mo Sha, and Dongsheng Luo. Parametric augmentation for time series contrastive learning. In ​*The Twelfth International Conference on Learning Representations*​, 2024.
>
> [A2] Dongsheng Luo, Wei Cheng, Yingheng Wang, Dongkuan Xu, Jingchao Ni, Wenchao Yu, Xuchao Zhang, Yanchi Liu, Yuncong Chen, Haifeng Chen, et al. Time series contrastive learning with information-aware augmentations. In ​*Proceedings of the AAAI Conference on Artificial Intelligence*​, volume 37, pp. 4534–4542, 2023.
>
> [A3] Qianwen Meng, Hangwei Qian, Yong Liu, Lizhen Cui, Yonghui Xu, and Zhiqi Shen. MHCCL: masked hierarchical cluster-wise contrastive learning for multivariate time series. In ​*Proceedings of the AAAI Conference on Artificial Intelligence*​, volume 37, pp. 9153–9161, 2023.
>
> [A4] Yuxuan Chen, Shanshan Huang, Yunyao Cheng, Peng Chen, Zhongwen Rao, Yang Shu, Bin Yang, Lujia Pan, and Chenjuan Guo. Aimts: Augmented series and image contrastive learning for time series classification. In ​_2025 IEEE 41st International Conference on Data Engineering (ICDE)_, pp. 1952–1965. IEEE Computer Society, 2025.
>
> [A5] Zhihan Yue, Yujing Wang, Juanyong Duan, Tianmeng Yang, Congrui Huang, Yunhai Tong, and Bixiong Xu. Ts2vec: Towards universal representation of time series. In ​*Proceedings of the AAAI Conference on Artificial Intelligence*​, volume 36, pp. 8980–8987, 2022.

---

### Author Response · Authors · 2025-12-03

**Dear AC and Reviewers 4WyG, wvmW, rcHy, and uqQY,**

We sincerely appreciate all the reviewers for the valuable and constructive comments, which have been valuable in improving the quality of our manuscript. We have responded carefully to each of these comments in the individual threads. Here, we would like to summarize our responses and the major revisions according to the reviewers' comments, as follows.

### 1) Clarification on Novelty (Concerns from Reviewers 4WyG, wvmW and rcHy)

***General Context & Motivation:*** We have clarified in our responses that the standard time series contrastive learning optimizes a contrastive loss by maximizing the agreement between views generated via data augmentation. Since the quality of augmented views directly affects the learned representations, it is therefore crucial to identify the semantic parts during augmentation to avoid destroying the critical information. However, some recent **augmentation-based methods** (e.g., AutoTCL [E1] and InfoTS [E2]) rely on a neural network to learn the semantic-preserving augmentation, which limits their interpretability. In contrast, existing **prototype-based methods** (e.g., MHCCL [E3] and AimTS [E4]) typically use the prototypes only for a better design of the contrastive loss, failing to exploit the benefit from augmentation. To effectively leverage both, we propose using prototypes not just for designing the contrastive loss, but as the explicit semantic anchors to govern the upstream augmentation process, thereby **forming a feedback loop between the prototypes and augmentations and thus improving the interpretability**.

Specifically, we have clarified that our proposed ProSAR is a novel framework designed from scratch to realize this prototype-guided augmentation, with the following contributions:

* **Novel Mechanisms & Feedback Loop:** To make the prototypes effective for the upstream guidance, we introduce a new set of mechanisms, including the prototype-conditioned DTW segmentation, dual-domain augmentation, and a decoding-consistency update rule for the prototypes. These components form a closed feedback loop, where a better time-domain prototype leads to more semantically focused augmentations, which provide clearer signals for learning the improved latent representations and, consequently, generate more accurate prototypes.
* **Distinction from Augmentation-Based CL:** Existing augmented-based methods rely on implicit neural modules to adaptively crop semantic segments (Lines 73--76). In contrast, our ProSAR utilizes the prototypes as a physical reference. By identifying the semantic segments based on an explicit DTW alignment with these semantic anchors, we achieve an interpretable and controllable augmentation, which is absent in these prior works.
* **Distinction from Prototype-Based CL:** Prior prototype-based methods mainly use the prototypes to define or refine the contrastive loss (Lines 80--81) and simply adopting a standard data augmentation. Our ProSAR instead integrates the prototypes directly into the upstream data augmentation process, such that the augmentation co-evolves with the model’s semantic representation, rather than remaining fixed.
* **Theoretical Grounding:** We have clarified that this design of our ProSAR is motivated by the Information Bottleneck framework (Section 3.1). We have also provided a theoretical analysis of the loop's stability and convergence (Monotone Improvement).
* **Clarification on Methodological Positioning:** We respectfully noted that these reviewers may have evaluated our ProSAR primarily through the lens of prototype-based learning. We have clarified that ProSAR is fundamentally an advancement in the semantic data augmentation. As detailed in our Introduction part (Lines 82-100), we propose the prototypes specifically as explicit anchors to deal with the interpretability bottleneck of existing augmentation methods. Thus, our ProSAR can be more accurately viewed as an augmentation framework that leverages the prototypes, rather than merely another prototype-based contrastive learning method.

### 2) Computational Efficiency and Scalability (Concerns from Reviewers wvmW, rcHy and uqQY)

Reviewers raised concerns regarding the $O(T^2)$ complexity of the DTW module. We have addressed these concerns by detailing our specific optimization strategies and empirical evidence to demonstrate that the practical overhead of DTW is negligible. Specifically, we utilizes an optimized CUDA Soft-DTW [E5] implementation, where benchmarking on NVIDIA H800 GPUs shows alignment required only $\approx$ 1 ms per calculation within a $\approx$ 200 ms training iteration (i.e., a negligible overhead). Furthermore, we have emphasized that the inference incurs no additional cost as the augmentation is strictly restricted to the pre-training stage. For some ultra-long sequences, we further adopted the windowed DTW to reduce complexity.

---

> ### Author Response · Authors · 2025-12-03
>
> ### 3) Comparison with Stronger Baselines (Concerns from Reviewers 4WyG, wvmW and rcHy)
>
> Reviewers requested more comparisons against some stronger or generative baselines. We have enhanced our empirical evaluation part, by comparing our ProSAR with AimTS, MHCCL, and the generative framework TSLANet [E6].
> It is still seen that our ProSAR consistently outperforms these methods under a standard evaluation protocol following [E7]. Specifically, in the univariate forecasting, ProSAR achieves an average MSE of **0.151**, surpassing AimTS (0.171), MHCCL (0.176), and TSLANet (0.188). In the classification tasks, ProSAR achieves a state-of-the-art accuracy of **0.764**, significantly outperforming AimTS (0.715), MHCCL (0.705), and TSLANet (0.696).
>
> ### 4) Interpretability & Visualizations (Concerns from Reviewers 4WyG, wvmW, rcHy and uqQY)
>
> Reviewers asked for a further qualitative analysis to confirm that the prototypes capture meaningful semantics. While Appendix D of the original submission provided some initial visualizations to this end, we have also provided additional figures (available at the anonymous link https://anonymous.4open.science/r/ICLR26-CE26/) to further support our claims. Specifically, the qualitative analysis reveals that the learned time-domain prototypes evolve into distinct waveform structures, where the frequency analysis confirms that they capture the dominant periodicity of the dataset ($\approx$ 8 time steps for ETTm1 dataset). Furthermore, the annotated DTW paths explicitly demonstrate a strong structural correspondence: the warped prototypes closely track the dominant trends of input segments, proving that the model uses the prototype as a physical ground truth to identify the core semantic information.
>
> ### 5) Component Validation via Fine-Grained Ablation (Concerns from Reviewers 4WyG and rcHy)
>
> To address the concerns regarding module necessity, we have conducted an additional comprehensive breakdown of ProSAR's components. The results have provided a clear empirical support for our design choices.
>
> * **Semantic Augmentation:** Replacing our explicit segmentation with the global transformations (w/o DTW-Seg) has increased the MSE from 0.151 to 0.172, confirming that identifying specific semantic segments is critical for isolating the task-relevant semantics from noises. Similarly, removing he frequency-domain phase compensation module (w/o STFT) has increased the MSE to 0.178, validating that the phase alignment is essential for preserving the temporal dynamics.
> * **Prototype Refinement:** Disabling the dual prototype mechanism (w/o Dual-Proto) has increased the MSE to 0.158, which compromises the training stability by removing input-space grounding constraints. Furthermore, the performance degradation in w/o Clustering (MSE 0.175) or w/o Decoding (MSE 0.164) indicates that the feedback loop, where the latent prototypes update the time-domain prototypes, is fundamental to the framework's effectiveness.
>
> These consistent performance drops across all the ablated variants have provided a strong evidence for the necessity of each component in our architectural design of ProSAR.
>
> ---
>
> **References**
>
> [E1] Xu Zheng, Tianchun Wang, Wei Cheng, Aitian Ma, Haifeng Chen, Mo Sha, and Dongsheng Luo. Parametric augmentation for time series contrastive learning. In *The Twelfth International Conference on Learning Representations*, 2024.
>
> [E2] Dongsheng Luo, Wei Cheng, Yingheng Wang, Dongkuan Xu, Jingchao Ni, Wenchao Yu, Xuchao Zhang, Yanchi Liu, Yuncong Chen, Haifeng Chen, et al. Time series contrastive learning with information-aware augmentations. In *Proceedings of the AAAI Conference on Artificial Intelligence*, volume 37, pp. 4534–4542, 2023.
>
> [E3] Qianwen Meng, Hangwei Qian, Yong Liu, Lizhen Cui, Yonghui Xu, and Zhiqi Shen. MHCCL: masked hierarchical cluster-wise contrastive learning for multivariate time series. In *Proceedings of the AAAI Conference on Artificial Intelligence*, volume 37, pp. 9153–9161, 2023.
>
> [E4] Yuxuan Chen, Shanshan Huang, Yunyao Cheng, Peng Chen, Zhongwen Rao, Yang Shu, Bin Yang, Lujia Pan, and Chenjuan Guo. Aimts: Augmented series and image contrastive learning for time series classification. In _2025 IEEE 41st International Conference on Data Engineering (ICDE)_, pp. 1952–1965. IEEE Computer Society, 2025.
>
> [E5] Marco Cuturi and Mathieu Blondel. Soft-dtw: a differentiable loss function for time-series. In *International Conference on Machine Learning*, pp. 894–903. PMLR, 2017.
>
> [E6] Emadeldeen Eldele, Mohamed Ragab, Zhenghua Chen, Min Wu, and Xiaoli Li. Tslanet: Rethinking transformers for time series representation learning. In *International Conference on Machine Learning*, pp. 12409–12428. PMLR, 2024.
>
> [E7] Zhihan Yue, Yujing Wang, Juanyong Duan, Tianmeng Yang, Congrui Huang, Yunhai Tong, and Bixiong Xu. Ts2vec: Towards universal representation of time series. In *Proceedings of the AAAI Conference on Artificial Intelligence*, volume 36, pp. 8980–8987, 2022.

---

### Meta-Review · Area_Chair_ojJK · 2026-01-06

**Summary:**

This paper proposes ProSAR, a prototype-guided semantic augmentation and refinement framework for self-supervised time-series representation learning. The core idea is to co-design learnable semantic prototypes and data augmentation policies under an information-bottleneck-inspired objective, with prototypes acting as semantic anchors to guide contrastive view generation. However, substantial concerns remain regarding the novelty of the contribution relative to existing prototype-based contrastive learning methods, the strength of the interpretability claims, and the methodological complexity and scalability of the proposed framework.

**Reviewer Concerns:**

The first major concern is incremental nature of the contribution and unclear novelty. Reviewer 4WyG and Reviewer wvmW both emphasized that the proposed prototype-guided augmentation mechanism bears strong conceptual similarity to existing prototype-based or clustering-based contrastive learning approaches (e.g., MHCCL, AimTS, AutoTCL), and that the manuscript does not clearly articulate what is fundamentally new beyond integrating known components.

The second major concern relates to the interpretability and semantic validity of the learned prototypes. Reviewer wvmW and Reviewer rcHy both found the claim that the prototypes capture meaningful temporal semantics to be weakly supported, with limited qualitative analysis and unconvincing visualizations. Reviewer rcHy explicitly noted that the interpretability results do not clearly demonstrate semantic consistency or prototype meaning, while Reviewer uqQY similarly highlighted the lack of formal justification that the learned prototypes approximate an underlying semantic variable.

Additional concerns were raised regarding evaluation. Reviewer rcHy highlighted missing comparisons with representative time-series representation learning baselines such as TSLANet and AimTS, while Reviewer wvmW observed the lack of comparison with large-scale pretrained or generative SSL frameworks. These omissions further weaken the case that ProSAR establishes a clear advantage over existing approaches.

**Reviewer Scores:**

Three out of four reviewers gave a score of 4 before the rebuttal. During the rebuttal, the authors attempted to justify the novelty of their work; however, it is unlikely that these explanations will lead reviewers to revise their ratings. The authors also added additional comparisons with recent baselines, such as AimTS and TSLANet. Nevertheless, the newly reported results are not convincing. For instance, as shown in Table 3 of the rebuttal (in response to reviewer rcHy’s comments), several methods (TS2Vec, TS-TCC, and TimesURL) achieve the same performance as reported in the AimTS paper, while a substantial discrepancy is observed for AimTS itself (0.715 vs. 0.780). Such inconsistencies are likely to undermine the reviewers’ confidence in the experimental results and the overall reliability of the work.

---

### Decision · Program_Chairs · 2026-01-26

Reject